EMBO
Molecular Medicine

# Proteome profiles of esophageal squamous cell carcinoma tie mitochondrial complex I to immunotherapy

Fahan Ma[1,9], Yan Li [ID][2,9], Chan Xiang[3,9], Bing Wang[1,9], Jie Lv[4,9], Zhanxian Shang[3,9], Weiguang Zhang[5,9], Zhaoyu Qin[1], Yan Pu[1], Kai Li[1], Jinzhi Wei[3], Su-bei Tan[1], Jinwen Feng[1], Haohua Teng[3], Peipei Zhang[5], Jiaying Deng[6,7,8], Yunzhi Wang[1], Chao Zhang[6,7], Sha Tian [ID][1], Guichao Li[6,7], Mingqiang Kang[5✉], Changsheng Du [ID][4✉], Yuchen Han[3✉] & Chen Ding [ID][1✉]

## Abstract

Immunotherapy has revolutionized cancer treatment, yet many patients show non-sensitivity. Here, we collected treatment-naïve samples from 190 esophageal squamous cell carcinoma (ESCC) patients undergoing anti-programmed death 1 (PD1) immunotherapy for proteome, phosphoproteome, and immunohistochemistry (IHC) analysis. Proteome-based stratification of ESCC identifies three proteomic subtypes (G-I-G-III) related to immunotherapy response and different molecular features, revealing that patients with high mitochondrial complex I protein expression show sensitivity to anti-PD1 immunotherapy. High mitochondrial complex I protein expression of ESCC cells or patient-derived organoids increases sensitivity to CD8 + T cell-mediated killing in the co-culture systems. Phosphoproteomic data analysis reveals YAP1 activation impairs immunotherapy efficacy. Inhibiting YAP1 or increasing mitochondrial complex I levels bolsters immunotherapy effectiveness in ESCC allograft tumors. Finally, we develop a highly accurate predictive model (AUC ≥ 0.90) by the signatures of mitochondrial complex I-mediated anti-tumor immune response and validate it in independent cohorts. This study provides a rich resource for investigating the mechanisms and indicators of immunotherapy in ESCC.

**Subject Categories** Biomarkers; Cancer; Proteomics

## Introduction

Esophageal cancer (EC) is one of the most aggressive cancer types of the digestive system and is the sixth most common cause of cancer-related death worldwide (Sung et al, 2021). It comprises two histological subtypes: esophageal squamous cell carcinoma (ESCC) and esophageal adenocarcinoma (EAC). ESCC is the predominant histological type of EC in China (Allemani et al, 2018; Pennathur et al, 2013), accounting for more than 90% of cases. Apart from aging and male sex, the risk factors associated with ESCC include unfavorable lifestyle, alcohol consumption, and smoking, etc. So far, the primary treatment of ESCC patients is chemotherapy, and there is still no effective targeted treatment for ESCC patients (Li et al, 2021). Immunotherapy has demonstrated remarkable anticancer activity and has promoted the recent approval of anti-programmed death 1 (PD1)/PD-L1 drugs in several solid tumors, especially in melanoma, achieving a 52% objective response rate (ORR) (Hamid et al, 2019). Recently, the immune checkpoint inhibitor camrelizumab, a humanized, selective IgG4-κ monoclonal antibody against PD1, has been approved as the first-line therapy in ESCC patients (Liu et al, 2022b; Zhao et al, 2023). However, the ORR for the first-line therapy was 54.2% in ESCC patients (Zhao et al, 2023), suggesting that nearly half of ESCC patients could not benefit from the immunotherapy and some patients could develop recurrence after immunotherapy. The biomarkers of immunotherapy, such as expression of checkpoint proteins, mutation load, neoepitope load, T-cell receptor clonality, and immune gene signature, correlate with response rate but largely overlap between sensitive and non-sensitive patients with low predictive values (Gibney et al, 2016). Many biomarkers have not been validated in the clinic. PD-L1 expression, as assessed by immunohistochemistry

[1]Clinical Research Center for Cell-based Immunotherapy of Shanghai Pudong Hospital, Fudan University Pudong Medical Center, State Key Laboratory of Genetics and Development of Complex Phenotypes, School of Life Sciences, Human Phenome Institute, Fudan University, Shanghai, China. [2]Department of Laboratory Medicine, Institute of Laboratory Medicine, Sichuan Provincial People's Hospital, School of Medicine, University of Electronic Science and Technology of China, Chengdu, China. [3]Department of Pathology, Shanghai Chest Hospital, Shanghai Jiao Tong University School of Medicine, Shanghai, China. [4]Key Laboratory of Spine and Spinal Cord Injury Repair and Regeneration of Ministry of Education, Orthopaedic Department of Tongji Hospital, School of Life Sciences and Technology, Tongji University, Shanghai, China. [5]Department of Thoracic Surgery, Fujian Medical University Union Hospital, Fuzhou, China. [6]Department of Radiation Oncology, Departments of Thoracic Surgery and State Key Laboratory of Genetic Engineering, Fudan University Shanghai Cancer Center, Shanghai, China. [7]Department of Oncology, Institute of Thoracic Oncology, Shanghai Medical College, Fudan University, Shanghai, China. [8]Shanghai Clinical Research Center for Radiation Oncology, Shanghai Key Laboratory of Radiation Oncology, Shanghai, China. [9]These authors contributed equally: Fahan Ma, Yan Li, Chan Xiang, Bing Wang, Jie Lv, Zhanxian Shang, Weiguang Zhang. ✉E-mail: 9199115045@fjmu.edu.cn; duchangsheng@tongji.edu.cn; ychan@cmu.edu.cn; chend@fudan.edu.cn

(IHC), is one of the companion diagnostic markers approved to guide anti-PD1 immunotherapy (Davis and Patel, 2019); however, the correlation between PD-L1 expression and response rate is controversial, possibly due to tumor heterogeneity. There was low specificity for the PD-L1 IHC score to detect immunotherapy responders across tumor types (Banchereau et al, 2021). No difference in ORR was observed between the PD-L1-positive and negative expression groups in some clinical trials (Luo et al, 2021; Sharma et al, 2016; Sun et al, 2021), among which patients with PD-L1-negative expression were also recruited to receive immunotherapy and could also respond to immunotherapy. These clinical trials showed that PD-L1 testing alone was insufficient for patient selection to receive immunotherapy in some cancer types. Therefore, it is crucial to identify reliable immunotherapy biomarkers to develop precision medicine for patients who could benefit from immunotherapy and to better understand the mechanisms of sensitivity and resistance.

Previous studies have investigated immunotherapy for cancer at the genomic and transcriptomic levels. Le et al found a large proportion of mutant neoantigens in mismatch repair-deficient cancers, making them sensitive to immunotherapy across 12 different tumor types (Le et al, 2017). Liu et al analyzed melanoma anti-PD1 immunotherapy by whole-exome and whole-transcriptome (Liu et al, 2019) and identified multiple novel genomic and transcriptomic features associated with immunotherapy response, such as MHC-I and MHC-II antigen presentation. The genomic profiling of ESCC identified mutational signatures associated with survival and tumor metastasis (Cui et al, 2022), and the integrated analysis of CNA, DNA methylation, mRNA, and microRNA expression data by The Cancer Genome Atlas (TCGA) defined molecular subgroups and offered potential therapeutic targets (Cancer Genome Atlas Research et al, 2017). However, these efforts did not explore the potential mechanisms of ESCC immunotherapy. Recently, mass spectrometry (MS)-based proteomics has been applied in tumor immunotherapy research. Harel et al examined melanoma immunotherapy response using MS-based proteomics and identified therapy response biomarkers and cellular mechanisms of immunotherapy (Harel et al, 2019), suggesting that proteomics could identify immunotherapy-related proteins (the "executors of life") and explore the potential mechanisms of ESCC immunotherapy.

Herein, aiming to better understand the mechanism of ESCC immunotherapy and identify predictive biomarkers for immunotherapy, we conducted a comprehensive proteomic analysis to investigate the response to first-line therapy (camrelizumab-based immunotherapy) for ESCC. We collected tumor biopsies from 190 treatment-naïve ESCC patients and constructed an ESCC anti-PD1 immunotherapy cohort, including the discovery cohort (53 patients), the validation cohort (55 patients), and the IHC validation cohort (82 patients). Proteomic clustering revealed that three molecular subtypes (G-I-G-III) with distinct molecular functions were associated with immunotherapy response. The G-I subtype was a sensitive subtype with high immune and mitochondrial features, the G-II subtype was a mixed subtype enriched in RNA metabolism, and the G-III subtype was a non-sensitive subtype involved in the platelet activation bioprocess. Among the three subtypes, we focused on the sensitive subtype (the G-I subtype) and found that patients with high mitochondrial complex I protein expression showed sensitivity to immunotherapy by increasing MHC-I molecule expression to present

antigen to CD8 + T cells. Further analysis of the phosphoproteomic data revealed that the inactivation of the hippo pathway in non-sensitive patients increased the YAP1 activity. The high mitochondrial complex I protein expression of ESCC cells or patient-derived organoids increases sensitivity to CD8 + T cell-mediated killing in the in vitro co-culture systems. The syngeneic mouse model of ESCC also indicated that tumors with high mitochondrial complex I protein expression could be more responsive to anti-PD1 immunotherapy, and the YAP1 inhibitor demonstrated a synergistic effect when combined with anti-PD1 antibody, resulting in an augmented anti-tumor efficacy in vivo. Furthermore, we developed a predictive model based on the four protein signatures (YAP1, NDUFB7, MHC-I, and CD8A) with high accuracy to predict the immunotherapy response in the discovery cohort, which was validated in an independent validation cohort and the IHC validation cohort. Collectively, this study provides a comprehensive proteomic analysis of ESCC immunotherapy and implicates its predictive and therapeutic significance as well as the putative underlying regulatory mechanism that may benefit clinical practice.

# Results

## Characteristics of the ESCC immunotherapy cohort

To investigate the proteomic patterns associated with immunotherapy response and develop a predictive model with high predictive ability for ESCC immunotherapy, we collected 190 ESCC patients, including the discovery cohort (53 patients), the validation cohort (55 patients), and the IHC validation cohort (82 patients), all of whom received camrelizumab-based anti-PD1 immunotherapy as the first-line therapy followed by surgery in clinic. In the discovery cohort, most patients were male (84.91%) and over 60 years old (81.13%), which was consistent with the characteristics of ESCC patients enrolled in the clinical trial (Yin et al, 2023). The detailed clinical characteristics are shown in Table 1 and Dataset EV1, and all the immunotherapy regimens were given at standard dosing as described in previous studies (Methods). All samples were histologically scored by the expert digestive system pathologists (ZXS, CX, YCH, WGZ, and MQK) according to the tumor regression grade (TRG) assessment criteria through measurement of the percentage of residual viable tumor on the resected tumor specimen after immunotherapy using previously reported methods (Liu et al, 2022b; Yang et al, 2022), which provide a more accurate evaluation of residual viable tumor after immunotherapy than the CT-based RECIST criteria (Chiou and Burotto, 2015; Di Giacomo et al, 2009; Forde et al, 2018). It also overcomes limitations of CT scans, such as pseudoprogression caused by the immune cell infiltration within the tumor. We grouped patients into: TRG-I, without residual tumor; TRG-II, 1–10% residual tumor; TRG-III, 11– 50% residual tumor; TRG-IV, > 50% residual tumor. Patients with TRG-I who had the pathological complete response (pCR) due to the absence of residual viable tumor were defined as sensitive (S), and those with TRG-II-TRG-IV were defined as non-sensitive (NS). The discovery cohort included 24 sensitive patients (TRG-I, $N = 24$) and 29 non-sensitive patients (TRG-II, $N = 9$; TRG-III, $N = 8$; TRG-IV, $N = 12$). The validation cohort comprised 19 sensitive patients (TRG-I, $N = 19$) and 36 non-sensitive patients (TRG-II, $N = 16$; TRG-III, $N = 6$; TRG-IV, $N = 14$). The IHC validation cohort comprised 30 sensitive patients (TRG-I, $N = 30$) and

**Table 1. ESCC patient demographics and baseline characteristics.**

| | S (N = 24) | NS (N = 29) | All (N = 53) | P value |
|---|---|---|---|---|
| Age/years | | | | 0.09 |
| ≤60 | 2 (8.33%) | 8 (27.59%) | 10 (18.87%) | |
| >60 | 22 (91.67%) | 21 (72.41%) | 43 (81.13%) | |
| Gender | | | | 0.44 |
| Female | 5 (20.83%) | 3 (10.34%) | 8 (15.09%) | |
| Male | 19 (79.17%) | 26 (89.66%) | 45 (84.91%) | |
| Smoking | | | | 0.27 |
| No | 6 (25.00%) | 3 (10.34%) | 9 (16.98%) | |
| Yes | 18 (75.00%) | 26 (89.66%) | 44 (83.02%) | |
| Drinking | | | | 0.09 |
| No | 13 (54.17%) | 8 (27.59%) | 21 (39.62%) | |
| Yes | 11 (45.83%) | 21 (72.41%) | 32 (60.38%) | |
| Tumor location | | | | 0.54 |
| Lower | 2 (8.33%) | 3 (10.34%) | 5 (9.43%) | |
| Middle | 17 (70.83%) | 16 (55.17%) | 33 (62.26%) | |
| Upper | 5 (20.83%) | 10 (34.48%) | 15 (28.30%) | |
| Histological grade | | | | 0.05 |
| Moderately differentiated | 8 (33.33%) | 18 (62.07%) | 26 (49.06%) | |
| Poorly differentiated | 14 (58.33%) | 11 (37.93%) | 25 (47.17%) | |
| Well differentiated | 2 (8.33%) | 0 (0.00%) | 2 (3.77%) | |
| PD-L1 expression | | | | 0.28 |
| Negative | 12 (50.00%) | 10 (34.48%) | 22 (41.51%) | |
| Positive | 12 (50.00%) | 19 (65.52%) | 31 (58.49%) | |
| cStage | | | | 0.11 |
| III | 19 (79.17%) | 16 (55.17%) | 35 (66.04%) | |
| IVa | 5 (20.83%) | 13 (44.83%) | 18 (33.96%) | |
| TRG | | | | <0.001 |
| I | 24 (100.00%) | 0 (0.00%) | 24 (45.28%) | |
| II | 0 (0.00%) | 9 (31.03%) | 9 (16.98%) | |
| III | 0 (0.00%) | 8 (27.59%) | 8 (15.09%) | |
| IV | 0 (0.00%) | 12 (41.38%) | 12 (22.64%) | |

*TRG* tumor regression grade.
PD-L1 expression level was assessed by immunohistochemical staining intensity and quantified as tumor proportion score, which was defined as the percentage of staining tumor cells relative to all viable tumor cells present in the sample. A tumor proportion score of 1% was used as the cutoff for PD-L1-positive and -negative.
*P* value was a two-sided Fisher's exact test.

52 non-sensitive patients (TRG-II, N = 17; TRG-III, N = 7; TRG-IV, N = 28).

Archival formalin-fixed paraffin-embedded (FFPE) tissues with at least 80% tumor purity before immunotherapy were collected from 190 patients with ESCC receiving the first-line camrelizumab-based anti-PD1 immunotherapy and then subsequently analyzed by the mass spectrometry (MS)-based label-free quantification strategy (Ge et al, 2018; Jiang et al, 2019; Li et al, 2022) or IHC analysis (Fig. 1A).

## PD-L1 expression has a poor association with immunotherapy response

PD-L1 expression, assessed by immunohistochemistry (IHC), is a diagnostic marker for guiding immunotherapy. We examined the association of PD-L1 expression (PD-L1-negative (PD-L1−) and PD-L1-positive (PD-L1+)) with the clinical characteristics included in our study and found no significant association of PD-L1 expression with gender, clinical stage (cStage), smoking, and drinking. More importantly, there was no significant association of PD-L1 expression with immunotherapy response (Fisher's exact test, $P = 0.28$) (Fig. 1B,C). We also analyzed the association of PD-L1 expression (based on IHC) with immunotherapy response in the study by Liu et al (Liu et al, 2023), in which ESCC patients received the same regimen (camrelizumab-based anti-PD1 immunotherapy) as in our study. In the study by Liu et al, 77 ESCC patients (7 patients for scRNA-seq analyses, 40 patients for bulk RNA-seq validation, and 30 patients for multiplex IHC validation) with intact clinical characteristics were enrolled, including PD-L1 expression and immunotherapy response. We analyzed the association between the immunotherapy response and PD-L1 expression in the study of Liu et al and found there was also no significant difference in the distribution of sensitive (S) and non-sensitive (NS) patients in the PD-L1-negative and positive groups (Fisher's exact test, $P = 1$) (Fig. 1C). This addressed an important point that the PD-L1 expression did not sufficiently indicate the efficacy of immunotherapy. It suggested that not all PD-L1-positive patients were sensitive to immunotherapy, and some PD-L1-negative patients might benefit from immunotherapy (Fig. EV1A), which was consistent with the clinical trial results (Luo et al, 2021; Sun et al, 2021). To further validate the poor relationship between PD-L1 expression and immunotherapy response, we analyzed PD-L1 mRNA expression and immunotherapy response in other independent immunotherapy cohorts, including the GSE78220 melanoma cohort (Hugo et al, 2016) and IMvigor210 metastatic urothelial carcinoma cohort (Mariathasan et al, 2018), and found there was no significant difference in PD-L1 (CD274) expression between sensitive group and non-sensitive group (Fig. EV1B,C). In addition, PD-L1 expression was also not related to overall survival (Fig. EV1B,C). We then evaluated the predictive power of PD-L1 expression on ESCC immunotherapy response by calculating the area under the receiver operating characteristic (ROC) curve (AUC) value in our study. The result revealed that the AUC value for predicting immunotherapy response based on PD-L1 expression was merely 0.594 (Fig. 1D), suggesting that PD-L1 expression could not serve as a predictor of immunotherapy efficacy in ESCC patients. We further reviewed the published literature on ESCC immunotherapy and found that the AUC values were 0.595, 0.547, 0.472, and 0.604 in the other four studies of ESCC immunotherapy (including Liu et al, 2023, ChiCTR1900026240, ChiCTR2000037488, and NCT04225364) (Liu et al, 2022a; Liu et al, 2022b; Liu et al, 2023; Yan et al, 2022) (Fig. 1E), respectively. These results further verified that PD-L1 expression alone was insufficient in determining which patients could benefit from anti-PD1 immunotherapy, highlighting the urgent need to identify reliable predictive biomarkers to indicate immunotherapy.

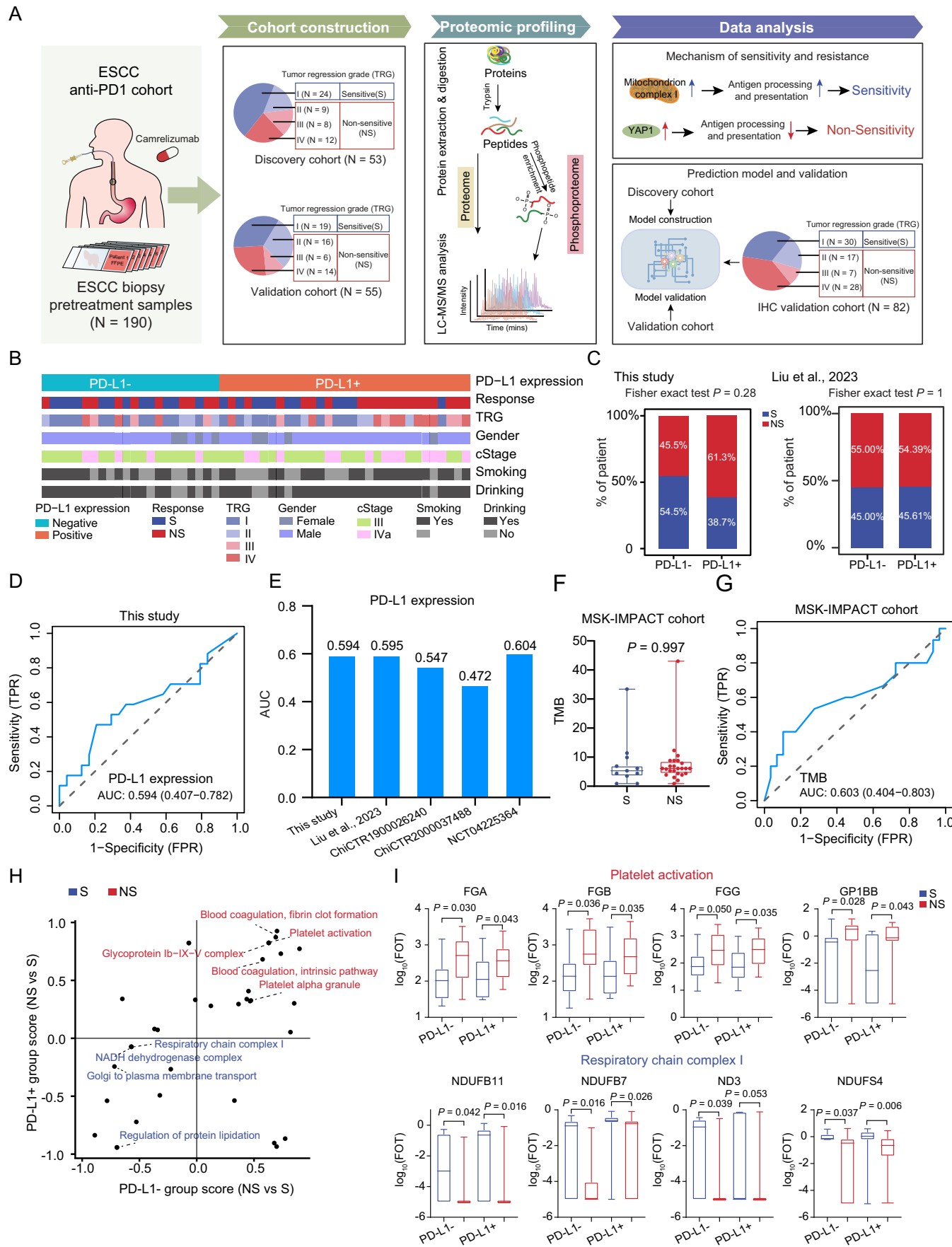

Figure 1. eusoft-did-translate-comment-enA summary of proteomic and phosphoproteomic analysis of esophageal squamous cell cancer immunotherapy cohort.

(A) The proteomics workflow involved three modules: cohort construction (including discovery cohort and validation cohort), proteomic profiling, and data analysis. The proteins were trypsin-digested, and then analyzed in a single-run (150 min) high-performance liquid chromatography mass spectrometry (HPLC-MS) using a Q Exactive HF-X Hybrid Quadrupole-Orbitrap Mass Spectrometer. MS proteomics data were quantified with the Firmiana proteomics workstation. (B) The clinical characteristics (including immunotherapy response, TRG, gender, cStage, smoking, and drinking) between PD-L1-positive (+) and negative (−) patients. (C) Proportion of S and NS in PD-L1+ and PD-L1− patients (two-sided Fisher's exact test) in our study and in the study of Liu et al, 2023. (D) Area under receiver operating characteristic (ROC) curves (AUC) for PD-L1 expression on prediction of ESCC immunotherapy response in this study. (E) AUC values of PD-L1 expression for the prediction of immunotherapy response in other studies. (F) Boxplots showing the differences of TMB level between S (n = 15) and NS (n = 29) patients of ESCC in the MSK-IMPACT cohort (two-sided Student's t test). Boxplots represent the interquartile range (IQR), with the box spanning the 25th to 75th percentiles and the median indicated by a horizontal line. Whiskers mark minimum or maximum values. (G) Area under the receiver operating characteristic (ROC) curve for TMB on the prediction of ESCC immunotherapy response in the MSK-IMPACT cohort. (H) Two-dimensional annotation enrichment analysis shows similar enrichments in the PD-L1+ and PD-L1− patients. In both PD-L1+ and PD-L1− patients, the S group is enriched with mitochondrial functional pathways, whereas the NS group is enriched with platelet activation-related pathways. (I) Boxplots show the protein expression of mitochondrial complex I and platelet activation pathway in S and NS groups of PD-L1+ and PD-L1− patients (two-sided Wilcoxon rank-sum test). Boxplots are defined as in (F). PD-L1− group: n = 10 (S) and 12 (NS), PD-L1+ group: n = 12 (S) and 19 (NS). Source data are available online for this figure.

Besides PD-L1 expression, tumor mutational burden (TMB) is another potential marker for immunotherapy. To evaluate the association between TMB and immunotherapy response in ESCC, we analyzed data from 44 ESCC patients with immunotherapy in the MSK-IMPACT cohort (Chowell et al, 2022), including 15 responders (S: sensitive to immunotherapy) and 29 non-responders (NS: non-sensitive to immunotherapy). As expected, survival analysis showed that S patients had significantly longer OS and PFS than NS patients (two-sided log-rank test, $P < 0.05$) (Fig. EV1D). However, when stratifying patients by TMB levels, no significant difference in OS or PFS was observed between patients with high and low TMB in the MSK-IMPACT cohort (two-sided log-rank test, $P > 0.05$) (Fig. EV1E). A similar lack of association was observed in the TCGA-ESCC cohort (89 patients) (two-sided log-rank test, $P > 0.05$) (Fig. EV1F). These results demonstrated that TMB might not be associated with immunotherapy response in ESCC patients. Furthermore, comparison of TMB levels between responders and non-responders in the MSK-IMPACT cohort revealed no significant difference (two-sided Student's t test, $P = 0.997$) (Fig. 1F), indicating that TMB might not be a reliable marker for ESCC immunotherapy. The AUC for predicting immunotherapy response based on TMB was only 0.603 in the MSK-IMPACT cohort (Fig. 1G). Overall, these results highlight the need for more reliable biomarkers to predict immunotherapy response in ESCC.

Then, we performed proteomic and phosphoproteomic analysis using liquid chromatography-mass spectrometry (LC-MS/MS). For the quality control of mass spectrometry performance, the HEK293T cell lysate was measured every 2 days, which was adopted in proteomic studies (Ge et al, 2018; Jiang et al, 2019; Xu et al, 2020). A Pearson's correlation coefficient was calculated for all the quality-control runs, and the results were shown in Fig. EV1G. The average correlation coefficient among the control samples was 0.956, demonstrating the consistent stability of the MS platform. Besides the HEK293T cell lysate, to directly assess the stability of the MS platform, we mixed the patient samples as a sample pool to evaluate the quality control of the MS platform during the sample measurement. The same strategy was adopted for the sample pool, resulting in an average Pearson's correlation coefficient of 0.993 (Fig. EV1G), indicating the reliability of the proteome data generated in this study. Proteomics measurement resulted in 5926–7397 proteins in each sample (Fig. EV1H). A total of 11,017 proteins were identified in all patient samples (N = 53). Furthermore, a total of 40,529 phosphosites corresponding to 6055 phosphoproteins were identified (Fig. EV1H,I; Dataset EV2). No

major difference in the proteome or phosphoproteome coverage between S and NS groups was observed (two-sided Student's t test, $P > 0.05$) (Fig. EV1H). In addition to protein identification, we further performed principal components analysis (PCA) to evaluate the expression variance, which revealed no batch effects at the quantification of proteome or phosphoproteome between S and NS samples (Fig. EV1J). To further evaluate the batch effects on proteomic and phosphoproteomic data, we also performed the principal component regression analysis (Bai et al, 2025; Lyu et al, 2024) ("Methods"). The results indicated that there were no observable batch effects in the proteomic and phosphoproteomic data (Fig. EV1K).

Given the weak association of PD-L1 expression with immunotherapy response, we wondered about the featured pathways related to non-sensitivity and sensitivity in PD-L1+ and PD-L1− groups, respectively. We examined the proteome difference between S and NS groups in patients with PD-L1-positive (N = 31: N (S) = 12, N (NS) = 19) and PD-L1-negative (N = 22: N (S) = 10, N (NS) = 12), and performed two-dimensional (2D) annotation enrichment analysis (Cox and Mann, 2012). It showed high resemblance of the enriched pathways between the PD-L1+ and PD-L1- groups. In both, the S group was significantly enriched for the mitochondrial functional pathways, including respiratory chain complex I (mitochondrial complex I), NADH dehydrogenase complex, Golgi to plasma membrane transport, and regulation of protein lipidation, while the NS group was significantly associated with platelet activation and platelet alpha granule pathways (Fig. 1H). The same tendency was observed in proteins involved in these pathways, for example, FGA, FGB, FGG, and GP1BB involved in platelet activation were upregulated in the NS group, while NDUFB11, NDUFB7, ND3, and NDUFS4 involved in mitochondrial complex I were upregulated in the S group (Fig. 1I). Overall, we found the functional similarity that mitochondrial complex I protein expression was associated with immunotherapy response in both the PD-L1+ and PD-L1− groups.

## Proteomic subtyping of the ESCC immunotherapy cohort reveals the association of mitochondrial complex I with immunotherapy response

To further explore the potential non-sensitive/sensitive mechanisms for immunotherapy response, we applied consensus clustering analysis, an unsupervised clustering method, to preliminarily determine the association between proteome pattern and

immunotherapy response. We employed consensus clustering to classify the patients into different clusters. Through the comprehensive evaluation of the consensus CDF (consensus cumulative distribution function), the delta area under the CDF curve, the maximal average silhouette width of each cluster, together with the association of consensus clusters with immunotherapy response, we determined 3 clusters as the optimal clusters for the 53 ESCC patients based on the proteomic data, which were designated G-I ($N = 16$), G-II ($N = 14$) and G-III ($N = 23$) subtypes (Figs. 2A and EV2A–F) ("Methods"). To evaluate the robustness of these subtypes, we also employed non-negative matrix factorization (NMF) (Gaujoux and Seoighe, 2010), an alternative unsupervised clustering algorithm, indicating that the three clusters (NMF-I, NMF-II, and NMF-III), as the optimal classification for proteomic clustering, were associated with immunotherapy response, with high consistency between consensus clustering and NMF clustering (Fig. EV2G–L) ("Methods"). The proteomic subtypes displayed different clinical outcomes, especially immunotherapy response, which was significantly associated with proteomic subtypes (Fisher's exact test, $P = 0.028$) (Fig. 2B,C). The immunotherapy response exhibited a gradual non-sensitive phenomenon from G-I to G-III, as the percentage of sensitive patients dramatically decreased from 69% in G-I to 26% in G-III. Conversely, the percentage of non-sensitive patients increased from 31% in G-I to 74% in G-III (Fig. 2C). In addition, except for immunotherapy response, there was no significant difference among the clinicopathological characteristics (including gender, cStage, smoking, drinking, and PD-L1 expression) of patients in the three proteomic subtypes (Fig. 2B).

To further characterize the three proteomic subtypes, the comparative analysis of proteomic profiling resulted in 1485 (G-I), 1699 (G-II), and 80 (G-III) differential expression proteins (DEPs) ($P < 0.05$; fold change > 2), showing distinct molecular features among the three proteomic subtypes (Dataset EV3). We then performed a functional enrichment analysis based on the DEPs and determined the dominant bioprocesses of each subtype according to the ConsensusPathDB (CPDB) database (Herwig et al, 2016). The G-I subtype had higher immune and mitochondrial features, such as immune system, CD8 T cell receptor signaling, antigen processing and presentation, respiratory transport, citrate cycle (TCA cycle), and metabolism of lipids (Fig. 2D). In the G-II subtype, the significantly upregulated proteins were enriched in pathways, such as metabolism of RNA, mRNA splicing, mRNA processing, and chromatin organization (Fig. 2D). The G-III subtype was more related to platelet function, including platelet aggregation, platelet activation, platelet adhesion to exposed collagen, and the complement system (Fig. 2D; Dataset EV3). We then evaluated the tumor microenvironment among three subtypes by xCell analysis. Consistent with pathway enrichment, we found the highest immune score in the G-I subtype, compared with other subtypes (ANOVA test, $P = 0.043$) (Fig. EV2M). The immune cell types, such as CD8 + T cells, CD8+ Tcm, CD8+ Tem, and macrophages, were also enriched in the G-I subtype, while platelets and their precursor megakaryocytes were significantly aggregated in the G-III subtype (Fig. EV2N; Dataset EV3). It has been reported that activated platelets can secrete some immune regulatory factors, such as TGFB1, affecting CD8 + T cell functions (Li et al, 2024; Rachidi et al, 2017). We performed functional experiments to assess the impact of platelet activation on CD8 + T cell function

(Fig. EV2O). Platelets isolated from ESCC patient blood samples were activated with thrombin and collected for MS-based proteome profiling to explore the molecular perturbation following platelet activation. Proteomic data analysis revealed upregulation of platelet activation markers (such as SELP and CD40LG), indicating their activation status. In addition, we found increased levels of TGFB1 after platelet activation, which has been reported to impair CD8 + T cell cytotoxicity and immune regulatory factors (Fig. EV2P). The addition of activated platelet supernatant to co-cultures of CD8 + T cells and ESCC cells resulted in a significant decrease in CD8 + T cell-mediated killing of ESCC cells (two-sided Student's $t$ test, $P < 0.05$) (Fig. EV2Q), suggesting that activated platelets might secrete immune regulatory factors to suppress CD8 + T cell effector function.

The tumor immune microenvironment analysis also showed that the tumor biopsy comprised not only a population of cancer cells but also non-cancer cells (Fig. EV2N), indicating that the pathway associated with ESCC immunotherapy might be shared between cancer cells and non-cancer cells. Single-cell RNA sequencing (scRNA-seq) represents a powerful approach to dissecting cellular heterogeneity and assessing the differences in gene expression between different cell populations at single-cell resolution. To explore the cell population distribution of the overrepresented proteins and their involved pathways observed in our study at the single-cell level, we incorporated the scRNA-seq dataset of ESCC published by Pan et al (Pan et al, 2022). We analyzed the scRNA-seq dataset from Pan et al and observed that the dominant proportion of cell type was malignant squamous epithelial cell (Fig. EV3A,B) ("Methods"). We also found that these overrepresented pathways associated with immunotherapy response observed in our study, including TCA cycle (IDH3A and SUCLG1) and lipid metabolism (FABP5 and PTPMT1), were highly enriched in malignant squamous epithelial cells in comparison to other cell types in the tumor biopsies by using the scRNA-seq dataset of Pan et al (Fig. EV3C,D), indicating these pathways potentially functioned in ESCC cells and regulated the immunotherapy response of ESCC. In conclusion, this integrated multi-omics analysis highlighted the distinct biological features of ESCC subtypes and demonstrated immunotherapy response-related pathways, particularly the TCA cycle and lipid metabolism, were notably enriched in malignant squamous epithelial cells, underscoring their potential role in influencing tumor-immune interactions critical for therapeutic response.

In accordance with the CPDB pathway enrichment, we performed gene set enrichment analysis (GSEA) analysis (Subramanian et al, 2005), which showed that the mitochondrion organization pathway was significantly enriched in the G-I subtype, compared with other subtypes (Fig. 2E). Mitochondria are vital for ATP synthesis via oxidative phosphorylation of mitochondrial complexes I-V and also deeply integrated into cellular metabolism and signaling pathways. We performed GSEA analysis based on the mitochondria-specific function gene set, human MitoCarta (Jayavelu et al, 2022), and found mitochondrial complex I was significantly enriched in the G-I subtype, but not other mitochondrial complexes II–V (Fig. 2F). The proteins of mitochondrial complex I, such as NDUFA10, NDUFB7, NDUFB10, and NDUFS1, were significantly overrepresented in the G-I subtype (Fig. 2G). To validate this finding, we performed single-sample gene set enrichment analysis (ssGSEA) (Barbie et al, 2009). The ssGSEA

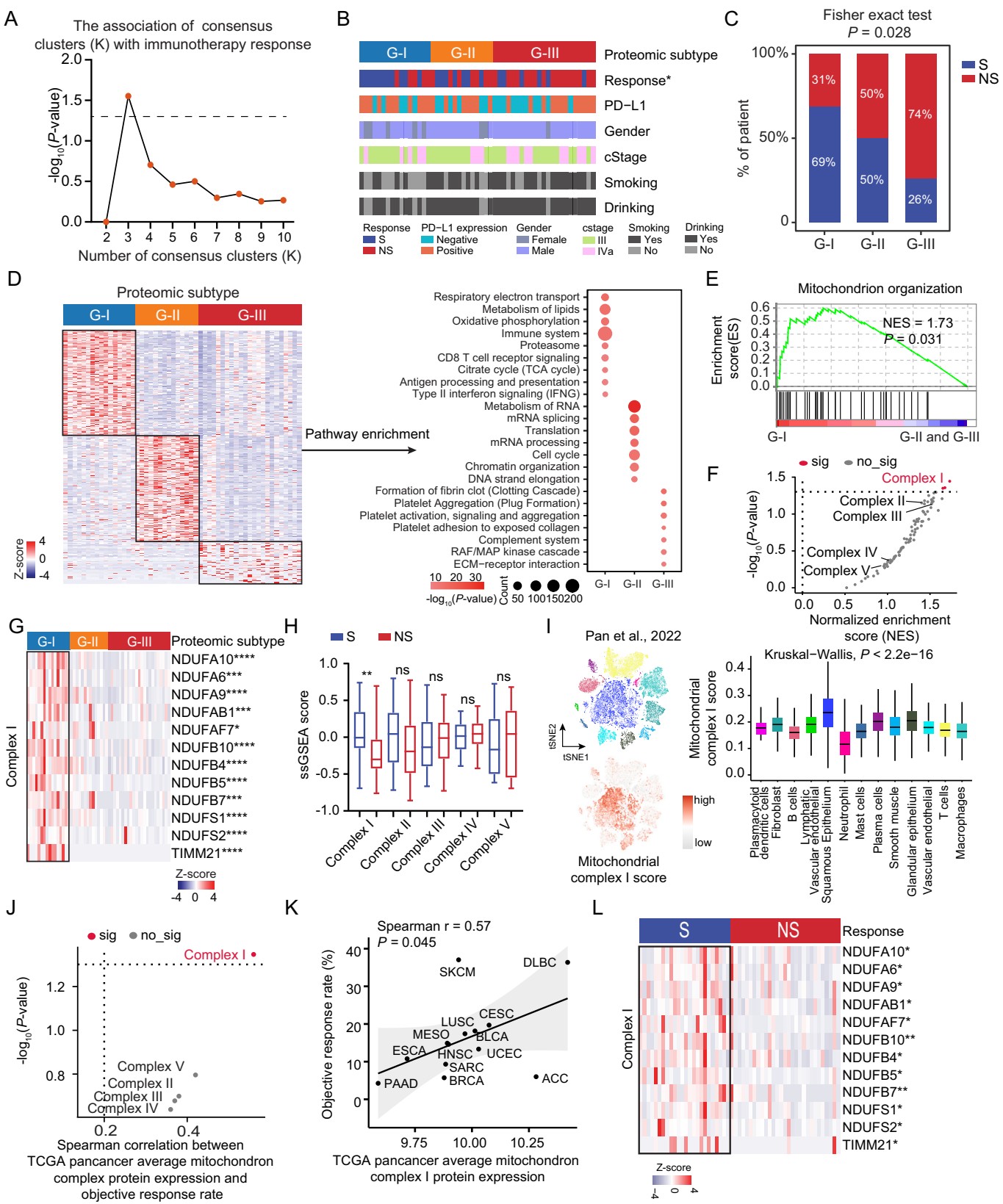

◄ **Figure 2. Proteomic subtyping of ESCC immunotherapy cohort and association with immunotherapy response.**

(A) The association of consensus clusters ($K = 2$–$10$) with ESCC immunotherapy response (*P* value from two-sided Fisher's exact test). (B) The heatmap shows the association of proteomic subtypes with clinicopathologic characteristics (including immunotherapy response, PD-L1 expression, gender, cStage, smoking, and drinking) (two-sided Fisher's exact test). (C) Barplot shows the distribution of immunotherapy response (S/NS) among three proteomic subtypes. *P* value was calculated by two-sided Fisher's exact test. (D) The significantly differential expression proteins and their involved pathways. The heatmap depicts the relative abundance of the signature proteins across three subtypes (Z score of FOT) (left). Bubble plot showing the pathway enrichment of each proteomic subtype (right). *P* value was calculated according to the hypergeometric test. (E) GSEA enrichment plots of mitochondrion organization in the G-I subtype. *P* value from phenotype-based permutation test. (F) Volcano plot depicting the gene set enrichment for the MtioPathways3.0. Mitochondrial complex gene sets (I–V) are highlighted. *P* value from Phenotype-based permutation test. (G) Heatmap illustrating the significantly expressed proteins of mitochondrial complex I in the G-I subtype. *P* value from two-sided Wilcoxon rank-sum test. Annotated *P* values are defined as follows: \**P* < 0.05, \*\**P* < 0.01, \*\*\**P* < 0.001, \*\*\*\**P* < 0.0001. (H) The ssGSEA score of mitochondrial complexes I–V between S ($n = 24$) and NS ($n = 29$). *P* value from two-sided Wilcoxon rank-sum test. *P* = 0.007 (Complex I), 0.173 (Complex II), 0.451 (Complex III), 0.894 (Complex IV), 0.894 (Complex V). \*\**P* < 0.01, ns not significant. Boxplots represent the interquartile range (IQR), with the box spanning the 25th to 75th percentiles and the median indicated by a horizontal line. Whiskers mark minimum or maximum values. (I) The t-SNE maps of individual cell AUC score overlay for the mitochondrial complex I score using the scRNA-seq data from Pan et al (Pan et al, 2022) (left), and the boxplots show the AUC score of mitochondrial complex I in each cell cluster (right) (two-sided Kruskal–Wallis test). *P* = 0. Boxplots represent the interquartile range (IQR), with the box spanning the 25th to 75th percentiles and the median indicated by a horizontal line. Whiskers extend to the most extreme data points within 1.5×IQR. $n = 62$ (plasmacytoid dendritic cells), 3101 (fibroblast), 2901 (B cells), 337 (lymphatic vascular endothelial), 11,658 (squamous epithelium), 360 (neutrophil), 452 (mast cells), 844 (plasma cells), 1140 (smooth muscle), 1564 (glandular epithelium), 1963 (vascular endothelial), 7184 (T cells), and 5671 (macrophages). (J) Volcano plot showing the Spearman correlation between average mitochondrial complex protein expression of TCGA pan-cancer and objective response rate of the immunotherapy. *P* value was calculated by two-sided Spearman's correlation test. (K) Correlation between mitochondrial complex I protein expression and objective response rate of the immunotherapy in the TCGA pan-cancer cohort. *P* value was calculated by two-sided Spearman's correlation test. (L) Heatmap showing mitochondrial complex I proteins in S and NS groups (two-sided Wilcoxon rank-sum test). Annotated *P* values are defined as follows: \**P* < 0.05, \*\**P* < 0.01, \*\*\**P* < 0.001, \*\*\*\**P* < 0.0001. Source data are available online for this figure.

results showed that the ssGSEA score of mitochondrial complex I was significantly higher in the sensitive group compared with the non-sensitive group (Fig. 2H), which was also predominantly enriched in the malignant squamous epithelial cells compared to other cell types in the ESCC tumor microenvironment at single-cell resolution (Pan et al, 2022) (Kruskal–Wallis test, *P* < 0.05) (Figs. 2I and EV3E). GSEA analysis also revealed that mitochondrial complex I, as well as the CD8 + T cell signaling pathway, was significantly enriched in S patients compared to NS patients across G-I, G-II, and G-III subtypes (Fig. EV3F). These results demonstrated that, regardless of subtype, S patients consistently exhibit enrichment of key pathways, notably mitochondrial complex I and CD8 + T cell signaling, compared to NS patients within each subtype. In other words, although the proportion of S and NS patients varied among subtypes, the characteristic enrichment of these pathways in S patients was robust within each subtype. This finding suggested that the association between these pathways and immunotherapy sensitivity was independent of subtype classification and reflected intrinsic molecular differences linked to patient response. Extending our studies to additional cancer types, we investigated whether there was a relationship between mitochondrial complex I gene expression and overall objective response to immunotherapy. We correlated mitochondrial complexes I–IV-related gene expression for each cancer type with the objective response rate (ORR) presented by Yarchoan et al (Yarchoan et al, 2017). We indeed observed a significantly positive relationship between immunotherapy ORR and mitochondrial complex I gene expression across different cancer types derived from the TCGA pan-cancer mRNA expression datasets (Spearman $r = 0.57$, *P* = 0.045) (Fig. 2J,K), instead of other mitochondrial complexes II–V, suggesting that tumors with higher mitochondrial complex I protein expression tend to be more sensitive to immunotherapy. We also observed the elevated expression of complex I proteins, including NDUFA6/9/10, NDUFAB1, NDUFAF7, NDUFB4/5/7/10, and NDUFS1/2, in the sensitive group (Fig. 2L). Overall, these results demonstrate that patients with high expression of mitochondrial complex I proteins can be conducive to immunotherapy.

To further examine whether the proteomic subtyping of our cohort was feasible in another ESCC cohort (the research of Liu et al) (Liu et al, 2021a), we conducted subtype determination of the ESCC cohort from Liu et al based on the overrepresented proteins in the proteomic subtypes of our cohort using the nearest template prediction (NTP) algorithm implanted in CMScaller R package (v2.0.1) (Tang et al, 2024; Yang et al, 2024) ("Methods"). The proteomic subtyping was applied to the proteomic data of 124 ESCC patients from Liu et al, which also clustered the 124 ESCC patients into three subtypes (G-I: $N = 49$, G-II: $N = 42$, G-III: $N = 33$) and showed significant associations with overall survival (OS) and disease-free survival (DFS) (two-sided log-rank test, *P* < 0.05) (Fig. EV4A,B). Among the three proteomic subtypes, patients from the G-I subtype had the best OS and DFS outcomes compared to those from G-II and G-III subtypes, which might be beneficial from anti-PD1 immunotherapy. Conversely, the G-III subtype had the worst OS and DFS among the three proteomic subtypes, which might not be suitable for receiving anti-PD1 immunotherapy. We compared our proteomic subtypes (G-I, G-II, and G-III) with those identified by Liu et al (S1 and S2), and a significant association was observed between the two subtype classifications (Fisher's exact test, *P* = 6.60E-4) (Fig. EV4C). Patients with the S1 subtype accounted for 65.3% of our G-I subtype. There was no significant difference in the distribution of S1 and S2 subtypes in our G-II subtype, while patients with the S2 subtype accounted for 73.8% of our G-III subtype. Furthermore, we also found that the G-I subtype exhibited the highest mitochondrial complex I ssGSEA score among the three proteomic subtypes identified by us in the study of Liu et al (ANOVA test, *P* = 2.3E-5) (Fig. EV4D). Consistently, the S1 subtype demonstrated a significantly higher score compared to the S2 subtype from Liu et al (two-sided Student's *t* test, *P* = 0.015) (Fig. EV4E), further confirming the association between these two kinds of proteomic subtypes. In the study of Liu et al, we also found that patients with high mitochondrial complex I ssGSEA score showed significantly improved OS and DFS compared to those with low score (two-sided log-rank test, *P* < 0.05) (Fig. EV4F). Notably, similar trends were observed within both the S1 and S2 subtypes, where high

mitochondrial complex I ssGSEA scores were associated with better OS and DFS (two-sided log-rank test, $P < 0.05$) (Fig. EV4G,H). In addition, we applied our proteomic subtypes to the proteomic data of Li et al (Li et al, 2021). We categorized the 94 ESCC patients of Li et al into three subtypes (G-I: 35, G-II: 28, and G-III: 31) (Fig. EV4I). In line with results from both our cohort and that of Liu et al, the analysis of mitochondrial complex I ssGSEA scores revealed that the G-I subtype consistently exhibited the highest score among the three subtypes (ANOVA test, $P = 0.040$) (Fig. EV4I). Collectively, these findings demonstrate the robustness and consistency of our proteomic subtypes across independent datasets and underscore the potential relevance of mitochondrial complex I in ESCC immunotherapy.

## Higher mitochondrial complex I levels are associated with enhanced ESCC immunotherapy sensitivity and elevated antigen processing and presentation

To explore whether mitochondrial complex I levels are associated with patient response to ESCC immunotherapy, we split patients into high and low mitochondrial complex I groups based on the mitochondrial complex I ssGSEA score. This stratification was significantly associated with immunotherapy response (Fisher's exact test, $P = 0.013$) (Fig. 3A). The immunotherapy-sensitive rate was 63% in the high complex I group, which was higher than 27% in the low complex I group (Fig. 3A). Pathway enrichment analysis revealed that, besides complex I biogenesis, antigen processing and presentation and autophagy pathway were also significantly enriched in the high complex I group (Fig. 3B; Dataset EV4). In particular, the MHC-I molecules (such as HLA-A, HLA-B, HLA-C, and HLA-F), the major determinant of CD8 + T cell activation and killing capacity of tumor cells, were increased in the high complex I group (Figs. 3C and EV4J). Consistently, the higher activation of antigen processing and presentation was observed in the high complex I group in comparison with the low complex I group (two-sided Wilcoxon rank-sum test, $P = 0.004$) (Fig. 3D), as well as in the G-I subtype compared with the other subtypes (ANOVA test, $P = 0.004$) (Fig. EV4K). Meanwhile, besides MHC-I molecules, the proteins (such as CALR, PSME2, TAP2, and CANX) involved in antigen processing and presentation were highly expressed in the high complex I group and the S group (Fig. 3E), which facilitated the presentation of antigens on the tumor cell surface for recognition by T cells and killed cancer cells.

As noted, hot tumors are characterized by abundant infiltration of CD8 + T cells, while cold tumors exhibit low or excluded CD8 + T cell infiltration (Chen and Mellman, 2017; Zhang et al, 2022). We performed comparative analysis and found no significant difference in CD8A expression, a marker for CD8 + T cells, between the PD-L1-positive and PD-L1-negative patients (two-sided Wilcoxon rank-sum test, $P = 0.811$) (Fig. 3F). However, CD8A expression was significantly higher in S patients compared to NS patients (two-sided Wilcoxon rank-sum test, $P = 0.015$), with 68.4% of patients in the high CD8A group showing sensitivity to immunotherapy (Fig. 3F,G). Further analysis showed higher CD8A expression in the high mitochondrial complex I group compared to the low group. Importantly, integrating both mitochondrial complex I and CD8A expression revealed that patients with both high levels had the highest proportion of S patients (84.6%), exceeding each of the high mitochondrial

complex I (63%) or high CD8A (68.4%) groups alone (Fig. 3H,I). In summary, these results indicate that the combination of high CD8 + T cell infiltration (hot tumor) and high mitochondrial complex I expression (elevated MHC-I on tumor cells) is associated with greater immunotherapy sensitivity.

## Phosphoproteomic subtyping of ESCC immunotherapy cohort reveals the association of hippo pathway with ESCC immunotherapy

Phosphorylation is a vital process within cells and initiates a cascade of downstream signaling events crucial for cellular function. Dysregulation of this process has been linked to cancer (Arshad et al, 2019). The phosphoproteome may also capture some signaling aberrations that are not displayed at the proteome level (Xu et al, 2020). Integrative proteomic and phosphoproteomic profiling could comprehensively characterize the regulatory network in tumor initiation, progression, and treatment (Drake et al, 2016; Liu et al, 2021b). To further explore the mechanism associated with immunotherapy response, we also employed consensus clustering for the phosphoproteome and identified four phosphoproteomic subtypes that had maximal average silhouette width, which were designated G-I ($N = 15$), G-II ($N = 13$), G-III ($N = 14$), and G-IV ($N = 11$) subtypes (Figs. 4A and EV5A–E; Dataset EV5). Further statistical analysis of phosphoproteomic subtypes and immunotherapy response showed that phosphoproteomic subtypes, like proteomic subtypes, were also significantly associated with ESCC immunotherapy response (Fisher's exact test, $P = 0.02$) (Fig. 4A), which suggested the potential consistent association of phosphoprotein or protein patterns with immunotherapy response. The G-I subtype had the highest proportion of sensitive patients, while the G-IV subtype had the highest proportion of non-sensitive patients. The immunotherapy response exhibited a gradual non-sensitive phenomenon for immunotherapy from G-I to G-IV, as the percentage of sensitive patients dramatically decreased from 73.3% in G-I to 18.2% in G-IV. Conversely, the percentage of non-sensitive patients increased from 26.7% in G-I to 81.8% in G-IV (Fig. 4A).

Comparison of the phosphoproteins among four phosphoproteomic subtypes, resulting in 687 (G-I), 1297 (G-II), 63 (G-III), and 80 (G-IV) differential phosphoproteins ($P < 0.05$; fold change > 1.5), showing distinct molecular features among the four phosphoproteomic subtypes (Dataset EV5). Next, we performed a pathway-based characterization of all phosphoproteomic subtypes to investigate specific biological features of the individual subtype (Fig. 4B). The G-I subtype was characterized by the upregulation of the immune system, antigen processing and presentation, lipid metabolism, and the hippo pathway. The G-II subtype exhibited enrichment in chromatin organization, mRNA splicing, and mRNA processing. Membrane traffic and cell cycle pathways were enriched in the G-III subtype. In the G-IV subtype, we observed the dominant enrichment of platelet-related pathways, including platelet activation and platelet aggregation. Kinase-substrate analysis revealed the elevated expression of the kinases and their downstream substrates among the four phosphoproteomic subtypes (Fig. 4C). In the G-I subtype, we found an elevated expression of SRC, PRKCA, and the core kinases of the hippo pathway (such as LATS1, LATS2, STK3, and STK4). The G-II subtype was characterized by AKT1, and the G-III subtype was predominantly

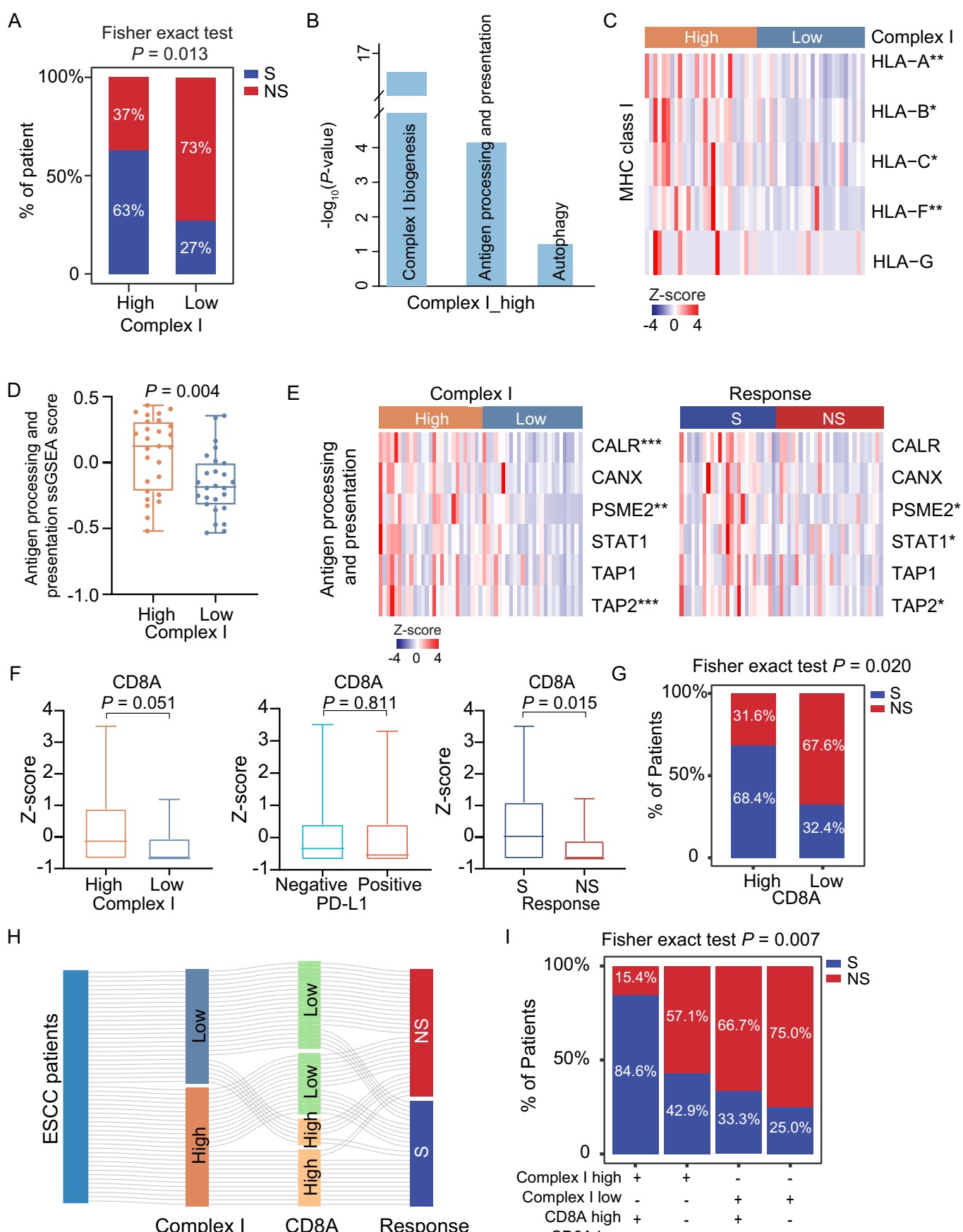

Figure 3.   Higher mitochondrial complex I levels are associated with enhanced ESCC immunotherapy sensitivity and elevated antigen processing and presentation.

(A) Bar chart displaying the distribution of S and NS proportions between high and low mitochondrial complex I groups. P value from two-sided Fisher's exact test. (B) Pathway alterations in the high mitochondrial complex I group (hypergeometric test). (C) Differential expression of MHC class I proteins between high and low mitochondrial complex I groups (two-sided Wilcoxon rank-sum test). Annotated P values are defined as follows: *P < 0.05, **P < 0.01, ***P < 0.001, ****P < 0.0001. (D) Boxplot showing the difference of antigen processing and presentation ssGSEA score between high (n = 27) and low (n = 26) mitochondrial complex I groups. P value from two-sided Wilcoxon rank-sum test. Boxplots represent the interquartile range (IQR), with the box spanning the 25th to 75th percentiles and the median indicated by a horizontal line. Whiskers mark minimum or maximum values. (E) Heatmap illustrating the differential expression of proteins involved in antigen processing and presentation between high and low mitochondrial complex I groups (left panel), as well as S and NS groups (right panel) (two-sided Wilcoxon rank-sum test). Annotated P values are defined as follows: *P < 0.05, **P < 0.01, ***P < 0.001, ****P < 0.0001. (F) Boxplots showing the expression of CD8A in high (n = 27) and low (n = 26) mitochondrial complex I groups, PD-L1-positive (n = 31) and PD-L1-negative (n = 22) groups, and S (n = 24) and NS (n = 29) groups. P value from two-sided Wilcoxon rank-sum test. Boxplots are defined as in (D). (G) Barplot shows the distribution of immunotherapy response (S/NS) in high and low CD8A groups. P value was calculated by the two-sided Fisher's exact test. (H) Sankey plot showing the flow of the ESCC patients featured with different mitochondrial complex I, CD8A expression, and immunotherapy response groups. (I) The integration of mitochondrial complex I and CD8A expression in immunotherapy response. P value was calculated by two-sided Fisher's exact test. Source data are available online for this figure.

featured with CDK2 and CDK7. Further investigation into the differentially altered phosphosites showed that elevated substrates involved in the immune system (such as HLA-A T345, HLA-B S352, LCK Y394, and CD44 T407) and the hippo signaling pathway (such as LATS2 S380, LATS1 T246, and YAP1 S127) were dominantly enriched in the G-I subtype. The phosphorylation of proteins involved in mRNA processing and splicing (including HNRNPA1 S6, HNRNPA1 S338, YBX1 S314, YBX1 S2 and S3) were dominant in the G-II subtype, and CDK2 T160 phosphorylated by CDK7 involved in cell cycle showed high expression in the G-III subtype (Fig. 4C). Due to the sensitive patients (the S group) accounting for the highest proportion both in proteomic G-I subtype and phosphoproteomic G-I subtype, this reminded us there might be an overlap between proteomic subtypes and phosphoproteomic subtypes. Comparison between the phosphoproteomic subtypes and the proteomic subtypes revealed a high concordance. Statistical analysis revealed a significant association between the proteomic subtypes and phosphoproteomic subtypes (Fisher's exact test, P = 8.947E-10) (Fig. 4D). Among the four phosphoproteomic subtypes, 12 of 15 ESCC patients in the phosphoproteomic G-I subtype overlapped with the proteomic G-I subtype, 11 of 13 ESCC patients in the phosphoproteomic G-II subtype overlapped with the proteomic G-II subtype, 9 of 14 ESCC patients in the phosphoproteomic G-III subtype overlapped with the proteomic G-III subtype, and 9 out of 11 ESCC patients in the phosphoproteomic G-IV subtype overlapped with the proteomic G-III subtype, respectively, which also indicated a link between the proteome and phosphoproteome (Fig. EV5F). In addition, we also found a consistency of pathway enrichment between proteomic and phosphoproteomic subtypes (Fig. EV5G). Specifically, pathways related to the immune system, proteasome, antigen processing and presentation, and metabolism of lipids were significantly enriched in both the proteomic G-I subtype and the phosphoproteomic G-I subtype. The proteomic G-II subtype and phosphoproteomic G-II subtype showed significant enrichment in the mRNA splicing, mRNA processing, and chromatin organization pathways. Moreover, pathways associated with platelet aggregation and activation, as well as ECM-receptor interactions, were enriched in both the proteomic G-III subtype and phosphoproteomic G-IV subtype. Meanwhile, membrane trafficking and endocytosis were only found to be significantly enriched in the phosphoproteomic G-III subtype, which further stratified the ESCC patients into different subgroups and reflected different molecular features at the phosphoproteome

level. Overall, these results suggested that there was a link between proteomic subtypes and phosphoproteomic subtypes associated with immunotherapy response, indicating the potential association of signal transduction regulation driven by phosphorylation.

Consistently, we performed GSEA analysis and found that the G-I subtype of phosphoproteome also significantly enriched mitochondrial complex I biogenesis at the proteome level (Fig. 4E). In addition, the phosphoproteomic G-I subtype had the highest mitochondrial complex I ssGSEA score among four phosphoproteomic subtypes (ANOVA test, P = 0.0014) (Fig. 4F), consistent with the proteomic G-I subtype, indicating the underlying regulatory mechanisms governing the functional interplay between proteins and phosphoproteins. GSEA analysis of the phosphoproteins between S and NS groups revealed that the hippo pathway showed most significant enrichment in the S patients (Figs. 4G and EV5H), and it was primarily enriched in malignant squamous epithelial cells dissected by scRNA-seq analysis compared to other cell types in the microenvironment of ESCC (Pan et al, 2022) (Kruskal–Wallis test, P < 0.05) (Fig. EV5I,J). In addition, LATS2 showed an obviously positive correlation with its substrate YAP1 S127 (representing YAP1 inactivation (Zhao et al, 2007)) (Spearman r = 0.49, P = 0.0002) (Fig. EV5K), which could inactivate YAP1 activity. To further explore whether these observed differences in phosphorylation patterns are driven by alterations in the corresponding protein expression, we normalized phosphoproteomic data to corresponding protein abundance. (Fig. EV5L–P). We also found high concordant results with the non-normalized approach in terms of differential phosphoprotein expression, pathway enrichment, and kinase-substrate profiles across all subtypes. Therefore, the phosphoproteomic data generated from the standardized procedure might be less influenced by its protein abundance (Li et al, 2020; Tanzer et al, 2021; Xu et al, 2020).

As previously reported, YAP1 has been shown to have prominent functions in cancer initiation, aggressiveness, metastasis, and therapy resistance (Zanconato et al, 2016), but its role in ESCC immunotherapy remains unclear. We analyzed YAP1 protein expression in S and NS patients and found that YAP1 protein was significantly higher in NS patients compared to S patients (two-sided Wilcoxon rank-sum test, P = 0.014) (Fig. EV5Q). We then compared the performance of the YAP1 protein and phosphorylation levels in predicting ESCC immunotherapy response by calculating AUC values. The results showed that the YAP1 protein had a higher AUC value than YAP1 phosphorylation (0.70 vs. 0.60)

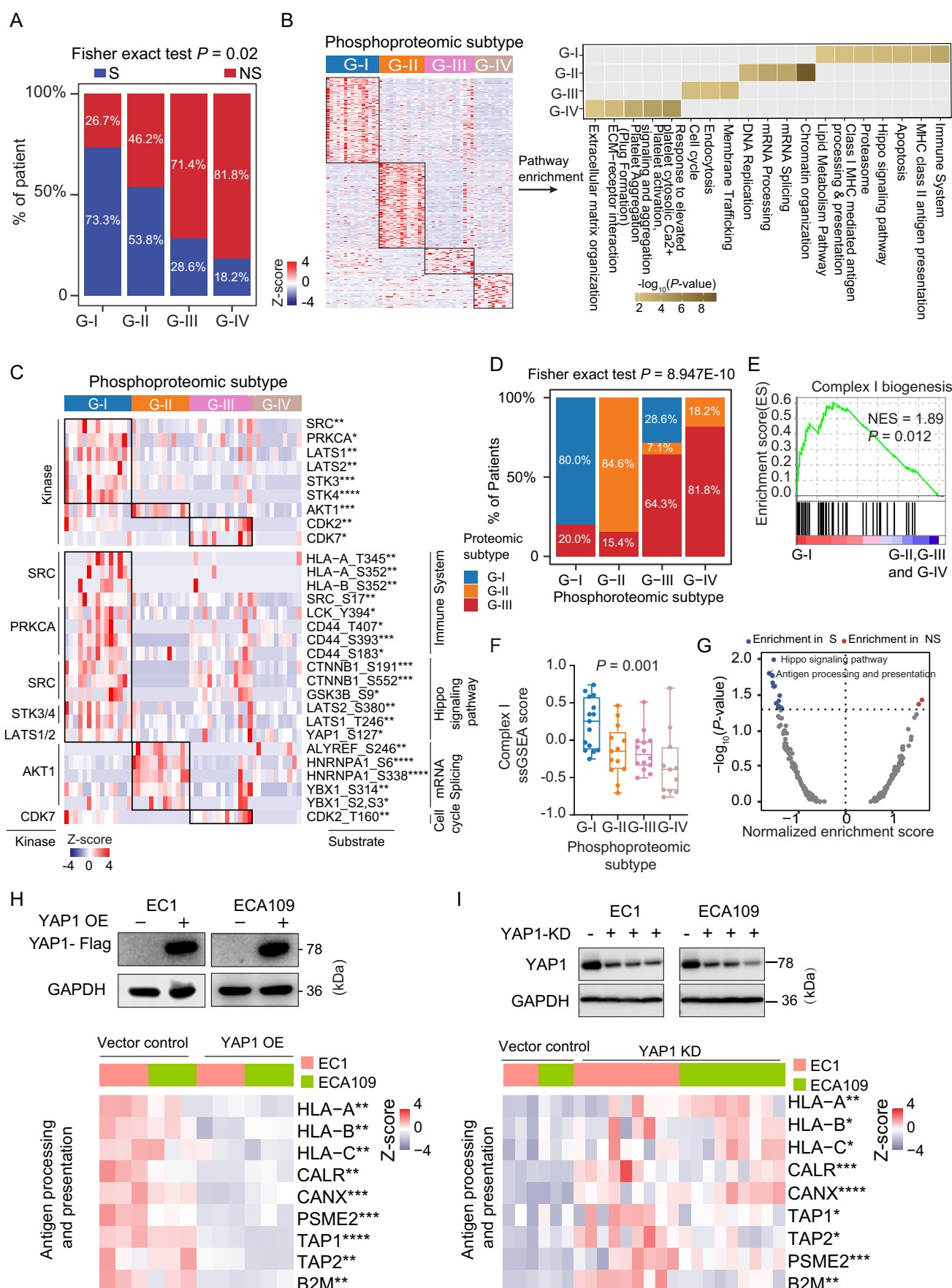

**Figure 4. Phosphoproteomic subtypes of the ESCC immunotherapy cohort.**

(A) Barplot shows the distribution of immunotherapy response (S/NS) among four phosphoproteomic subtypes. *P* value was calculated by two-sided Fisher's exact test. (B) The heatmap depicts the relative abundance of the signature phosphoproteins among four phosphoproteomic subtypes (left). Enriched pathways by these phosphoproteins in the phosphoproteomic subtypes (right) (hypergeometric test). (C) The heatmap showing the kinases and their substrates in each phosphoproteomic subtype (two-sided Wilcoxon rank-sum test). Annotated *P* values are defined as follows: *$P < 0.05$, **$P < 0.01$, ***$P < 0.001$, ****$P < 0.0001$. (D) Barplot indicating the comparison of phosphoproteomic subtypes and proteomic subtypes (two-sided Fisher's exact test). (E) GSEA enrichment plot of complex I biogenesis in the phosphoproteomic G-I subtype. *P* value from Phenotype-based permutation test. (F) Boxplot of mitochondrial complex I ssGSEA score among four phosphoproteomic subtypes. *P* value was from the ANOVA test. Boxplots represent the interquartile range (IQR), with the box spanning the 25th to 75th percentiles and the median indicated by a horizontal line. Whiskers mark minimum or maximum values. $n = 15$ (G-I), 13 (G-II), 14 (G-III), and 11 (G-IV). (G) Volcano plot showing the pathway enrichment between S and NS groups based on the phosphoproteome by GSEA analysis. *P* value from Phenotype-based permutation test. (H) Western blot showing the overexpression (OE) of YAP1 in ESCC cells (EC1 and ECA109). Heatmap showing the differential expression of proteins involved in antigen processing and presentation in vector control and YAP1 OE ESCC cells (two-sided Student's *t* test). Annotated *P* values are defined as follows: *$P < 0.05$, **$P < 0.01$, ***$P < 0.001$, ****$P < 0.0001$. (I) Western blot showing the knockdown (KD) of YAP1 in ESCC cells (EC1 and ECA109). The heatmap showing the differential expression of proteins involved in antigen processing and presentation in vector control and YAP1 OE ESCC cells (two-sided Student's *t* test). Annotated *P* values are defined as follows: *$P < 0.05$, **$P < 0.01$, ***$P < 0.001$, ****$P < 0.0001$. Source data are available online for this figure.

(Fig. EV5R), suggesting that the protein level of YAP1 was better than the phosphorylation level in predicting immunotherapy response. Collectively, the integrative analysis of phosphoproteomic data demonstrated that the core kinase/substrate cascade of the hippo signaling pathway resulted in the inhibition of downstream effector protein YAP1 activity, which could facilitate ESCC immunotherapy.

To verify the regulation of YAP1 with ESCC immunotherapy in vitro, we utilized a vector bearing a constitutively active mutant YAP1 (2SA) as the overexpression (OE) of YAP1 and an empty vector as a control to transfect two ESCC cell lines (EC1 and ECA109) (Fig. 4H). Proteome measurement also showed that YAP1 activation significantly reduced the expression of proteins involved in antigen processing and presentation (including HLA-A/B/C, CALR, and CANX) (Fig. 4H). Based on the above findings, we speculate that YAP1 may influence immunotherapy through regulating antigen processing and presentation. In addition, we constructed YAP1-knockdown (KD) ESCC cell lines and investigated whether YAP1-KD could reverse antigen processing and presentation downregulation in ESCC cells by proteomic measurement. The results demonstrated that YAP1-KD ESCC cells exhibited increased expression of antigen processing and presentation molecules (such as HLA-A, HLA-B, HLA-C, CALR, and CANX) (Fig. 4I). Overall, these results further supported that YAP1 activation might impair immunotherapy response through downregulating antigen processing and presentation.

## High mitochondrial complex I protein expression of ESCC cells and tumor organoids enhances tumor killing by activated CD8 + T cells

Based on the proteomic subtyping analysis, we found that mitochondrial complex I proteins such as NDUFB7, NDUFB10, and NDUFS1 were significantly upregulated in the proteomic G-I subtype (immunotherapy-sensitive subtype). Among these proteins, we found NDUFB7, the subunit of mitochondrial complex I, showed a high correlation with antigen processing and presentation ssGSEA score (Spearman $r = 0.40$, $P = 0.003$) (Fig. EV6A). To validate that NDUFB7 was dominantly expressed in the ESCC cells in the tumor biopsy, we performed multiplex immunofluorescence staining based on the markers of different cell types in the tumor microenvironment (PANCK for ESCC cells, CD45 for immune cells, and α-SMA for stromal fibroblast cells), which was adopted in

the previous researches (Chen et al, 2024; Ren et al, 2024), as well as NDUFB7 for mitochondrial complex I. The result showed that NDUFB7, the component of mitochondrial complex I, was mainly expressed in PANCK + ESCC cells instead of in CD45+ immune cells or α-SMA+ stromal fibroblast cells (Fig. EV6B). The same result was also observed in the scRNA-seq dataset of ESCC (Pan et al, 2022), where NDUFB7 was mainly expressed in the malignant squamous epithelial cell cluster compared with other cell clusters within the ESCC biopsies (Fig. EV3E), further validating that NDUFB7, associated with ESCC immunotherapy response, was highly enriched in tumor cells, rather than in other cells. Moreover, we observed that the S group had a significantly increased percentage of NDUFB7-positive tumor cells (81.9%) compared with the NS group (41.5%) by IHC staining (two-sided Student's *t* test, $P < 0.0001$) (Fig. EV6C). Multiplex immunofluorescence staining results also showed that NDUFB7 expression was higher in the immunotherapy-sensitive (S) group compared to the non-sensitive (NS) group, further confirming the association between mitochondrial complex I protein expression and ESCC immunotherapy response (Fig. 5A). Regarding the phenotype of CD8 + T cells, we found that patients with high expression of NDUFB7 exhibited high infiltration of GZMB + CD8 + T cells (representing the effector phenotype) and low levels of TIM3 + CD8 + T cells (representing the exhausted phenotype) (Fig. 5A). To investigate the effect of NDUFB7 on T cell functions, in addition to multiplex immunofluorescence staining, we also collected fresh treatment-naïve ESCC samples and analyzed intratumoral T cells by flow cytometry. Consistently, flow cytometric analysis revealed that tumors with high NDUFB7 expression displayed a higher proportion of CD8 + T cell infiltration (48.9%) compared to those with low NDUFB7 expression (24.4%) (two-sided Student's *t* test, $P = 0.046$) (Fig. 5B and EV6D). Furthermore, we found the upregulation of GZMK expression (two-sided Student's *t* test, $P = 0.022$), along with the downregulation of PD1 expression in CD8 + T cells from high NDUFB7 tumors compared with low NDUFB7 tumors (two-sided Student's *t* test, $P = 0.002$) (Fig. 5C), underscoring the high effector function of CD8 + T cells in tumors with high NDUFB7. Based on these results, we next assessed the association between NDUFB7 in tumor cells and CD8 + T cell function in the tumor microenvironment at the single-cell level. Correlation analysis also revealed a significant positive correlation between mitochondrial complex I score and CD8 + T cell cytotoxicity score in the scRNA-seq dataset

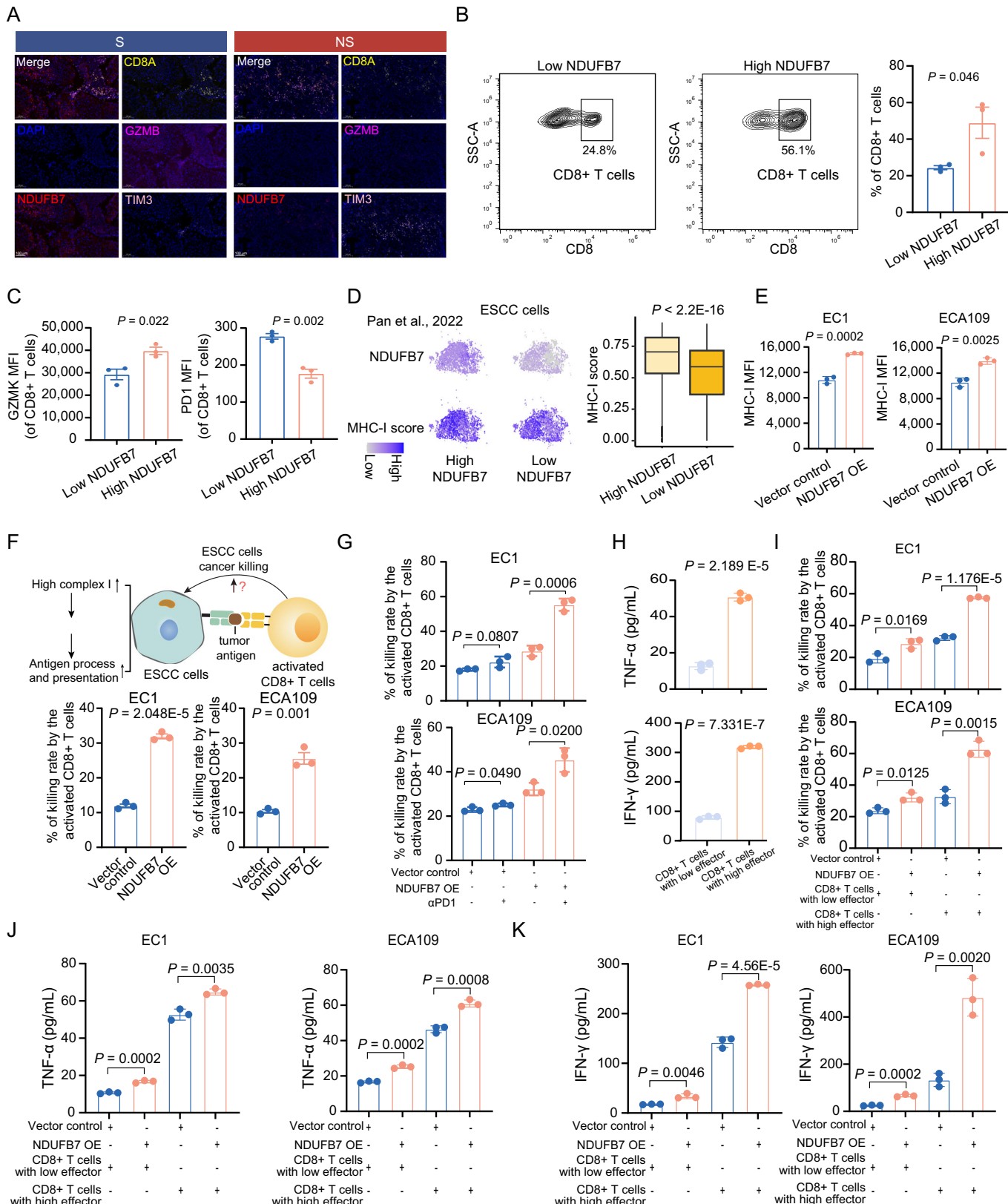

**Figure 5. NDUFB7 overexpression and its association with ESCC immunotherapy response.**

(A) Representative images of the multiplex immunofluorescence staining of NDUFB7, CD8A, TIM3, and GZMB in the S and NS groups. Scale bar: 100 μm.
(B) Representative flow cytometry plots (left) and quantification (right) for CD8 + T cells in human ESCC tumors with high ($n = 3$) and low ($n = 3$) NDUFB7 expression (two-sided Student's $t$ test). Data are presented as mean with standard error of the mean. (C) Mean fluorescence intensities (MFI) for GZMK and PD1 staining by the indicated CD8 + T cell subset in human ESCC tumors with high ($n = 3$) and low ($n = 3$) NDUFB7 expression (two-sided Student's $t$ test). Data are presented as mean with standard error of the mean. (D) t-SNE plots (left) and boxplots (right) showing the MHC-I score between high ($n = 5812$) and low ($n = 5704$) NDUFB7 ESCC cells (two-sided Wilcoxon rank-sum test). $P = 3.54E$-177. Boxplots represent the interquartile range (IQR), with the box spanning the 25th to 75th percentiles and the median indicated by a horizontal line. Whiskers extend to the most extreme data points within 1.5×IQR. (E) Flow cytometry for MHC-I expression on the surface of vector control ($n = 3$) and NDUFB7 OE ($n = 3$) ESCC cells (EC1 and ECA109) (two-sided Student's $t$ test). Data are presented as mean with standard error of the mean. (F) Diagram showing the co-cultured system composed of ESCC cells and activated CD8 + T cells. Barplots showing the killing rate by the activated CD8 + T cells co-cultured with vector control and NDUFB7 OE ESCC cells ($n = 3$ independent experiments, two-sided Student's $t$ test). Data are presented as mean with standard error of the mean.
(G) Barplots showing the killing rate by the activated CD8 + T cells co-cultured with NDUFB7 OE and vector control ESCC cells with or without the addition of anti-PD1 antibody ($n = 3$ independent experiments, two-sided Student's $t$ test). Data are presented as mean with standard error of the mean. (H) The production of TNF-α and IFN-γ in CD8 + T cells derived from different ESCC patients ($n = 3$ independent experiments, two-sided Student's $t$ test). Data are presented as mean with standard error of the mean. (I) The anti-tumor effect of high and low effector functions of CD8 + T cells on ESCC cells (vector control and NDUFB7 OE) ($n = 3$ independent experiments, two-sided Student's $t$ test). Data are presented as mean with standard error of the mean. (J, K) The production of TNF-α and IFN-γ by CD8 + T cells in the co-culture experiments ($n = 3$ independent experiments, two-sided Student's $t$ test). Data are presented as mean and standard error of the mean. Source data are available online for this figure.

of ESCC (Pan et al, 2022) (Spearman $r = 0.83$, $P = 0.048$), as well as with the frequency of GZMB + CD8 + T cells (Spearman $r = 0.89$, $P = 0.033$) (Fig. EV6E). Overall, these results indicate higher cytotoxicity of CD8 + T cells within tumors with high NDUFB7 expression compared to those with low NDUFB7 expression.

To further explore the effect of mitochondrial complex I in ESCC cells, we established two NDUFB7 overexpression (OE) stable cell lines, EC1 and ECA109 (Fig. EV6F), respectively. Blue-native polyacrylamide gel electrophoresis (BN-PAGE) followed by immunoblotting in NDUFB7 OE and vector control ESCC cells revealed that NDUFB7 overexpression increased assembly and amount of mitochondrial complex I (Fig. EV6G) ("Methods"). To elucidate the molecular perturbation associated with the increase of mitochondrial complex I protein expression in ESCC cells, we performed proteomic analysis using a label-free technique and compared the proteome change between vector control and NDUFB7 OE ESCC cells. Proteomic measurement resulted in 4198–4548 GPs in each sample (Fig. EV6H). A total of 5263 GPs were identified in all ESCC cells ($N = 12$, three repeats of each kind of ESCC cell) (Dataset EV6). Consistent with western blotting detection, we found higher NDUFB7 expression in the NDUFB7 OE ESCC cells compared with vector control by LC-MS/MS analysis (two-sided Student's $t$ test, $P = 0.046$) (Fig. EV6I). Furthermore, we found that the ssGSEA score of mitochondrial complex I, but not mitochondrial complexes II–V, was significantly higher in NDUFB7 OE ESCC cells compared with vector control ESCC cells (Fig. EV6J). Differential proteomic analysis between NDUFB7 OE and vector control ESCC cells identified 494 significantly upregulated proteins in the NDUFB7 OE group, which were enriched in complex I biogenesis ($P = 0.017$), autophagy ($P = 0.011$), antigen processing and presentation ($P = 0.025$), etc. (Fig. EV6K). Proteins involved in mitochondrial complex I (including ND5, NDUFA3, NDUFAF3, and NDUFB7), autophagy (including ATG4C, BECN1 (ATG6), CTSB, and TBK1), and antigen processing and presentation (including CALR, CANX, CREB, HLA-C, and TAP2) showed an obvious increase in NFUFB7 OE cells (Fig. EV6L; Dataset EV6).

Beyond the association between the mitochondrial complex I protein expression and the immune response, we wonder whether this association depends on the response to T cell-mediated killing.

Antigen processing and presentation efficiency were one of the fundamental determinants of tumor immunogenicity. CD8 + T cells are more likely to bind to tumor antigens presented on tumor cells by MHC-I molecules via their T-cell receptor, causing enhanced tumor killing (Blankenstein et al, 2012; Waldman et al, 2020). To validate the relationship between mitochondrial complex I and MHC-I expression, we analyzed the scRNA-seq dataset from ESCC patients (Pan et al, 2022). Notably, in the scRNA-seq dataset of ESCC, tumor cells with high NDUFB7 expression exhibited significantly elevated MHC-I scores compared to those with low NDUFB7 expression (Pan et al, 2022) (two-sided Wilcoxon rank-sum test, $P < 2.2E$-16) (Fig. 5D). Consistently, the flow cytometry analysis demonstrated a markedly elevated level of MHC-I expression on the surface of ESCC cells with high mitochondrial complex I protein expression (NDUFB7 OE) in comparison to the vector control ESCC cells, which was a major determinant of CD8 + T cell activation and cytotoxicity on tumor cells (Fig. 5E). This suggested that mitochondrial complex I regulated MHC-I expression on ESCC cells, which in turn affected their interaction with CD8 + T cells and the immune response against the tumor cells. To directly examine the efficacy of mitochondrial complex I on T cell-dependent tumor immune responses, we constructed a co-culture system in which the NDUFB7 OE or vector control ESCC cells were co-cultured with CD8 + T cells and performed cytotoxic T-cell killing assays as described previously (Pan et al, 2018; Zhou et al, 2022). We found that the killing rate by the activated CD8 + T cells on the NDUFB7 OE ESCC cells had at least a two-fold increase compared with the vector control ESCC cells. In agreement with the increase of antigen processing and presentation proteins, CD8 + T cell killing was significantly augmented upon the high mitochondrial complex I protein expression of ESCC cells (Fig. 5F). To further demonstrate the role of mitochondrial complex I in ESCC cells in the context of anti-PD1 treatment, we added a human anti-PD1 antibody into the co-culture system and observed that the anti-PD1 treatment enhanced the anti-tumor effect of CD8 + T cells on ESCC cells, particularly those with high mitochondrial complex I protein expression (Fig. 5G). This suggested a positive role of mitochondrial complex I in ESCC cells in anti-PD1 immunotherapy response.

To further illustrate that the high mitochondrial complex I protein expression in ESCC cells can enhance CD8 + T cell-mediated killing irrespective of CD8 + T cell effector function, we isolated CD8 + T cells from patients with different CD8 + T cell effector functions, as indicated by the secretion levels of TNF-α and IFN-γ. ELISA analysis revealed that CD8 + T cells with high effector function produced higher levels of TNF-α and IFN-γ compared to those with low effector function, indicating these patients had different effector functions of CD8 + T cells (Fig. 5H). Next, we co-cultured these CD8 + T cells with ESCC cells (vector control and NDUFB7 OE ESCC cells). The results showed that CD8 + T cells, irrespective of their high or low effector activity, exhibited a superior killing effect on NDUFB7 OE ESCC cells compared to the vector control ESCC cells (Fig. 5I). In addition, CD8 + T cells (both high and low effector functions) produced more TNF-α and IFN-γ when co-cultured with NDUFB7 OE ESCC cells (Fig. 5J,K), which suggested that the elevated mitochondrial complex I protein expression of ESCC cells was responsible for the significant anti-tumor activity of CD8 + T cells.

Besides, dendritic cells (DCs) are a vital type of antigen-presenting cell that capture tumor antigens and play an important role in assisting CD8 + T cells in targeting tumors. Therefore, to simulate antigen processing and presentation, we added DCs with a high purity of 91.6% verified by surface markers CD11c using flow cytometry (Fig. EV7A), the professional antigen-presenting cell, in the co-culture system. We co-cultured immature DCs with NDUFB7 OE ESCC cells or vector controls to induce DC maturation, making them better present tumor antigens to CD8 + T cells. Mature DCs were co-cultured with CD8 + T cells to activate CD8 + T cells. Finally, the CD8 + T cells activated by mature DCs were further co-cultured with the NDUFB7 OE or vector control ESCC cells, and cytotoxic T cell killing assays were performed using cell counting kit-8 (CCK8). The result showed that the killing rate of NDUFB7 OE ESCC cells by activated CD8 + T cells exhibited a significant increase in both EC1 and ECA109, compared with vector controls (Fig. EV7B), which further validated that high mitochondrial complex I protein expression of ESCC cells enhanced tumor killing by activated CD8 + T cells. These results demonstrated that the high mitochondrial complex I protein expression of ESCC cells could promote DC maturation to augment CD8 + T cell-mediated killing.

Tumor organoids, the three-dimensional primary tumor cell cultures, stably preserve the complex diversity and physical architecture of their original tissues compared with conventional two-dimensional monolayer tumor cells, which makes them an ideal tool in tumor-immune recognition research. To recapitulate the association of ESCC mitochondrial complex I with the immune response of CD8 + T cells, we obtained fresh tumor tissues from the surgical resections of treatment-naïve ESCC patients and established ESCC tumor organoids, then co-cultured these tumor organoids with autologous CD8 + T cells isolated from their matched peripheral blood mononuclear cells (PBMCs) according to the widely adopted method for studying the tumor-immune interaction in vitro (Cattaneo et al, 2020; Chalabi et al, 2020; Dijkstra et al, 2018; Zhou et al, 2021) (Fig. 6A) ("Methods"). We performed proteome measurements of the tumor tissues and evaluated the mitochondrial complex I level according to the ssGSEA score. The ESCC organoids were divided into high ($n = 3$, average mitochondrial complex I score = 0.46) and low ($n = 3$,

average mitochondrial complex I score = -0.61) mitochondrial complex I groups based on their intrinsic differential expression of mitochondrial complex I (Fig. 6B). The 6 ESCC tumor organoids were then co-cultured with or without matched CD8 + T cells. During the co-culture of tumor organoids and CD8 + T cells, live imaging showed that tumor organoids with high mitochondrial complex I score exhibited obvious disintegration after 24-hour co-culture, suggesting strong CD8 + T cell-mediated killing of tumor organoids. With the increase of co-culture time (up to 48 h and 72 h), the disintegration became more obvious, indicating a stronger killing effect. On the contrary, tumor organoids with low mitochondrial complex I score showed little killing effect during the co-culture, even after 72-hour co-culture, suggesting a low proportion of cell death (Fig. 6C). As a further control, the tumor organoids without CD8 + T cells were not affected morphologically regardless of the organoids with high mitochondrial complex I score or low mitochondrial complex I score (Fig. 6C). Quantification of killing rate of tumor organoids showed a significant decrease in the proportion of death cell in tumor organoids with high mitochondrial complex I score, compared with organoids with low mitochondrial complex I score (two-sided Student's $t$ test, $P = 1.047E-12$) (Fig. 6D), indicating the stronger killing effect of CD8 + T cells on tumor organoids with high mitochondrial complex I score. In conclusion, the results demonstrated that high mitochondrial complex I protein expression was linked to an efficacious immunotherapeutic response. Consequently, ESCC patients with high mitochondrial complex I protein expression might derive superior therapeutic benefit from immunotherapy in clinical treatment.

## Tumor-intrinsic mitochondrial complex I enhances the anti-tumor response of anti-PD1 treatment in mice

To investigate whether ESCC mitochondrial complex I affects the efficacy of anti-PD1 immunotherapy in vivo, we utilized the well-established AKR murine ESCC model (Jiang et al, 2024; Opitz et al, 2002). We first generated the vector control and Ndufb7 OE AKR cells, and the expression of Ndufb7 was assessed by immunoblotting (Fig. EV8A). The vector control and Ndufb7 OE cells were implanted subcutaneously into immunocompetent C57BL/6 J mice. These tumor-bearing mice were treated with either anti-PD1 or isotype IgG when tumors grew to an appropriate size (Fig. 6E) ("Methods"). No significant body weight loss was observed in these mice during the treatment (Fig. EV8B). In line with the in vitro co-culture experiment, mice implanted with mouse ESCC cells (AKR) expressing high mitochondrial complex I protein (Ndufb7 OE) showed a significant decrease in tumor growth compared with the vector control group (two-sided Student's $t$ test, $P < 0.05$). It is of particular note that tumor with high mitochondrial complex I protein expression (Ndufb7 OE) demonstrated a significant response to anti-PD1 antibody treatment, with a notable reduction in tumor volume and tumor weight when compared to the vector control group (two-sided Student's $t$ test, $P < 0.05$) (Fig. 6F–H). Furthermore, as aforementioned, YAP1 expression was associated with immunotherapy non-sensitivity. We speculated that the inhibition of YAP1 by its specific inhibitor (CA3) might contribute to improving immunotherapy outcomes. Therefore, we treated mice with CA3 and the combination of anti-PD1 and CA3, respectively. The results demonstrated that the combination

treatment of anti-PD1 and CA3 significantly inhibited the tumor growth and tumor weight in comparison to anti-PD1 or CA3 treatment alone (two-sided Student's $t$ test, $P < 0.05$), but had no influence on the mouse body weight (ANOVA test, $P = 0.4308$) (Figs. 6F–H and EV8B).

We employed immunohistochemistry (IHC) staining on the paraffin-embedded tumor blocks of the allograft tumor. Firstly, IHC staining of PD-L1 revealed that neither Ndufb7 OE nor CA3 treatment affected PD-L1 expression, which further confirmed that PD-L1 has a low potential to guide ESCC immunotherapy (two-sided Student's $t$ test, $P > 0.05$) (Fig. EV8C). In contrast, tumor treatment with YAP1 inhibitor (CA3) could significantly reduce Yap1 expression, but increase the expression of Ndufb7 (two-sided Student's $t$ test, $P < 0.05$) (Fig. EV8C). Secondly, the IHC staining of CD8 and CD4 revealed that there was more infiltration of CD8 + T cells in the Ndufb7 OE tumor in comparison to the vector control tumor (two-sided Student's $t$ test, $P < 0.05$) (Fig. 6I,J). However, no significant difference was observed in the infiltration of CD4 + T cells (two-sided Student's $t$ test, $P > 0.05$) (Fig. 6I,J). A similar tendency of the infiltration of CD8+ and CD4 + T cells was also observed both in the Ndufb7 OE tumor and vector control tumor with the treatment of YAP1 inhibitor (CA3). This indicated that the inhibition of YAP1 primarily influenced the infiltration of CD8 + T cells. More importantly, the combination of anti-PD1 and CA3 treatment showed a more pronounced impact on the infiltration of CD8 + T cells in comparison to anti-PD1 or CA3 treatment alone, with no effect on the CD4 + T cell infiltration in the NDUFB7 OE tumor and vector control tumor. The IHC staining of TNF-α (CD8 + T cell effect marker) showed the same tendency of CD8 + T cells in the combination of anti-PD1 and CA3 treatment. In contrast, in the tumor microenvironment of the combination of anti-PD1 and CA3 treatment, the infiltration of CD8 + T cells expressed less PD1 (CD8 + T cell exhausted marker) in comparison to anti-PD1 or CA3 treatment alone in both vector control tumor and NDUFB7 OE tumor (two-sided Student's $t$ test, $P < 0.05$) (Fig. 6I,J). Overall, these results indicated that ESCC with high mitochondrial complex I protein expression could be more responsive to anti-PD1 immunotherapy and facilitate more significant CD8 + T cell infiltration within the tumor microenvironment, and there was a synergistic anti-tumor effect of anti-PD1 and CA3 (YAP1 inhibitor) in the in vivo experiment.

## Construction and validation of the predictive model for ESCC immunotherapy

The limited predictive power of PD-L1 expression on ESCC immunotherapy response highlights the necessity to identify highly effective biomarkers for ESCC immunotherapy response. Aiming to personalize immunotherapy guidance, we next set out to determine whether the signatures associated with immunotherapy response could distinguish sensitive patients from non-sensitive patients in response to immunotherapy (Fig. 7A). We randomized our discovery cohort into a training set (70%, $N = 37$; $N$ (NS) = 19, $N$ (S) = 18), for which we developed the prediction classifier, and a testing set (30%, $N = 16$; $N$ (NS) = 10, $N$ (S) = 16), on which we evaluated the trained classifier ("Methods"). First, we calculated the AUC values of 124 proteins from the pathways associated with immunotherapy response (including the Hippo pathway, mitochondrial complex I, and antigen processing and presentation), for

immunotherapy prediction and selected the top 10 proteins based on their AUC values as candidate biomarkers for the model construction (Robin et al, 2011; Weller et al, 2025) (Fig. 7A). Second, considering practical translational applications and clinical feasibility, we aimed to optimize predictive performance with a reduced number of proteins. Therefore, we employed a commonly adopted feature selection approach (Demir et al, 2025; Fujii et al, 2021; Li et al, 2023; Yoneshiro et al, 2025), the backward stepwise method, to select a subset of proteins (including YAP1, NDUFB7, MHC-I, and CD8A) from these top 10 candidates for model construction. Finally, these four signature proteins were used to construct a predictive model for ESCC immunotherapy response. Based on these signatures, we applied three-fold cross-validation to tune the hyperparameter on the training set to derive the ESCC immunotherapy response predictive model with high accuracy (0.81), sensitivity (84%), specificity (78%), and F1 score (0.81) (Fig. 7B). When applied to the independent testing set samples, the predictive model achieved 0.88 accuracy, 90% sensitivity, 83% specificity, and 0.88 F1 score (Fig. 7C). We then calculated the area under the receiver operating characteristic (ROC) curves and the area under the curve (AUC). The ROC curves showed high sensitivity and specificity of immunotherapy response prediction with an AUC of 0.90 on both the training and testing sets (Fig. 7B,C).

Furthermore, the model was also validated in an independent validation cohort, which consisted of 55 ESCC patients receiving the same immunotherapy regimen, including 19 S patients and 36 NS patients (Fig. 7A; Dataset EV7). Notably, the model also achieved high accuracy (0.91), sensitivity (93%), specificity (87%), and F1 score (0.91) in the independent validation cohort (Fig. 7D). The ROC curve with the AUC of 0.91 also indicated the excellent predictive power in the independent validation cohort (Fig. 7D). Altogether, these results demonstrated that the predictive model could forecast response before immunotherapy administration based on the four signatures.

To further validate the association of the YAP1 or mitochondrial complex I with ESCC immunotherapy response, we also collected additional samples following the same patient enrollment criteria as the IHC validation cohort. In the IHC validation cohort, a total of 82 treatment-naïve ESCC samples undergoing anti-PD1 immunotherapy were collected, including 30 S and 52 NS patients. The represented IHC staining results of the four signature proteins, including YAP1, NDUFB7, CD8A, and MHC-I, were shown in Fig. 7E. For the IHC score analysis, protein abundance was quantified according to the adopted scoring system (on a scale of 0–12) (Luo et al, 2023) ("Methods"). The IHC results demonstrated that staining of YAP1 showed a significant increase in NS patients (two-sided Student's $t$ test, $P = 1.19E-7$), whereas NDUFB7, one of the components of mitochondrial complex I, was significantly increased in S patients (two-sided Student's $t$ test, $P = 2.88E-9$) (Fig. 7E). Meanwhile, the staining of the biomarker of CD8 + T cells (CD8A) also displayed higher CD8 + T cell infiltration in S patients compared to NS patients (two-sided Student's $t$ test, $P = 5.43E-7$), as well as the MHC-I expression (two-sided Student's $t$ test, $P = 8.65E-11$) (Fig. 7E), which were involved in the antigen processing and presentation of CD8 + T cells in anti-tumor response. We further assessed the clinical significance of these proteins in ESCC immunotherapy response based on the IHC score analysis in the IHC validation cohort. When applying the

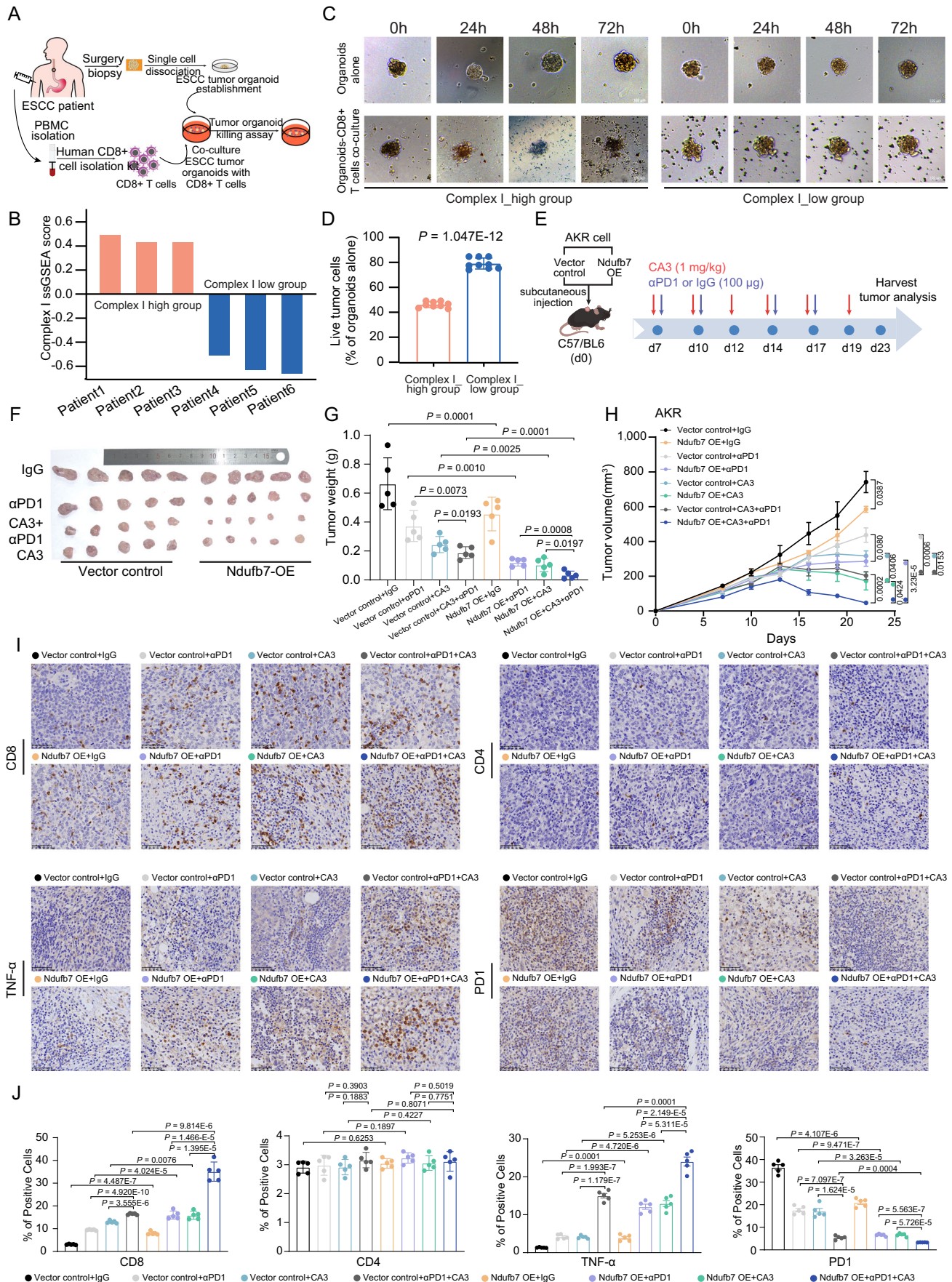

◀ **Figure 6. The mitochondrial complex I of ESCC enhances the CD8 + T cell- mediated killing in vitro and vivo models.**

(A) Experiment workflow. Tumor organoids were established from ESCC surgical resections and co-cultured with autologous CD8 + T cells isolated from PBMCs. After co-culture, a killing assay of tumor organoids by CD8 + T cells was performed. (B) Tumor organoids from patients with high or low mitochondrial complex I scores calculated by the intrinsic protein expression. (C) Bright-field images on days 0–3 showed the microphotograph of ESCC tumor organoid morphology in the mitochondrial complex I high and low groups with or without CD8 + T cells co-culture. Scale bar: 100 μm. (D) Barplots showing the quantification of the killing rate of ESCC organoids co-cultured with matched CD8 + T cells (n = 3 independent experiments, two-sided Student's t test). Data are presented as mean with standard deviation. (E) Schematic representation of AKR (vector control and Ndufb7 OE) allograft experiments in vivo. AKR (Ndufb7 OE and vector control) cells were subcutaneously injected into immunocompetent C57BL/6 mice, and mice were intraperitoneally treated with isotype IgG antibody, anti-PD1 antibody, YAP1 inhibitor (CA3), and the combination of anti-PD1 and CA3, respectively. (F) Representative tumor images in immunocompetent C57BL/6 J mice bearing subcutaneous AKR (Ndufb7 OE and vector control) allografts (n = 5/group), and mice were intraperitoneally treated with IgG or anti-PD1 antibodies, YAP1 inhibitor (CA3), and the combination of anti-PD1 and CA3. (G) Tumor weights in the AKR allograft model (n = 5/group, two-sided Student's t test, mean with standard deviation). (H) Mean tumor growth curve showing tumor volumes in mice bearing subcutaneous AKR (Ndufb7 OE and vector control) allografts, and mice were administered to IgG or anti-PD1 antibodies, YAP1 inhibitor (CA3), and the combination of anti-PD1 and CA3 (n = 5/group, two-sided Student's t test, mean with standard error of the mean). (I) Representative IHC staining images of CD8, CD4, TNF-α, and PD1 in AKR (Ndufb7 OE and vector control) allografts and mice received IgG or anti-PD1 antibodies, YAP1 inhibitor (CA3), and the combination of anti-PD1 and CA3 (n = 5/group). (J) Barplots showing the qualification of CD8, CD4, PD1, and TNF-α stained by immunohistochemistry (IHC) in the AKR (Ndufb7 OE and vector control) tumor samples (n = 5/group, two-sided Student's t test, mean with standard deviation). Source data are available online for this figure.

predictive model on the IHC validation cohort, the model also achieved high accuracy (0.90), sensitivity (92%), specificity (87%), and F1 score (0.90) (Fig. 7F). ROC analysis demonstrated that a combined prediction of these signatures showed good performance (AUC = 0.96) in distinguishing S patients from NS patients receiving immunotherapy (Fig. 7F). We also calculated the AUC value of the combination of PD-L1 with four signature proteins in predicting immunotherapy response and found the combination of PD-L1 with four signature proteins could not improve the AUC for immunotherapy response prediction when compared to the use of the four signature proteins alone in the discovery cohort, validation cohort, and IHC validation cohort (Fig. EV8D); on the contrary, the combination resulted in a decrease in AUC. When the immunotherapy response prediction model was applied to the TCGA-ESCC cohort based on the expression of four signatures (YAP1, NDUFB7, MHC-I, and CD8A), it predicted 89 ESCC patients as 33 S patients and 56 NS patients. Survival analysis showed that the predicted S patients had significantly longer survival than NS patients (two-sided log-rank test, P = 0.012) (Fig. EV8E), while no significant difference in TMB levels between S and NS patients was observed in the TCGA-ESCC cohort (two-sided Student's t test, P = 0.641) (Fig. EV8F), which further demonstrated the association of our predictive model with ESCC clinical outcome. Overall, these results further supported the clinical value of YAP1-mitochondrial complex I in predicting immunotherapy response.

## Discussion

ESCC is the principal histological type of esophageal cancer in Asia, especially in China, which has the highest incidence and mortality compared with other countries (Allemani et al, 2018). Due to the lack of targeted approaches for treatment, the 5-year survival rate of ESCC patients remains dismal. In recent years, immune checkpoint inhibition and immunotherapy in combination with cytotoxic agents have been investigated to increase the therapeutic response to immune checkpoint inhibitors for ESCC patients and achieve durable efficacy (Minn and Wherry, 2016). However, significant challenges remain, including the absence of reliable biomarkers to predict immunotherapy response and unresolved issues related to the development of resistance to treatment. Therefore, it is

imperative to make a preclinical diagnosis of immunotherapy response using reliable biomarkers. Herein, we collected 190 FFPE biopsy samples from patients with ESCC before the initiation of camrelizumab-based anti-PD1 immunotherapy and presented an unprecedented large-scale clinical proteomic landscape for ESCC immunotherapy. In addition, we have evaluated the equivalence of proteomic and phosphoproteomic profiling between FFPE and fresh ESCC tissues. Consistent with previous reports (Humphries et al, 2025; Sprung et al, 2009), we found that FFPE samples yield highly consistent proteomic and phosphoproteomic data when compared with fresh tissues, with substantial overlap in protein and phosphosite identifications as well as concordant detection of immunotherapy-relevant molecular features (Fig. EV9). FFPE samples are suitable for use in proteomic studies, supporting biomarker discovery and mechanistic investigations of disease (Deng et al, 2023; Knol et al, 2025; Qin et al, 2025; Xu et al, 2022), with a level of equivalence comparable to fresh tissues.

In the study, we found a weak correlation between PD-L1 expression assessed by IHC and the immunotherapy response, as confirmed in other studies for ESCC patients who received the same regimen. Patients with PD-L1-negative expression could also benefit from immunotherapy, and not all PD-L1-positive patients were sensitive to immunotherapy, highlighting the poor detection power of PD-L1 in immunotherapy response. As reported, in some clinical trials, patients with different PD-L1 expression (including PD-L1-negative and -positive cases) were enrolled to assess the efficacy of immunotherapy, and the results showed that some PD-L1-negative patients could also benefit from immunotherapy (Luo et al, 2021; Sun et al, 2021). This suggested multifactorial tumor-specific mechanisms of immunotherapy response (such as the downregulation of MHC-I molecule) and the need to identify complementary biomarkers to discriminate tumor immunotherapy response more accurately. Further analysis of the proteomic difference between S and NS groups with different PD-L1 statuses, including PD-L1-positive and PD-L1-negative, found a similar featured pathway in S or NS groups at both PD-L1 statuses. In both PD-L1 expression statuses, respiratory chain complex I was related to immunotherapy sensitivity, while platelet activation was associated with resistance.

The proteomic data analysis from the clinical patient cohort revealed a positive role of ESCC mitochondrial complex I in anti-PD1 immunotherapy. The increase in mitochondrial complex I

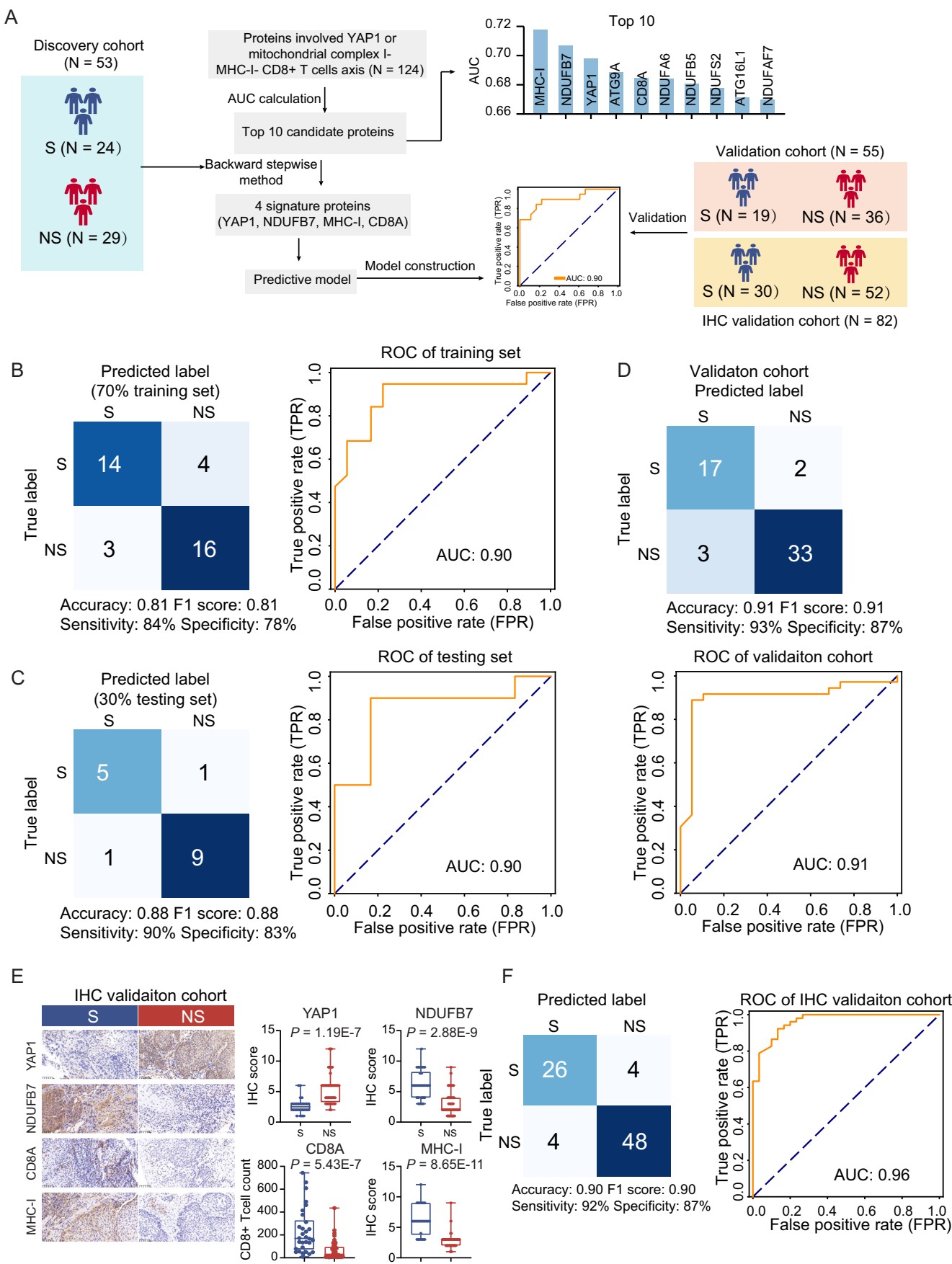

**Figure 7. The construction and validation of the predictive model for immunotherapy response.**

(A) Diagram describing the construction and validation of the predictive model for sensitive (S) and non-sensitive (NS) groups. (B, C) Classification error matrix and ROC curves of 70% training set (B) and 30% testing set (C) in the discovery cohort. The number of samples identified is noted in each box (right). (D) Classification error matrix and ROC curve showing high sensitivity and specificity of the signatures in the independent validation cohort. (E) The IHC validation cohort (including 82 ESCC patients: *N* (S) = 30, *N* (NS) = 52) was used to verify the level of YAP1, NDUFB7, CD8A, and MHC-I in the S and NS patients. Scale bar: 50 μm. Boxplots showing the differences of four signatures using IHC staining between S and NS patients (two-sided Student's *t* test). Boxplots represent the interquartile range (IQR), with the box spanning the 25th to 75th percentiles and the median indicated by a horizontal line. Whiskers mark minimum or maximum values. (F) Classification error matrix and ROC curve showing high sensitivity and specificity of the signatures in the IHC validation cohort. Source data are available online for this figure.

protein expression could upregulate the expression of MHC-I molecules to enhance the CD8 + T cell-mediated killing. Thus, activating mitochondrial complex I by its specific agonist may be a strategy to improve ESCC immunotherapy. Through phosphoproteomic data analysis, we found that YAP1 expression was associated with ESCC immunotherapy response. YAP1, an oncogenic protein, the duration of activation can promote cancer development, invasion, and migration in a range of cancers (Dey et al, 2020) (including colorectal cancer, esophageal carcinoma, and hepatocellular carcinoma). It is also an attractive target for therapeutic intervention, and its inhibitors, such as the small-molecule CA3, which synergizes with 5-FU to inhibit esophageal cancer cell proliferation (Song et al, 2018). In our study, we found that inhibition of YAP1 could increase the T cell-mediated anti-PD1 immunotherapy in the syngeneic murine ESCC model. Thus, our study demonstrated that the inhibitor of YAP1 had a synergistic effect with PD1 blockade to improve the efficacy of immunotherapy in ESCC. The combination of YAP1 inhibition with anti-PD1 immunotherapy could serve as a potential therapeutic strategy for ESCC patients.

In this study, we employed proteomic technology implemented with a machine learning method to validate that the signatures associated with immunotherapy response (including YAP1, NDUFB7, MHC-I, and CD8A) for immunotherapy response prediction. The predictive markers exhibited high predictive power in discriminating between S and NS patients, with an AUC of 0.90 observed in the discovery cohort. Furthermore, these signatures were validated in an independent cohort, achieving an AUC of 0.91, as well as in the IHC validation cohort with an AUC of 0.96, indicating their clinical significance for ESCC immunotherapy.

In summary, the comprehensive proteomic analysis described an atlas of immunotherapy in ESCC. This study identified three proteomic subtypes and four phosphoproteomic subtypes correlated with immunotherapy response and found that high expression of mitochondrial complex I protein could enhance anti-PD1 immunotherapy response. Through both in vivo and in vitro experiments, we further demonstrated that ESCC with high mitochondrial complex I protein expression promoted CD8 + T cell-mediated anti-PD1 immunotherapy response. The combination of YAP1 inhibition and anti-PD1 immunotherapy showed great value in improving the immunotherapy response in vivo. Finally, we constructed a predictive model for predicting ESCC immunotherapy response, which could distinguish S patients from NS patients and contribute to personalized immunotherapy of ESCC patients.

## Limitations

For the systemic treatment, we used a YAP1 inhibitor in the allograft experiment instead of YAP1 conditional mouse models,

which was a valid alternative method to explore the impact of YAP1 on immunotherapy. The impact of genetic ablation of YAP1 on immunotherapy response in ESCC remains to be elucidated and warrants further investigation in future studies.

# Methods

**Reagents and tools table**

| Reagent/resource | Reference or source | Identifier or catalog number |
|---|---|---|
| **Experimental models** | | |
| C57BL/6J mice | Shanghai Laboratory Animal Center | |
| EC1 | Shanghai Institutes for Biological Science (Cao et al, 2020) | |
| ECA109 | Shanghai Institutes for Biological Science (Cao et al, 2020) | |
| AKR | Otwo Biotech | HTX2545 |
| **Recombinant DNA** | | |
| pMD2.G plasmid | Addgene | 12259 |
| psPAX2 plasmid | Addgene | 12260 |
| pLKO.1-puro vector | Addgene | 8453 |
| pCDH vector | Addgene | 72265 |
| **Antibodies** | | |
| Mouse anti-GAPDH | Proteintech | 60004-1-Ig |
| Rabbit anti-NDUBF7 | Abcam | ab188575 |
| Mouse anti-flag | Sigma | M20008L |
| Rabbit anti-HLA Class 1 ABC | Abcam | ab225636 |
| Rabbit anti-YAP1 | Cell Signaling Technology | 14074 |
| Mouse anti-CD8A | Cell Signaling Technology | 70306 |
| Rabbit anti-CD4 | Cell Signaling Technology | 25229 |
| Rabbit anti-PD1 | Abcam | ab214421 |
| Rabbit anti-PD-L1 | Cell Signaling Technology | 64988 |
| Mouse anti-PANCK | Dako | GA05361-2 |
| Rabbit anti-CD45 | Abcam | ab40763 |
| Rabbit anti-α-SMA | Cell Signaling Technology | 19245 |
| Rabbit anti-TIM3 | Abcam | ab241332 |

| Reagent/resource | Reference or source | Identifier or catalog number |
|---|---|---|
| Rabbit anti-GZMB | Abcam | ab255598 |
| Mouse anti-CD3 | BioXCell | CDE-M120a |
| Mouse anti-CD28 | BioXCell | BE0248 |
| InVivo anti-mouse PD-1 | BioXcell | BE0146 |
| InVivo IgG isotype control | BioXcell | BE0089 |
| InVivo anti-human PD-1 | BioXcell | BE0188 |
| **Oligonucleotides and other sequence-based reagents** | | |
| YAP1-shRNA1 | This study | CCCAGTTAAAT GTTCACCAAT |
| YAP1-shRNA2 | This study | GCCACCAAGCT AGATAAAGAA |
| YAP1-shRNA3 | This study | CTCAGGAATTG AGAACAATGA |
| shRNA control | This study | CAACAAGATG AAGAGCACCAA |
| NDUFB7 overexpression | This study | F:acctccatag aagattctagaA TGGGGGCGC ACCTGGTC R:atccttcgcgg ccgcggatccTT AAACCTTATC GTCGTCATCCTTG |
| **Chemicals, enzymes, and other reagents** | | |
| Trypsin | Promega | V5280 |
| High-Select Fe-NTA kit | Thermo Fisher Scientific | A32992 |
| Lipofectamine 2000 transfection reagent | Invitrogen | 11668-019 |
| Hexadimethrine bromide | Sigma | H9268 |
| Human IL-2 | PeproTech | 200-02 |
| Human GM-CSF | PeproTech | 300-03-100UG |
| Human IL-4 | PeproTech | 200-04-50UG |
| Human TNF-α | PeproTech | 300-01A-50UG |
| Human IFN-γ | PeproTech | 300-02 |
| Collagenase Type II | VETEC | V900892 |
| Collagenase Type I | VETEC | V900891 |
| Dispase | Sigma | D4693 |
| DNase I | Merck | 10104159001 |
| Gelmatrix | Corning | 356231 |
| advanced DMEM/F12 | Gibco | 12634010 |
| Glutamax | Gibco | 35050061 |
| HEPES | Gibco | 15630080 |
| Penicillin/Streptomycin | Gibco | 15140122 |
| Gentamicin | Corning | 30-005-CR |
| N2 Supplement | Gibco | 17502048 |
| B27 Supplement | Gibco | 17504044 |
| N-acetylcysteine | Sigma | A9165 - 5 G |
| Y-27632 | PeproTech | 1293823 |

| Reagent/resource | Reference or source | Identifier or catalog number |
|---|---|---|
| Human EGF | PeproTech | AF-100-15 |
| Noggin | PeproTech | 120-10 C |
| R-spondin | PeproTech | 120-38 |
| TrypLE | Gibco | 12604021 |
| YAP1 inhibitor (CA3) | SELLECK | S8661 |
| **Software** | | |
| Firmiana | http://www.firmiana.org | |
| iProx | www.iprox.cn | |
| PhosphoSitePlus | https://www.phosphosite.org | |
| ConsensusPathDB | http://cpdb.molgen.mpg.de/ | |
| Molecular Signature Database | https://www.gsea-msigdb.org/gsea/msigdb | |
| R software | https://www.r-project.org/ | |
| R Studio | https://rstudio.com/products/rstudio/download | |
| GSEA software | http://software.broadinstitute.org/gsea/index.jsp | |
| Visual Studio Code | https://code.visualstudio.com/ | |
| FlowJo software | https://www.flowjo.com/ | |
| ImageJ software | https://imagej.nih.gov/ij/ | |
| **Other** | | |
| Animal Cell Mitochondrial Isolation and Electrophoresis Sample Preparation Kit | Real-times | RTD8117 |
| Blue/Clear Native Page Electrophoresis Kit | Real-times | RTD6139-0312 |
| Tumor Dissociation Kits | Miltenyi Biotec | 130-095-929 |
| MojoSort™ Human CD8 T Cell Isolation Kit | Biolegend | 480129 |
| Human TNF-α ELISA Kit | StemCell | 100-1142 |
| Human IFN-γ ELISA Kit | StemCell | 02002 |
| Q Exactive HF-X Hybrid Quadrupole-Orbitrap Mass Spectrometer | Thermo Fisher Scientific | |

## Samples collection

Treatment-naïve archival formalin-fixed, paraffin-embedded (FFPE) tissues from ESCC patients were obtained. Samples of the discovery cohort and validation cohort were reviewed in the Department of Pathology, Shanghai Chest Hospital, Shanghai Jiao Tong University (Shanghai, P. R. China), those of IHC validation

cohort were from Fujian Medical University Union Hospital (FuJian, P. R. China). The study was compliant with the ethical standards of the Helsinki Declaration and was approved by the institutional review board of Shanghai Chest Hospital (IS23072) and Fujian Medical University Union Hospital (2024KY059). Written informed consent was obtained from each patient before any study-specific investigation was conducted.

This study consisted of 190 ESCC patients treated with the first-line camrelizumab-based immunotherapy regimen. All the therapy regimens were given at standard dosing as described in previous studies. Patients received two cycles of drug treatment; in each 21-day cycle, the following therapy regimens were administered intravenously: camrelizumab (200 mg) on day 1, nab-paclitaxel (260 mg/m²) on day 1, and carboplatin (area under the curve 5; 5 mg/ml/min) on day 1 (Yang et al, 2022). After the first two treatment cycles, all samples were histologically scored by three expert digestive system pathologists (ZXS, CX, YH, WGZ, and QMK) according to the tumor regression grade (TRG) assessment criteria through measurement of the percentage of residual viable tumor on the resected tumor specimen after immunotherapy using previously reported methods (Liu et al, 2022b; Yang et al, 2022), and respectively grouped into: TRG-I, without residual tumor; TRG-II, 1–10% residual tumor; TRG-III, 11–50% residual tumor; TRG-IV, >50% residual tumor. Patients with TRG-I who had the pathological complete response (pCR) due to the absence of residual viable tumor were defined as sensitive (S), and those with TRG-II-TRG-IV were defined as non-sensitive (NS), consistent with the previous definition (Yang et al, 2022). The discovery cohort included 24 sensitive patients (TRG-I, $N = 24$) and 29 non-sensitive patients (TRG-II, $N = 9$; TRG-III, $N = 8$; TRG-IV, $N = 12$). The validation cohort comprised 19 sensitive patients (TRG-I, $N = 19$) and 36 non-sensitive patients (TRG-II, $N = 16$; TRG-III, $N = 6$; TRG-IV, $N = 14$). The IHC validation cohort included 82 ESCC patients from Fujian Medical University Union Hospital, including 30 S patients (TRG-I, $N = 30$) and 52 NS patients (TRG-II, $N = 17$; TRG-III, $N = 7$; TRG-IV, $N = 28$).

FFPE tissues with at least 80% tumor purity were collected from a total of 150 treatment-naïve patients with ESCC before the initial camrelizumab-based immunotherapy. The PD-L1 expression of all tumor samples was assessed by a PD-L1 immunohistochemistry kit (6E8 antibody: Abcam) and characterized according to tumor proportion score (TPS), consistent with the clinical trial (Luo et al, 2021). A tumor proportion score of 1% was used as the cutoff for PD-L1-positive and negative. The IHC staining intensity of PD-L1 less than 1% was defined as PD-L1 negative (PD-L1−), while PD-L1-positive (PD-L1 + ) was defined as having at least 1% PD-L1-positive tumor cells according to the reported method (Shen and Zhao, 2018). The detailed clinical characteristics were shown in Dataset EV1.

## Sample preparation

The biopsy tumor FFPE samples derived from the treatment-naïve ESCC patients were collected, and the tumor regions were determined by pathological examination. For proteomic and phosphoproteomic sample preparation, sections (10-μm thick) from FFPE blocks were macro-dissected, deparaffinized with xylene, and washed with ethanol. The ethanol was removed completely, and the sections were left to air-dry. For this purpose,

a hematoxylin-eosin-stained section of the same tumor was used as a reference. Areas containing 80% or more tumor cells were examined independently by the expert gastrointestinal pathologists. Each sample was assigned a new research ID, and the patient's name or medical record number used during hospitalization was de-identified.

## Protein extraction and trypsin digestion

The biopsy tumor FFPE samples were lysed in TCEP buffer (2% deoxycholic acid sodium salt, 40 mM 2-Chloroacetamide, 100 mM Tris-HCl, 10 mM Tris(2-chloroethyl) phosphate, 1 mM PFSM, 1 mM Cocktail, pH 8.5) supplemented with protease inhibitors and phosphatase at 99 °C for 30 min. After cooling to room temperature, trypsin (Promega, Madison, WI, USA, catalog No.: V5280) was added and digested for 18 h at 37 °C. 10% formic acid was added and vortexed for 3 min, followed by sedimentation for 5 min (12,000× $g$). Next, a new 1.5 ml tube with extraction buffer (0.1% formic acid in 50% acetonitrile) was used to extract the supernatant (vortex for 3 min, followed by 12,000× $g$ of sedimentation for 5 min). The collected supernatant was divided into two parts and dried using a speed-vac, one for the proteome and the other for the phosphoproteome.

## The enrichment of phosphorylated peptides

Tryptic peptides were used for phosphopeptide enrichment using a High-Select Fe-NTA kit (Thermo Fisher Scientific, Rockford, IL, USA, catalog No.: A32992) according to the kit manual and a previous report (Gao et al, 2019) with some modifications. In brief, peptides were suspended in binding/wash buffer (contained in the enrichment kit) and mixed with the equilibrated resins. The peptide-resin mixture was incubated for 30 min with three gentle blows at room temperature. Following incubation, the resins were washed thrice with binding/wash buffer and twice with water. The enriched peptides were eluted with elution buffer (contained in the enrichment kit) and immediately dried using a speed-vac at 30 °C for mass spectrometry analysis.

## Proteome and phosphoproteome analysis by LC-MS/MS analysis

For the proteomic profiling of samples, peptides were analyzed on a Q-Exactive HFX Hybrid Quadrupole-Orbitrap Mass Spectrometer (Thermo Fisher Scientific, Rockford, IL, USA) coupled with a high-performance liquid chromatography system (EASY nLC 1200, Thermo Fisher). Dried peptide samples re-dissolved in Solvent A (0.1% FA in water) were loaded to a 2-cm self-packed trap column (100-μm inner diameter, 3-μm ReproSil-Pur C18-AQ beads, Dr. Maisch GmbH) using Solvent A and separated on a 150-μm-inner-diameter column with a length of 15 cm (1.9-μm ReproSil-Pur C18-AQ beads, Dr. Maisch GmbH) over a 150 min gradient (Solvent A: 0.1% FA in water; Solvent B: 0.1% FA in 80% ACN) at a constant flow rate of 600 nL/min (0–150 min, 0 min, 4% B; 0–10 min, 4–15% B; 10–125 min, 15–30% B; 125–140 min, 30–50% B; 140–141 min, 50–100% B; 141–150 min, 100% B). The eluted peptides were ionized under 2.0 kV and introduced into the mass spectrometer. MS was performed under a data-dependent acquisition mode. For the MS1 Spectra full scan, ions with $m/z$ ranging from 300 to 1400

were acquired by the Orbitrap mass analyzer at a high resolution of 120,000. The automatic gain control (AGC) target value was set as 3E6. The maximal ion injection time was 80 ms. MS2 Spectra acquisition was performed in the ion trap mode at a rapid speed. Precursor ions were selected and fragmented with higher-energy collision dissociation (HCD) with a normalized collision energy of 27%. Fragment ions were analyzed by the ion trap mass analyzer with the AGC target at 5E4. The maximal ion injection time of MS2 was 20 ms. Peptides that triggered MS/MS scans were dynamically excluded from further MS/MS scans for 25 s.

For the phosphoproteomic samples, peptides were analyzed on a Q Exactive HF-X Hybrid Quadrupole-Orbitrap Mass Spectrometer (Thermo Fisher Scientific) coupled with a high-performance liquid chromatography system (EASY nLC 1200, Thermo Fisher Scientific). Dried peptide samples re-dissolved in Solvent A (0.1% formic acid in water) were loaded onto a 2-cm self-packed trap column (100 μm inner diameter, 3 μm ReproSil-Pur C18-AQ beads, Dr. Maisch GmbH) using Solvent A and separated on a 150-μm-inner-diameter column with a length of 30 cm (1.9-μm ReproSil-Pur C18-AQ beads, Dr. Maisch GmbH) over a 150-min gradient (buffer A: 0.1% formic acid in water; buffer B: 0.1% formic acid in 80% ACN) at a constant flow rate of 600 nL/min (0–150 min, 0 min, 4% B; 0–9 min, 4–15% B; 9–129 min, 15–30% B; 129–140 min, 30–50% B; 140–141 min, 50–100% B; 141–150 min, 100% B). The eluted phosphopeptides were ionized and detected by a Q-Exactive HF-X Hybrid Quadrupole-Orbitrap mass spectrometry. Mass spectra were acquired over the scan range of m/z 300-1400 at a resolution of 120,000 (AUG target value of 3E6 and maximum injection time 80 ms). For the MS2 scan, higher-energy collision dissociation fragmentation was performed at a normalized collision energy of 30%. The MS2 AGC target was set to $5E + 04$ with a maximum injection time of 100 ms. The peptide mode was selected for monoisotopic precursor scan, and charge state screening was enabled to reject unassigned $1+$, $7+$, $8+$, and $> 8+$ ions with a dynamic exclusion time of 40 s to discriminate against previously analyzed ions between ± 10 ppm. All data were acquired using Xcalibur software v2.2 (Thermo Fisher Scientific).

## Peptide and protein identification

MS raw files were processed using the Firmiana proteomics workstation (Feng et al, 2017) (a one-stop proteomic cloud platform: http://www.firmiana.org). Briefly, raw files were searched against the NCBI human Refseq protein database using the Mascot search engine (version 2.3, Matrix Science Inc). The mass tolerances were 20 ppm for precursor and 50 mmu for product ions collected by Q Exactive HF-X. Up to two missed cleavages were allowed. The database searching of the proteome considered cysteine carbamidomethylation as a fixed modification and N-acetylation and oxidation of methionine as variable modifications. The database searching of phosphoproteome considered cysteine carbamidomethylation as a fixed modification and N-acetylation, phosphorylation of serine, threonine, and tyrosine, and oxidation of methionine as variable modifications. Precursor ion score charges were limited to $+2$, $+3$, and $+4$. For the quality control of protein identification, the target-decoy-based strategy was applied to confirm the FDR of both peptide and protein, which was lower than 1%. Percolator was used to obtain the quality value

($q$ value), validating the FDR (measured by the decoy hits) of every peptide-spectrum match (PSM), which was lower than 1%. Subsequently, all the peptides shorter than seven amino acids were removed. The cutoff ion score for peptide identification was 20. All the PSMs in all fractions were combined to comply with a stringent protein quality control strategy. We employed the parsimony principle and dynamically increased the $q$ values of both target and decoy peptide sequences until the corresponding protein FDR was less than 1%. Finally, to reduce the false positive rate, the proteins with at least one unique peptide were selected for further investigation. Phosphosites were confirmed by the PhosphoSitePlus database (https://www.phosphosite.org).

## Label-free-based MS quantification of proteins

The one-stop proteomic cloud platform "Firmiana" was further employed for protein quantification. Identification results and the raw data from the mzXML file were loaded. Then, for each identified peptide, the extracted-ion chromatogram (XIC) was extracted by searching against the MS1 based on its identification information, and the abundance was estimated by calculating the area under the extracted XIC curve. For protein abundance calculation, the nonredundant peptide list was used to assemble proteins following the parsimony principle. The protein abundance was estimated using a traditional label-free, intensity-based absolute quantification (iBAQ) algorithm (Schwanhausser et al, 2011), which divided the protein abundance (derived from identified peptides' intensities) by the number of theoretically observable peptides. We built a dynamic regression function based on the commonly identified peptides in tumor samples. According to the correlation value $R^2$, Firmiana chose linear or quadratic functions for regression to calculate the retention time (RT) of corresponding hidden peptides and to check the existence of the XIC based on the m/z and calculated RT. Subsequently, the fraction of total (FOT), a relative quantification value, was defined as a protein's iBAQ divided by the total iBAQ of all identified proteins in one experiment and was calculated as the normalized abundance of a particular protein among experiments. Finally, the FOT was further multiplied by $10^5$ for ease of presentation, and FOTs less than $10^{-5}$ were replaced with $10^{-5}$ to adjust extremely small values (Ge et al, 2018).

## Quality control of the mass spectrometry data

To the quality control of MS performance, the HEK293T cell lysate was measured every three days as the quality control standard. Besides HEK293T cell lysate, we also mixed all ESCC samples into a pool to assess the stability of the MS platform and measured every three days. The quality control standard (including HEK293T cell lysate and ESCC samples pool) was digested and analyzed using the same method and conditions as the ESCC samples. Pearson's correlation coefficient was calculated for all quality control runs using the R statistical analysis software v.3.5.1 (Fig. EV1G). The average Pearson's correlation coefficient among the standards was 0.956 (HEK293T cell lysate) and 0.993 (ESCC samples pool), respectively. The $\log_{10}$-transformed FOTs proteome and phosphoproteome for 53 samples (Fig. EV1I) were plotted to show the consistency of data quality.

## Batch effect evaluation

The batch effect was evaluated for proteome and phosphoproteomic data using principal component regression analysis. Specifically, we set the two evaluated metrics to evaluate the effects during MS detection as (1). Count the number of the significant correlated principal components; (2). Calculated the batch-related information (BRinfo) as following formula (Bai et al, 2025; Lyu et al, 2024):

$$BRinfo = \sum_{i=1}^{n} abs(rho_i) \times eigenvalue_i$$

For which the $rho_i$ is the Spearman correlation between the principal component and the potential factor, $n$ is the count of the principal components (PC). The BRinfo represents the fraction of the proteomic/phosphoproteomic data information that correlated with the specific factor. In addition, we performed the permutation test to evaluate the significance of the BRinfo metric.

## Consensus clustering analysis

The protein expression matrix of the ESCC immunotherapy cohort (53 treatment-naïve ESCC samples) was used to identify the proteomic subtypes using the consensus clustering method implemented in the R package ConsensusClusterPlus v.3.8 (Wilkerson and Hayes, 2010). The ConsensusClusterPlus is an efficient machine learning approach for the identification of distinct molecular patterns and molecular classification and has been applied in many high-throughput biological experiments (Li et al, 2017; Tao et al, 2020). All FOTs less than $10^{-5}$ were replaced with $10^{-5}$ (Ge et al, 2018). Prior to the consensus clustering analysis, we performed a scale normalization to facilitate the interpretation of the expression data. Then, the top 1000 proteins with the highest median absolute deviation were subjected to ConsensusClusterPlus in R v.3.5.1 for unsupervised consensus clustering. The cluster analysis was performed using pam, with the following setting: maxK = 10, reps = 10000, pItem = 0.8, pFeature = 0.8, clusterAlg = "pam", distance = "minkowski" for the clustering runs. A preferred cluster result was selected according to the two combined criteria: (i) considering the profiles of the consensus cumulative distribution function (CDF) and delta area under the CDF curve for clustering solutions between 2 and 10 clusters; (ii) calculating the average silhouette width score to assess the fit of individual patients in the classification, as well as the quality of clusters; (iii) considering the association between classification and immunotherapy response. As shown in Figs. 2A and EV2A–F, the rank survey profiles of the consensus CDF and the delta area under the CDF curve, along with the consensus membership heat maps, indicated three subtypes of solution for 53 cases of ESCC using the proteomic data. The average silhouette width was calculated using the R package Cancersubtype v.3.16.0. The average silhouette width for $K = 3$ was larger than for other subtypes (Fig. EV2E,F). As considering the association of the consensus clusters ($K = 2–10$) with immunotherapy response, we found that only three clusters ($K = 3$) showed significant association with immunotherapy response, which further indicated the three clusters ($K = 3$) were the optimal choice (Fig. 2A). Thus, the ESCC patients were finally clustered into three molecular subtypes G-I-G-III and showed significant association with the immunotherapy response. For the phosphoproteomic subtypes identification, the top 1000 phosphoproteins with the highest median absolute deviation were also subjected to ConsensusClusterPlus in R v.3.5.1 for unsupervised consensus clustering. According to the consensus cumulative distribution function (CDF), delta area under the CDF curve, and average silhouette width (Fig. EV5A–E), the four subtypes were chosen as the phosphoproteomic subtyping, named G-I-G-IV.

## Non-negative matrix factorization (NMF) clustering

Non-negative matrix factorization (NMF), an alternative unsupervised clustering algorithm, was used to evaluate the reliability and robustness of the consensus subtypes (G-I, G-II, and G-III) in our proteomic data. The NMF clustering for proteomic data was performed using NMF R package v.0.26 (Gaujoux and Seoighe, 2010), as previously described for the proteomic clustering (Krug et al, 2020; Lehtio et al, 2021; Ramberger et al, 2024). The following parameters were used: $K = 2:10$, method = "brunet", nrun = 100. The cophenetic correlation coefficient, maximum average silhouette width score, and the association of NMF cluster and immunotherapy response were used to determine the optimal factorization rank (K) for NMF analysis. First, the cophenetic coefficient plot at rank $K = 3$ indicated substantial stability (Fig. EV2G). Second, the average silhouette width score was calculated to assess the fit of patients in the classification, as well as the quality of the NMF clusters. The result showed that the three clusters ($K = 3$) had the maximum average silhouette width score, demonstrating that the number of NMF clusters of three was the optimal classification (Fig. EV2H). After determining the optimal clusters of NMF, the 3 NMF clusters were designated as (NMF-I ($N = 18$), NMF-II ($N = 15$), NMF-III (N = 20)). As for the association of three NMF clusters and ESCC immunotherapy response, the three NMF clusters were also significantly associated with ESCC immunotherapy response (Fisher's exact test, $P = 0.037$), while other NMF clusters showed no statistical significance in association with immunotherapy response (Fig. EV2I,J), which further demonstrated the robustness of the classification and clinical significance. Specifically, the immunotherapy response exhibited a gradual non-sensitive phenomenon from NMF-I to NMF-III, as the percentage of sensitive patients dramatically decreased from 66.7% in NMF-I to 25.0% in NMF-III. Conversely, the percentage of non-sensitive patients increased from 33.3% in NMF-I to 75.0% in NMF-III, showing a similar tendency to the consensus clusters (G-I, G-II, and G-III). It was found that 83% (15 out of 18) of the NMF-I cluster corresponded to the G-I cluster, 87% (13 out of 15) of the NMF-II cluster corresponded to the G-II cluster, and 95% (19 out of 20) of the NMF-III cluster corresponded to the G-III cluster (Fig. EV2K,L).

## The association between proteomic/phosphoproteomic subtypes and clinical features

For the purpose of measuring correlations between proteomic subtypes and clinical features, Fisher's exact test was performed on categorical variables, including gender, age group, smoking, drinking, PD-L1 expression, and immunotherapy response. $P$ values less than 0.05 were considered as significantly different.

## Subtype-specific expressed proteins analysis

To generate the abundance heatmap, we rearranged the ESCC samples within each subtype from G-I to G-III, utilizing the

signature protein abundance matrix associated with the signature pathways for each subtype (Dataset EV3). The signature proteins of each subtype defined here should meet the following criteria: (1) detected in at least 10% of patients in each subtype, (2) differentially expressed of one subtype compared with other subtypes with a fold change >2 and $P < 0.05$ (two-sided Wilcoxon rank-sum test). ConsensusPathDB (CPDB) molecular interaction data were obtained from 32 different public repositories (Herwig et al, 2016), and the dominant bioprocesses of each subtype were determined. Similar criteria were also applied to phosphoproteins and phosphorylation sites in phosphoproteomic subtypes for the differential analysis.

## Validation of the proteomic subtyping in another independent cohort

The validation of proteomic subtypes identified in our cohort using another ESCC cohort reported by Liu et al (Liu et al, 2021a). The nearest template prediction (NTP) algorithm implanted in CMScaller R package (v2.0.1) was used to determine the subtypes (G-I, G-II, and G-III) of the other ESCC cohort based on the overrepresented proteins in the subtypes of our cohort (Tang et al, 2024; Yang et al, 2024).

## Immune cell type composition

The abundance of 64 different cell types was computed via xCell based on proteomic profiles (Aran et al, 2017). The Dataset EV3 contains the final score computed by xCell for different cell types.

## The single sample gene set enrichment analysis (ssGSEA) analysis

Single-sample gene set enrichment analysis (ssGSEA) was utilized to obtain pathway scores for each sample based on proteomic data using the R package GSVA (Hanzelmann et al, 2013). The gene set (c2.all.v7.4.symbols) of the Molecular Signature Database (MSigDB) was used for ssGSEA. The mitochondrial-specific function gene set, human MitoCarta v3.0 (https://www.broadinstitute.org/mitocarta), was used to calculate the pathway score associated with mitochondrial function.

## Gene set enrichment analysis (GSEA)

GSEA was performed by the GSEA software (http://software.broadinstitute.org/gsea/index.jsp). The gene set (c2.all.v7.4.symbols), including KEGG, GO Biological Process (BP), Reactome, and HALLMARK, was downloaded from the Molecular Signatures Database (MSigDB v7.1, http://software.broadinstitute.org/gsea/msigdb/index.jsp). Version 3.0 of the human MitoCarta was retrieved from https://www.broadinstitute.org/mitocarta.

## Construction and validation of predictive models for ESCC immunotherapy response

The extreme gradient boosting (XGBoost) algorithm was used to construct the therapeutic response prediction model based on the significantly differentially expressed proteins between S and NS

groups in Python software (v3.9.2) with the scikit-learn (v1.1.3) package. The 53 ESCC patients in the discovery cohort were randomly divided into 70% of individuals (the training set) and the remaining 30% (the testing set). First, we calculated the AUC values of 124 proteins associated with YAP1, mitochondrial complex I, and antigen processing and presentation in immunotherapy prediction with the roc function in the pROC R package and selected the top ten proteins based on their AUC values as candidate biomarkers for the predictive model (Robin et al, 2011; Weller et al, 2025). Second, we used the backward stepwise method to select a subset of proteins from these top ten candidates for model construction (Demir et al, 2025; Fujii et al, 2021; Li et al, 2023; Yoneshiro et al, 2025). Then, we trained XGBoost classifiers based on the four signatures (YAP1, NDUFB7, MHC-I, and CD8A) for hyperparameter tuning by the GridSearchCV function with 3-fold cross-validation to optimize its performance. The adjusted parameters are as follows: max_depth = 5, learn_rate = 0.01, n_estimators = 50, objective = 'binary: logistic' and min_child_weight = 1. Moreover, the diagnostic ability of the model was validated in an independent validation cohort (55 patients) and an IHC validation cohort (82 patients) (Dataset EV7).

## Cell lines

Human esophageal squamous cell carcinoma cell lines EC1 and ECA109 were obtained from the Shanghai Institutes for Biological Science (Shanghai, China) (Cao et al, 2020). Murine ESCC cell line AKR was purchased from Otwo Biotech (catalog No.: HTX2545) and cultured in PRMI 1640 medium (GIBCO) supplemented with 10% FBS (GIBCO), 100U/ml penicillin, and 100U/ml streptomycin in 5% $CO_2$ at 37 °C. All cell lines were tested negative for mycoplasma contamination.

## Platelet isolation and activation

Blood samples were collected from ESCC patients, and platelets were isolated utilizing established methods (Li et al, 2024; Rachidi et al, 2017). Human blood was withdrawn into tubes containing acid citrate dextrose—a buffer. Platelet-rich plasma (PRP) was obtained by centrifugation at 200 g for 15 min at room temperature. Afterward, PRP was centrifuged at 640 g for 5 min to pellet the platelets, which were then washed twice and resuspended in modified Tyrode-HEPES buffer for subsequent use. Subsequently, platelets were activated with thrombin (1 IU/mL) for 45 min at 37 °C (125 rpm), and the activated platelets were collected for MS-based proteome profiling to explore the molecular perturbation after platelet activation. The supernatant of activated platelets was added to the co-cultures of CD8 + T cells and ESCC cells (EC1 and ECA109).

## Plasmid construction

To explore the function of YAP1 in ESCC cells, the constitutively active mutant YAP1 (2SA) was used. The full-length human YAP cDNA was PCR-amplified and ligated to XbaI restriction sites of the expression vectors used. The YAP1 (2SA) mutants in which serine 127 and 381 were mutated to alanine were constructed by site-directed mutagenesis or deletion by PCR in accordance with a previous study (Miyamura et al, 2017).

For constructing YAP1-knockdown ESCC cells, the pLKO.1-puro vectors carrying different YAP1 shRNA constructs were custom-designed by using online software available from Sigma-Aldrich. YAP1 shRNA sequences including shRNA1 (5'-CCCAGT-TAAATGTTCACCAAT-3'), shRNA2 (5'-GCCACCAAGCTAGA-TAAAGAA-3'), and shRNA3 (5'-CTCAGGAATTGAGAACAA-TGA-3'). A nonsense sequence shRNA (5'-CAACAAGATGAA-GAGCACCAA-3') was used as a negative control.

For the analysis of the function of NDUFB7 in ESCC cells, we constructed stable cell lines overexpressing NDUFB7-FLAG. The cDNA of NDUFB7 was cloned into the pCDH-CMV-EF1-Puro vector via the unique XbaI site and the neighboring BamHI site. For convenient detection, a FLAG-tag encoding sequence (GATTA-CAAGGATGACGACGATAAG) was inserted before the stop codon (TAG) to express the NDUFB7-FLAG fusion protein.

The primers used for plasmid construction are as follows:

Forward primer: 5'-acctccatagaagattctagaATGGGGGCGCACC-TGGTC-3', Reverse primer: 5'-atccttcgcggccgcggatccTTAAACCT-TATCGTCGTCATCCTTG-3'.

## Lentivirus production and cell transduction

For NDUFB7, the double-stranded DNA was cloned into the pCDH-CMV-EF1-Puro vector; 8 µg of packaging plasmids pMD2.G: psPAX2 (1:3) and 8 µg of Lentivector containing the target gene were co-transfected into $2.5 \times 10^6$ HEK-293T cells using Lipofectamine 2000 transfection reagent (Invitrogen, catalog No.: 11668-019). The media containing the lentivirus particles were collected after 24 h and 48 h, separately, and centrifuged at $1500 \times g$ for 10 min. These 24-h and 48-h supernatants were used independently to infect EC1 and ECA109 cells in the presence of 10 µg/ml hexadimethrine bromide (Sigma, catalog No.: H9268) for 12 h. After infection, cells were cultured and selected with puromycin for the generation of stably overexpressed cells. The empty pCDH vector was used as a negative control. For YAP1 (2SA), pLVX-YAP1 (2SA)-PURO vector was then transfected EC1 and ECA109 cells, and the empty pLVX vector was used as a negative control.

## Proteome profiling and differential analysis of ESCC cell lines

EC1 and ECA109 cells were divided into two groups according to the following experimental conditions: scramble vector (Vector OE) and NDUFB7-FLAG overexpression (NDUFB7 OE). Each group contained at least three biological replicates. The protein concentration was determined using the Bradford assay. Cells were boiled in a 99 °C metal bath for 30 min with 100 µL of 50 mM ABC buffer (ammonium bicarbonate) containing SDS at a final concentration of 4%. Protein samples underwent trypsin digestion (enzyme-to-substrate ratio of 1:50 at 37 °C for 18–20 h). All peptide samples were desalinated using a C18 column (50% acetonitrile and 0.1% formic acid) and then analyzed using a Q Exactive HF-X Hybrid Quadrupole-Orbitrap Mass Spectrometer (Thermo Fisher Scientific, Rockford, IL, USA) coupled with a high-performance liquid chromatography system (EASY nLC 1200, Thermo Fisher). MS raw files generated by LC-MS/MS were searched against the NCBI human Refseq protein database (released on 04-07-2013; 32,015 entries) using MaxQuant (version 1.6.2.10) software enabled

with Andromeda search engine. The protease was Trypsin, and up to 2 missed cleavages were allowed. Carbamidomethyl (C) was considered a fixed modification. For the proteome profiling data, variable modifications were oxidation (M) and acetylation (Protein N-term). We screened the differentially expressed proteins in NDUFB7-overexpressing and control EC1 and ECA109 ESCC cells (FC (NDUFB7-OE/Vector) > 1.5 or <0.67, $P < 0.05$ (log10-transformed FOT, two-sided Student's $t$ test) (Dataset EV6). Pathway enrichment analysis was performed according to the Consensus-PathDB (CPDB) database.

## Western blotting

For western blotting, cells were lysed with 0.5% NP-40 buffer containing 20 mM Tris-HCl (pH 8.0), 100 mM NaCl, 1 mM EDTA, 1 mM PMSF, and Nonidet P-40. The protein concentration was quantified using the Bradford assay. For each sample, 30 µg of protein extract was separated using 10% sodium dodecyl sulfate-polyacrylamide gel electrophoresis and transferred to nitrocellulose membranes. After blocking with 5% milk (BD Science) solution in TBST (Tris-buffered saline with Tween) for 1–2 h, the membranes were incubated with TBST containing the appropriate primary antibodies overnight at 4 °C, followed by a 2 h incubation with horseradish peroxidase-conjugated secondary antibodies. The target protein bands were detected using the Chemiluminescent detection reagent. The mouse anti-GAPDH antibody (Proteintech, catalog No.: 60004-1-Ig), the rabbit anti-NDUBF7 (Abcam, catalog No.: ab188575), and the mouse anti-flag monoclonal antibody (Sigma, catalog No.: M20008L) were used, and their specificity was confirmed by western blotting. Western blot quantification was performed using ImageJ software (Version 1.52a, National Institutes of Health, MD, USA).

## Blue-native (BN)-PAGE

ESCC cells were collected by trypsinization and washed twice with ice-cold PBS. Mitochondria were isolated using the Animal Cell Mitochondrial Isolation and Electrophoresis Sample Preparation Kit (Real-times, catalog No.: RTD8117) following the manufacturer's instructions. Mitochondrial protein concentration was determined with the BCA Protein Assay Kit (Beyotime, catalog No.: P0012) with a BSA standard curve. Mitochondrial pellets were resuspended in BN resuspension buffer at a protein concentration of 1 mg/mL and solubilized with 1% digitonin. A total of 20 µg of mitochondrial proteins were separated using the Blue/Clear Native Page Electro-phoresis Kit (Real-times, catalog No.: RTD6139-0312) at 4 °C with cathode and anode buffers. After fractionation, gels were electro-blotted onto PVDF membrane for 5 h at 4 °C and processed for immunoblot analysis with anti-NDUFB7 antibody (Abcam, catalog No.: ab188575, 1:1000).

## Flow cytometry analysis

For the analysis of cellular MHC-I expression, cells were scraped upon reaching up to 90% confluence. Cell pellets were then washed and incubated with HLA antibodies (Abcam, catalog No.: ab225636) at a 1:10 ratio in FACS buffer containing 1% FBS and 0.1% sodium azide in PBS. Measurements were performed using the Gallios flow cytometer (BeckmanCoulter).

Three biological replicates were analyzed, each with technical triplicates.

For the analysis of intratumoral T cells, fresh treatment-naïve ESCC samples with different NDUFB7 intrinsic expression levels were used. Specifically, single cells were prepared from tumors using Tumor Dissociation Kits (Miltenyi Biotec, catalog No.: 130-095-929) following the manufacturer's instructions. Digested tumors were filtered through a 70 μm cell strainer and centrifuged at 500 g for 5 min to obtain single cells. The single-cell suspension was subjected to flow cytometry based on different cell markers. To analyze surface markers, cells were stained with antibodies (CD8, PD1) at 4 °C for 30 min. For intracellular cytokine staining, cells were fixed, permeabilized, and labeled with intracellular cytokine antibodies (GZMK). Data were acquired using the Verse system (BD) and analyzed with FlowJo (v.10.8.1).

## Direct ESCC cell killing by activated CD8 + T cells

Human peripheral blood was collected from healthy volunteer donors, heparin anticoagulant, and diluted with PBS at a 1:1 ratio. PBMCs were isolated from human peripheral blood and counted. CD8 + T cells were purified from PBMC using the MojoSort™ Human CD8 T Cell Isolation Kit (Biolegend, catalog No.: 480129), according to the manufacturer's instructions. Isolated CD8 + T cells were activated by the anti-human CD3 antibody (BioXCell, catalog No.: CDE-M120a) and anti-human CD28 antibody (BioXCell, catalog No.: BE0248) at a 1:1 ratio, cultured in complete culture assay medium (DMEM-GlutaMAX™-I, 10% FBS) supplemented with 1% penicillin–streptomycin and 100 U/ml human IL-2 (PeproTech, catalog No.: 200-02) and maintained at a density of $1 \times 10^6$ cells/ml. The activated CD8 + T cells were co-cultured with ESCC cell lines including EC1-vector-OE, EC1-NDUFB7-OE, ECA109-vector-OE, and ECA109-NDUFB7-OE in a 96-well plate at 20:1 ratio with or without treatment with anti-PD1 mAb (BioXCell, catalog No.: BE0188), respectively. After 3 days of co-culture, a CCK-8 assay was performed to analyze the effect of overexpression of NDUFB7 in ESCC cells on the killing of tumor cells by CD8 + T cells.

## ESCC cell killing by CD8 + T cells activated by mature dendritic cells (mDCs) induced maturation by ESCC cell lines

Human peripheral blood was collected from healthy volunteer donors, heparin anticoagulant, and diluted with PBS at a 1:1 ratio. PBMCs were isolated from human peripheral blood and counted. Cells ($2 \times 10^6$ cells/ml) were cultured (5% $CO_2$, 37 °C) in 3 ml fresh RPMI 1640 medium containing 10% FBS in six-well plates for 4 h. Unadherent cells (called "PBL", frozen at −80 °C for reserve) were collected and washed twice with serum-free RPMI 1640 culture solution. Adherent cells were used to induce DC cells. To generate immature DCs (iDCs), subsequently, human recombinant granulocyte-macrophage colony-stimulating factor (hrGM-CSF) (PeproTech, catalog No.: 300-03-100UG) (100 ng/ml) and human recombinant Interleukin-4 (hrIL-4) (PeproTech, catalog No.: 200-04-50UG) (400 ng/ml) were added and the culture dish was incubated for 6 days in a humidified 5% CO2 incubator at 37 °C with media change after 1 day and a half. Following 6 days of incubation, iDCs were collected and co-cultured with ESCC cancer

cells including EC1-vector-OE, EC1-NDUFB7-OE, ECA109-vector-OE, and ECA109-NDUFB7-OE with 25 ng/ml hTNF-α (Pepro-Tech, catalog No.: 300-01A-50UG) in media for 8 days. After 8 days of co-culture, CD8 + T cells purified from PBMC using the MojoSort™ Human CD8 T Cell Isolation Kit (Biolegend, catalog No.: 480129), and mDCs induced by ESCC cancer cells were co-cultured with CD8 + T cells for 3 days. After 3 days of co-culture, activated CD8 + T cells were collected as the effector cells, and the corresponding ESCC cells, including EC1-vector-OE, EC1-NDUFB7-OE, ECA109-vector-OE, and ECA109-NDUFB7-OE were as target cells. The effector cells were co-cultured for 3 days, and a CCK-8 assay was performed to analyze the effect of overexpression of NDUFB7 in ESCC cells on the killing of tumor cells by different effector cells (CD8 + T cells).

## ELISA assay

TNF-α and IFN-γ concentrations in cell culture supernatants were measured using commercial ELISA kits (TNF-α: StemCell, catalog No.: 100-1142; and IFN-γ: StemCell, catalog No.: 02002) according to the instructions of the manufacturer. At 450 nm, the OD values of each sample were measured to indicate the concentration of cytokines.

## ESCC organoid culture

Fresh tumor tissues were obtained by surgical resection. Tumor tissues were processed for organoid establishment within 24 h. Tumor tissues derived from surgical resections were cut into small pieces and enzymatically digested using 1 mg/ml collagenase II (VETEC, catalog No.: V900892), 0.5 mg/ml hyaluronidase (Sigma, catalog No.: H3506), 1 mg/ml collagenase I (VETEC, catalog No.: V900891), 125 μg/ml dispase (Sigma, catalog No.: D4693), and 0.2 μg/ml DNase I (Merck, catalog No.: 10104159001), we then filtered the dissociated tissue through a 70-μm cell strainer to remove large undigested fragments and centrifuged the cells at $300 \times g$ for 5 min before resuspending the cells in Gelmatrix (Corning, catalog No.: 356231). The cells were resuspended in PBS, and centrifugation was repeated. This procedure was repeated twice to remove debris and collagenase. The isolated cells were resuspended in Gelmatrix. After Gelmatrix solidification for 10 min at 37 °C, cells were overlaid with human ESCC organoid medium. Human ESCC organoids medium was composed of basal medium (advanced DMEM/F12 (Gibco, catalog No.: 12634010) supplemented with 1× Glutamax (Gibco, catalog No.: 35050061), 10 mM HEPES (Gibco, catalog No.: 15630080), 100/100 U/ml Penicillin/Streptomycin (Gibco, catalog No.: 15140122), 5 μg/ml Gentamicin (Corning, catalog No.: 30-005-CR), 1× N2 Supplement (Gibco, catalog No.: 17502048), 1× B27 Supplement (Gibco, catalog No.: 17504044), 1 mM N-acetylcysteine (NAC) (Sigma, catalog No.: A9165 - 5 G), 50 ng/ml human epidermal growth factor (EGF) (PeproTech, catalog No.: AF-100-15), 10 μM Y-27632 (PeproTech, catalog No.: 1293823), 100 ng/ml Noggin (PeproTech, catalog No.: 120-10 C), and 50 ng/ml R-spondin (PeproTech, catalog No.: 120-38).

Organoid culture medium was refreshed every 2 days. To pass the organoids, Gelmatrix was disassociated by pipetting. The organoids were collected in a Falcon tube, and TrypLE (Gibco, catalog No.: 12604021) was added before being incubated at 37 °C

for approximately 5 min. A vigorous manual shake would ensue before the suspension was centrifuged at 300-400×g for 2 min. The remaining cell pellet was resuspended in Gelmatrix. After allowing the Gelmatrix to polymerize, complete media was added and incubated at 37 °C.

## Organoid-CD8 + T cell co-culture

We used the widely adopted methods for organoid-CD8 + T cell co-culture (Cattaneo et al, 2020; Chalabi et al, 2020; Dijkstra et al, 2018; Zhou et al, 2021). Similarly, Chalabi et al employed the same co-culture method to investigate the correlation between mismatch repair (MMR) and immunotherapy response in colon cancer through artificial interventions (Chalabi et al, 2020), such as treating these organoids with IFN-γ to enhance antigen presentation and activating CD8 + T cells by anti-CD3/CD28 antibodies. In detail, the autologous peripheral blood was collected while the tumor tissue for organoid establishment from surgical resections, heparin anticoagulant, and diluted with PBS at a 1:1 ratio. PBMCs were isolated from human peripheral blood by Ficoll density gradient centrifugation and counted. CD8 + T cells were purified from PBMC using the MojoSort™ Human CD8 T Cell Isolation Kit (Biolegend, catalog No.: 480129), according to the manufacturer's instructions. CD8 + T cells were then expanded and cryopreserved until later use. Culture media for CD8 + T cells was composed of RPMI 1640 medium, 50 ng/ml human IL-2 (PeproTech, catalog No.: 200-02), 0.5 μg/ml anti-human CD3 antibody (BioXCell, catalog No.: CDE-M120a), and 2 μg/ml anti-human CD28 antibody (BioXCell, catalog No.: BE0248).

One day before co-culture, CD8 + T cells were thawed in the pre-warmed T cell medium and cultured overnight at 37 °C for pre-activation. Prior to co-culture, tumor organoids were stimulated overnight with 200 ng/ml human recombinant IFN-γ (PeproTech, catalog No.: 300-02). Transfer the suspension of organoids and medium into a 15-ml tube. Split the total volume into two aliquots: one of the total volumes will be used for dissociation into single cells and counting to determine the effector: target cell ratio, and the rest of the total volume will be used as fully formed organoids for the experiment, as described previously (Cattaneo et al, 2020). For the co-culture, organoids and CD8 + T cells were resuspended in medium in a flat-bottom 96-well plate at a 20:1 effector: target cell ratio. Bright-field imaging was performed using the Leica DMi8 inverted fluorescence microscope during culture each day. After 3 days of co-culture, CCK-8 assay was performed to analyze the cell death of tumor cells killed by CD8 + T cells.

## Syngeneic mouse ESCC models and in vivo treatments

For the syngeneic murine ESCC model and allograft experiments, 6-week-old male immunocompetent C57BL/6 J mice were obtained from SLAC (Shanghai). Constructed stable mouse ESCC cell line, AKR (Ndufb7 OE or vector control AKR cells), was subcutaneously injected into the right flank of mice ($3×10^6$ cells for each injection). On day 7, mice bearing tumors of similar size were randomly divided into four groups (5 mice per group) in the vector control and Ndufb7 OE ESCC tumors before treatment with anti-PD1 antibody, isotype IgG antibody, YAP1 inhibitor (CA3), and the combination of anti-PD1 antibody and CA3, respectively. The anti-PD1 antibody (BioXCell, catalog No.: BE0146, 100 μg/injection/

mouse, twice a week), isotype IgG antibody (BioXCell, catalog No.: BE0089, 100 μg/injection/mouse, twice a week), YAP1 inhibitor (CA3) (SELLECK, catalog No.: S8661, 1 mg/kg/mouse, three times per week), and the combination of anti-PD1 antibody with CA3 were administered by intraperitoneal injection, respectively. Tumor growth was monitored over 3 weeks, and tumor size was measured every 3 days. Tumor volumes were calculated by the formula: Length×Width×Width×0.5. Mice were euthanized with isoflurane (RWD), followed by cervical dislocation at the end of the experiment (22 days after cell injection). At the end of the allograft experiments, tumors were harvested, and paraffin-embedded tumor blocks were prepared for immunohistochemistry (IHC) staining. This study is under the guidelines of the Institutional Animal Care and Use Committee (IACUC), Fudan University. All procedures were approved by IACUC, Fudan University.

## Immunohistochemistry staining and evaluation

For FFPE tumor blocks obtained from human ESCC patients, a standard immunohistochemistry (IHC) protocol was followed to stain the tumor tissue samples using the anti-human NDUFB7 (Abcam, catalog No.: ab188575), the anti-human/mouse YAP1 (Cell Signaling Technology, catalog No.: 14074), anti-human CD8 (Cell Signaling Technology, catalog No.: 70306) and the anti-human HLA class I ABC (Abcam, catalog No.: ab225636) antibodies. For the IHC analysis in the IHC validation cohort, protein abundance was quantified according to the adopted scoring system (on a scale of 0-12). In detail, the IHC staining results were evaluated independently by two pathologists who were blinded to the clinicopathologic data. According to the proportion of positive cells, samples were scored as follows: 0 + , none; 1 + , <25%; 2 + , 25–50%; 3 + , 50–75%; and 4 + , 75–100%. The staining intensity was evaluated as follows: 0, none; 1, weak; 2, medium; and 3, strong. The final score (range 0–12) was calculated by multiplying the two sub-scores.

For the tumor blocks from the murine tumor, the tumor tissues were stained by the anti-mouse CD8 (Abcam, catalog No.: ab209775), the anti-mouse CD4 (Cell Signaling Technology, catalog No.: 25229), the anti-mouse PD1 (Abcam, catalog No.: ab214421), the anti-mouse TNF-α (Abcam, catalog No.: ab307164), the anti-mouse PD-L1 (Cell Signaling Technology, catalog No.: 64988), the anti-human/mouse YAP1 (Cell Signaling Technology, catalog No.: 14074), and the anti-mouse NDUFB7 (Proteintech, catalog No.: 68362-1-Ig) antibodies. IHC evaluation was analyzed using an IHC profiler-compatible plugin in ImageJ software with integrated options for the quantitative analysis of digital IHC images stained for cytoplasmic or nuclear proteins (Varghese et al, 2014). Moreover, the intensity of the cytoplasmic staining and the percentage of positively stained tumor cells were also scored numerically.

## Multiplex immunofluorescence staining

The formalin-fixed paraffin-embedded tissue sections (4 μm) from ESCC treatment-naïve patients were collected, and immunofluorescence staining was done according to the previously described method (Park et al, 2023). Briefly, Tumor FFPE slides were baked at 50 °C overnight, deparaffinized in xylene, and rehydrated in decreasing concentrations of ethanol (100%, 90%, 70%, 50%, and

**The paper explained**

**Problem**

Despite being a first-line therapy, nearly half of esophageal squamous cell carcinoma (ESCC) patients derive no benefit or experience recurrence following immunotherapy, and the predictive value of PD-L1 expression remains limited. Currently, there is still a lack of comprehensive proteomic profiling to explore alternative mechanisms or identify robust biomarkers associated with ESCC immunotherapy response.

**Results**

Using proteomic profiling, this study delineated the molecular landscape of ESCC immunotherapy response, identifying alterations of mitochondrial complex I and the hippo pathway features that are associated with patient response. Furthermore, a predictive biomarker panel was developed to guide immunotherapy patient selection.

**Impact**

This comprehensive proteomic profiling provides molecular insights into the ESCC immunotherapy response and establishes a predictive model with high accuracy for distinguishing sensitive patients from non-sensitive patients. It lays the foundation for developing potential therapeutic strategies and personalized treatment for ESCC patients.

ddH$_2$O). Sample slides were incubated in pH 6 or pH 9 buffers at 95 °C for 30 min for antigen retrieval, then in 3% hydrogen peroxide for 15 min for permeabilization and in serum-free protein block solution for 30 min for blocking non-specific binding. Then, different primary antibodies were sequentially applied, followed by horseradish peroxidase-conjugated secondary antibody incubation and tyramide signal amplification working solution. Multiplex IHC was performed with the same protocols but different primary antibodies for two panels containing antibodies against: Panel 1: PANCK (Dako, catalog No.: GA05361-2), CD45 (Abcam, catalog No.: ab40763), α-SMA (Cell Signaling Technology, catalog No.: 19245) and NDUFB7 (Abcam, catalog No.: ab188575); panel 2: CD8 (Cell Signaling Technology, catalog No.: 70306), TIM3 (Abcam, catalog No.: ab241332), GZMB (Abcam, catalog No.: ab255598) anti-mouse NDUFB7 (Abcam, catalog No.: ab188575). Nucleic acid was stained with DAPI (Invitrogen, catalog No.: P36931). Finally, stained slides were mounted using Prolong Gold Antifade, and slides were imaged with a confocal microscope.

### ESCC scRNA-seq analysis

The single-cell data of ESCC were downloaded from GSE18955 (Pan et al, 2022). The Seurat R package was used to quantify and visualize single-cell RNA data. To score individual cells for pathway activities, we used the R package AUCell. First, for each cell, we used an expression matrix to compute gene expression rankings in each cell with the AUCell_buildRankings function, with default parameters. The canonical pathway database was downloaded from the Molecular Signature Database (https://www.gsea-msigdb.org/gsea/msigdb), and canonical pathway gene sets were then used to score each cell where, for each gene set and cell, area under the curve (AUC) values were computed (AUCell_calcAUC function) based on gene expression rankings, where AUC values represent the fraction of genes within the top-ranking genes for each cell that are defined as part of the pathway gene set.

## Data availability

The MS raw data in this study have been deposited in iProX repository (www.iprox.cn) (Ma et al, 2019) under the accession IDs IPX0006917001 and IPX0006917002 for proteomic data and phosphoproteomic data in the discovery cohort, as well as IPX0006917003 for the validation cohort. Source data are provided with this paper.

The source data of this paper are collected in the following database record: biostudies:S-SCDT-10_1038-S44321-026-00413-9.

## Peer review information

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

## Acknowledgements

This work is supported by the National Key Research and Development Program of China (2022YFA1303200 [CD] and 2022YFA1303201 [CD]), National Natural Science Foundation of China (32330062 [CD], 31972933 [CD], 82372669 [YCH], 32301236 [YL], 82003154 [CX], 32201215 [JWF], 32471498 [YZW], and 32201212 [YZW]), sponsored by Program of Shanghai Academic/Technology Research Leader (22XD1420100 [CD]), the Major Project of Special Development Funds of Zhangjiang National Independent Innovation Demonstration Zone (ZJ2019-ZD-004 [CD]), the Science and Technology Commission of Shanghai Municipality (2023SHZDZX02 [CD]), Shanghai Sailing Program (23YF1402800 [YL]), the Fudan original research personalized support project, and the China Postdoctoral Science Foundation (2024T170177 [YL]). The computations in this research are performed using the CFFF platform of Fudan University. This work is supported by the Shanghai Municipal Science and Technology Major Project, the Human Phenome Data Center of Fudan University, and the Shanghai Phenomic precision measurement professional technical service platform (23DZ2290800).

## Author contributions

**Fahan Ma**: Conceptualization; Data curation; Formal analysis; Validation; Investigation; Visualization; Methodology; Writing—original draft; Writing—review and editing. **Yan Li**: Conceptualization; Data curation; Funding acquisition; Validation; Writing—original draft. **Chan Xiang**: Resources; Funding acquisition. **Bing Wang**: Data curation; Validation. **Jie Lv**: Validation. **Zhanxian Shang**: Resources. **Weiguang Zhang**: Resources. **Zhaoyu Qin**: Methodology. **Yan Pu**: Methodology. **Kai Li**: Methodology. **Jinzhi Wei**: Resources. **Su-Bei Tan**: Methodology. **Jinwen Feng**: Funding acquisition; Methodology. **Haohua Teng**: Methodology. **Peipei Zhang**: Resources. **Jiaying Deng**: Methodology. **Yunzhi Wang**: Funding acquisition; Methodology. **Chao Zhang**: Methodology. **Sha Tian**: Methodology. **Guichao Li**: Methodology. **Mingqiang Kang**: Resources; Supervision. **Changsheng Du**: Supervision; Validation. **Yuchen Han**: Resources; Supervision; Funding acquisition. **Chen Ding**: Conceptualization; Supervision; Funding acquisition; Writing—original draft; Project administration; Writing—review and editing.

Source data underlying figure panels in this paper may have individual authorship assigned. Where available, figure panel/source data authorship is listed in the following database record: biostudies:S-SCDT-10_1038-S44321-026-00413-9.

## Disclosure and competing interests statement

The authors declare no competing interests.

# Expanded View Figures

**Figure EV1. The protein identification and quality control using the mass spectrometry platform.**

(A) The representative images of immunohistochemistry (IHC) staining of PD-L1 expression in sensitive (S) and non-sensitive (NS) patients of immunotherapy. (B, C) The association of PD-L1 (CD274) mRNA expression with cancer immunotherapy response in the GSE78220 melanoma cohort (B) and IMvigor210 metastatic urothelial carcinoma cohort (C). Boxplots showing the PD-L1 expression difference between S and NS groups. Boxplots show median (central line), upper and lower quartiles (box limits), and minimum or maximum (whiskers) (left). The Kaplan–Meier curves of the expression of PD-L1 with overall survival (OS) (right). $P$ value from two-sided log-rank test. GSE78220 melanoma cohort: $n = 15$ (S) and 13 (NS). IMvigor210 metastatic urothelial carcinoma cohort: $n = 68$ (S) and 230 (NS). (D) The Kaplan–Meier plots of S and NS patients of ESCC in OS and PFS in the MSK-IMPACT cohort (two-sided log-rank test). (E) The association of TMB level with OS and PFS in the MSK-IMPACT cohort (two-sided log-rank test). (F) The association of TMB level with OS in the TCGA-ESCC cohort (two-sided log-rank test). (G) Longitudinal quality control of MS using tryptic digests of HEK293T cells (left) and ESCC samples pool (right). The bottom-left half of the panel represents the pairwise Pearson's correlation coefficients of the samples (two-sided Pearson's correlation test), and the top-right half of the panel depicts the pairwise scatter plots from the same comparison. (H) The proteome and phosphoproteome identification between the S and NS groups. Venn diagrams show the overlap of proteins, phosphoproteins, and phosphosites in S and NS groups (upper). Barplots display the identified number of proteins, phosphoproteins, and phosphosites (bottom). $P$ value from two-sided Student's $t$ test. $n = 24$ (S) and 29 (NS). Data are presented as mean with standard error of the mean. (I) Boxplot for $\log_{10}$(FOT) of proteins (upper) and phosphoproteins (bottom) in 53 ESCC patients. Boxplots show median (central line), upper and lower quartiles (box limits), and 1.5 × interquartile range (whiskers). (J) Principal components analysis (PCA) of proteins (left) and phosphoproteins (right) levels in 53 ESCC patients. (K) The batch effects evaluation of the proteomic (left) and phosphoproteomic (right) data during MS detection. The $x$ axis showed the principal components. The right $y$ axis (for the scatter plot) indicates the explained ratio of principal components. The left $y$ axis (for the barplot) was the Spearman correlation of batch effects variable and each principal component.

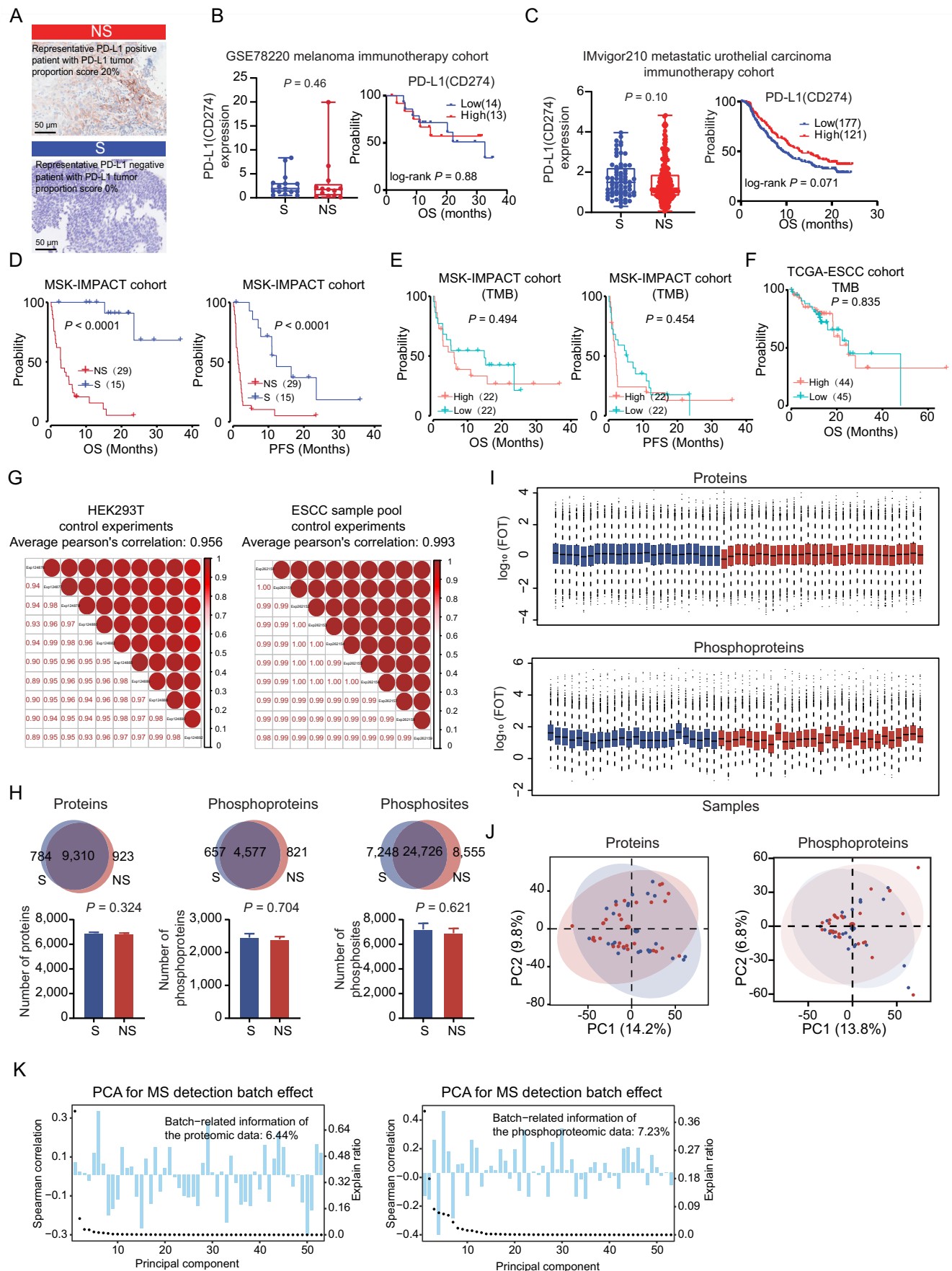

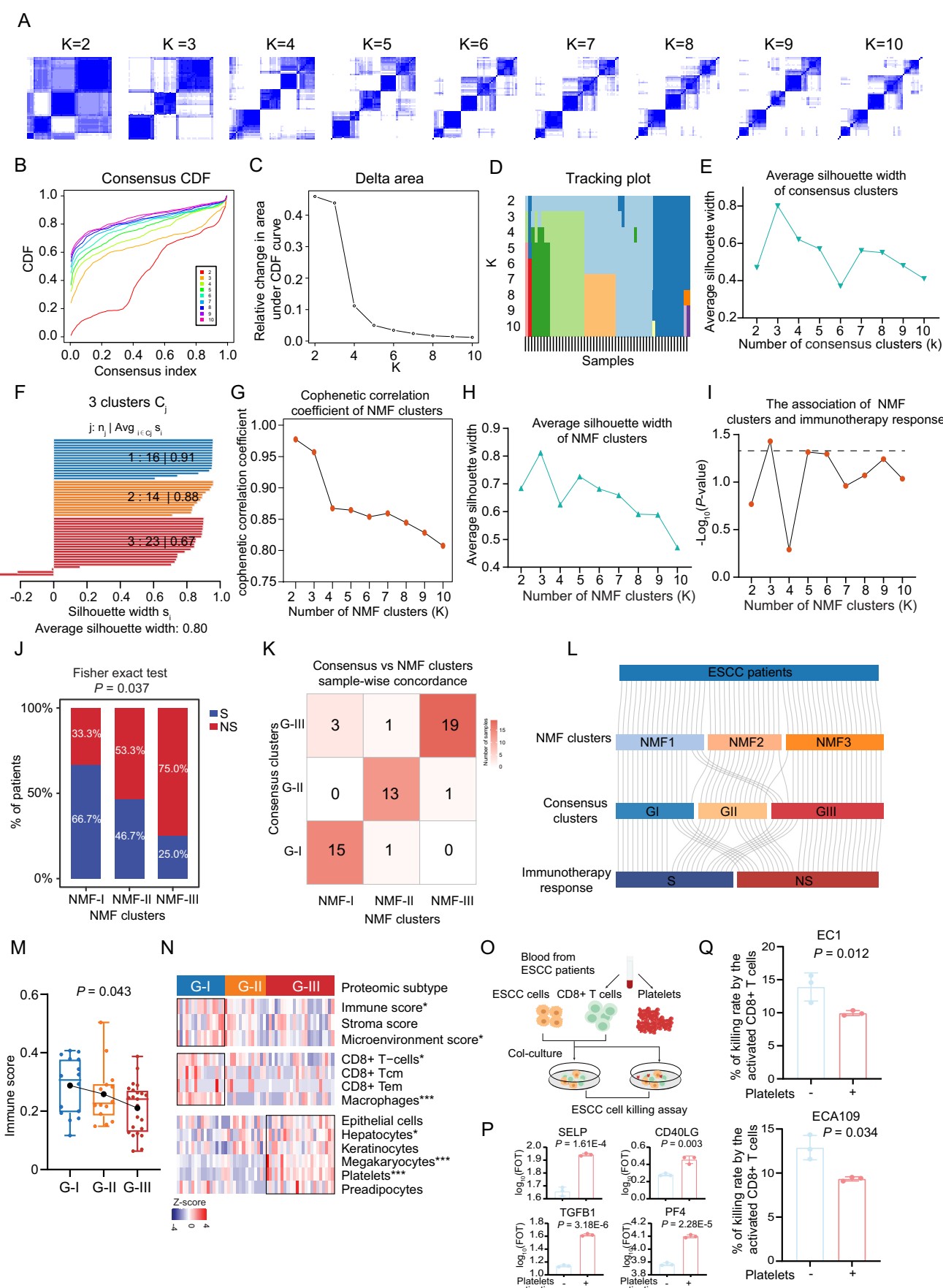

◄ **Figure EV2.  Consensus clustering analysis of the ESCC immunotherapy cohort identified three proteomic subtypes.**

(A–D) The consensus clustering analysis of 53 ESCC samples and three subtypes was generated. $K$ was tested from 2 to 10 (A). Consensus matrices, as well as the consensus cumulative distribution function (CDF) plot (B), delta area (change in CDF area) plot (C), and tracking plot (D) are shown. (E) Average silhouette-width of the consensus subtypes ($K = 2$–10). The average silhouette-width takes the maximum value when the number of subtypes is 3 ($K = 3$). (F) The silhouette-width plot of three clusters. (G) Cophenetic correlation coefficient for the different choices of clusters in the non-negative matrix factorization (NMF) clustering. (H) The average silhouette-width score in different choices of clusters in NMF clustering. (I) The association of NMF clusters and ESCC immunotherapy response ($P$ value from two-sided Fisher's exact test). (J) Barplot shows the distribution of immunotherapy response (S/NS) at the optimal choice of NMF clusters ($K = 3$). $P$ value from two-sided Fisher's exact test. (K) Comparison of sample overlap in subtype assignment between consensus clusters and NMF clusters. (L) Sankey diagram indicates the comparison of NMF clusters and consensus clusters. (M) Boxplot for immune score among three proteomic subtypes (ANOVA test). Boxplots show median (central line), upper and lower quartiles (box limits), and minimum and maximum (whiskers). $n = 16$ (G-I), 14 (G-II), and 23 (G-IV). (N) Heatmap illustrating the dominant cell type compositions of G-I and G-III subtypes (two-sided Wilcoxon rank-sum test). *$P < 0.05$, **$P < 0.01$, ***$P < 0.001$, ****$P < 0.0001$. (O) Schematic illustrating the experimental design. (P) Barplots showing the molecular perturbation related to activated platelets ($n = 3$ independent experiments, two-sided Student's $t$ test, mean with standard deviation). (Q) The influence of activated platelets on CD8 + T cell-mediated killing against ESCC cells ($n = 3$ independent experiments, two-sided Student's $t$ test, mean with standard deviation).

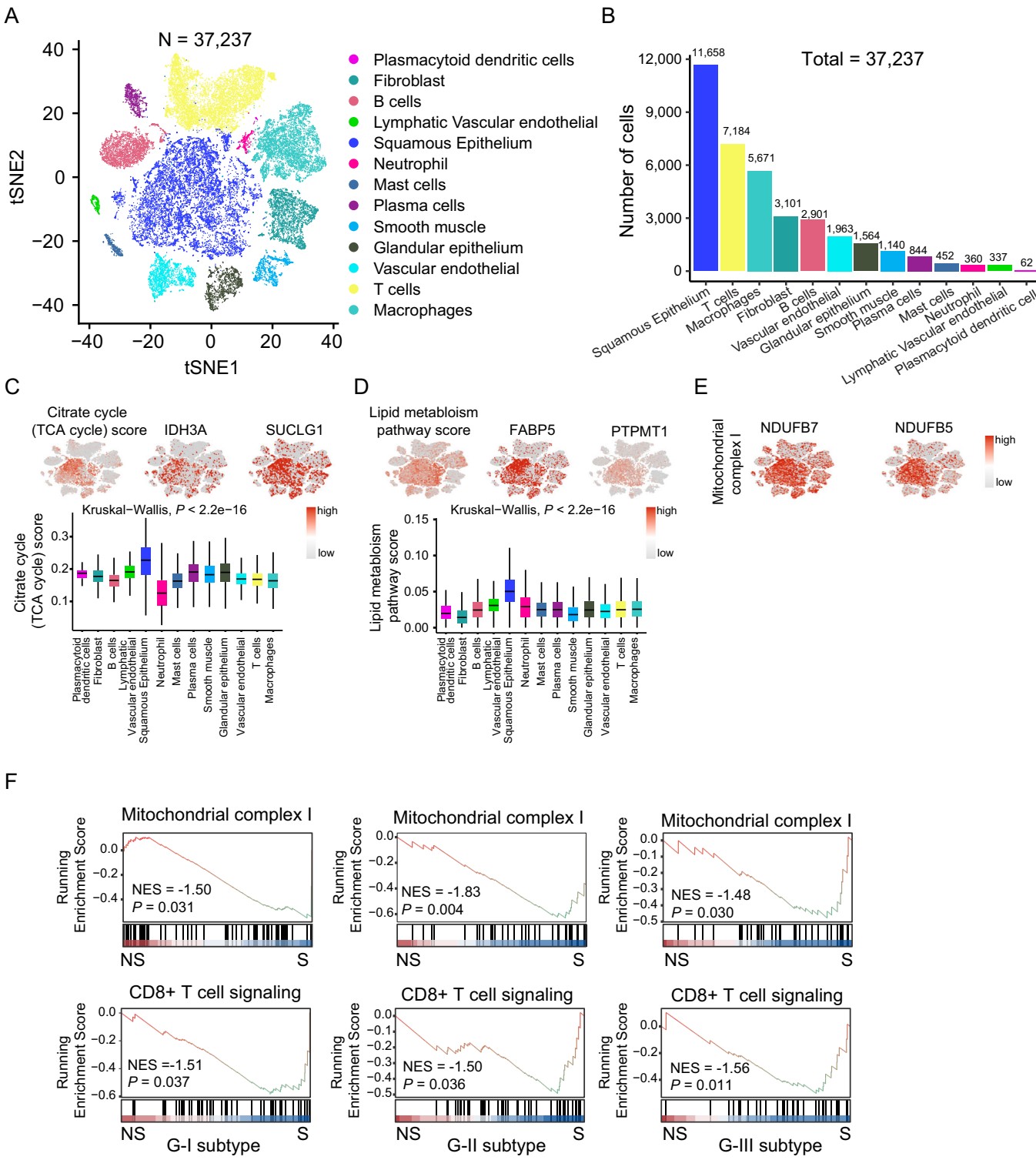

◀ **Figure EV3. The scRNA-seq analysis of ESCC from Pan et al.**

(A) The t-SNE map of the single-cell ESCC landscape was colored by major cell subtypes. (B) The number of each cell subtype in the ESCC tumor microenvironment. (C–E) The t-SNE maps of individual cell AUC score overlay for selected pathway activity and the protein involved in the pathways (upper), and the boxplots show the AUC score of selected pathways in each cell subtype (lower) (two-sided Kruskal–Wallis test). Boxplots represent the interquartile range (IQR), with the box spanning the 25th to 75th percentiles and the median indicated by a horizontal line. Whiskers extend to the most extreme data points within 1.5×IQR. $n = 62$ (plasmacytoid dendritic cells), 3101 (fibroblast), 2901 (B cells), 337 (lymphatic vascular endothelial), 11,658 (squamous epithelium), 360 (neutrophil), 452 (mast cells), 844 (plasma cells), 1140 (smooth muscle), 1564 (glandular epithelium), 1963 (vascular endothelial), 7184 (T cells), and 5671 (macrophages). (F) Pathway enrichment of mitochondrial complex I and CD8 + T cell signaling pathways between S and NS groups in three proteomic subtypes (phenotype-based permutation test).

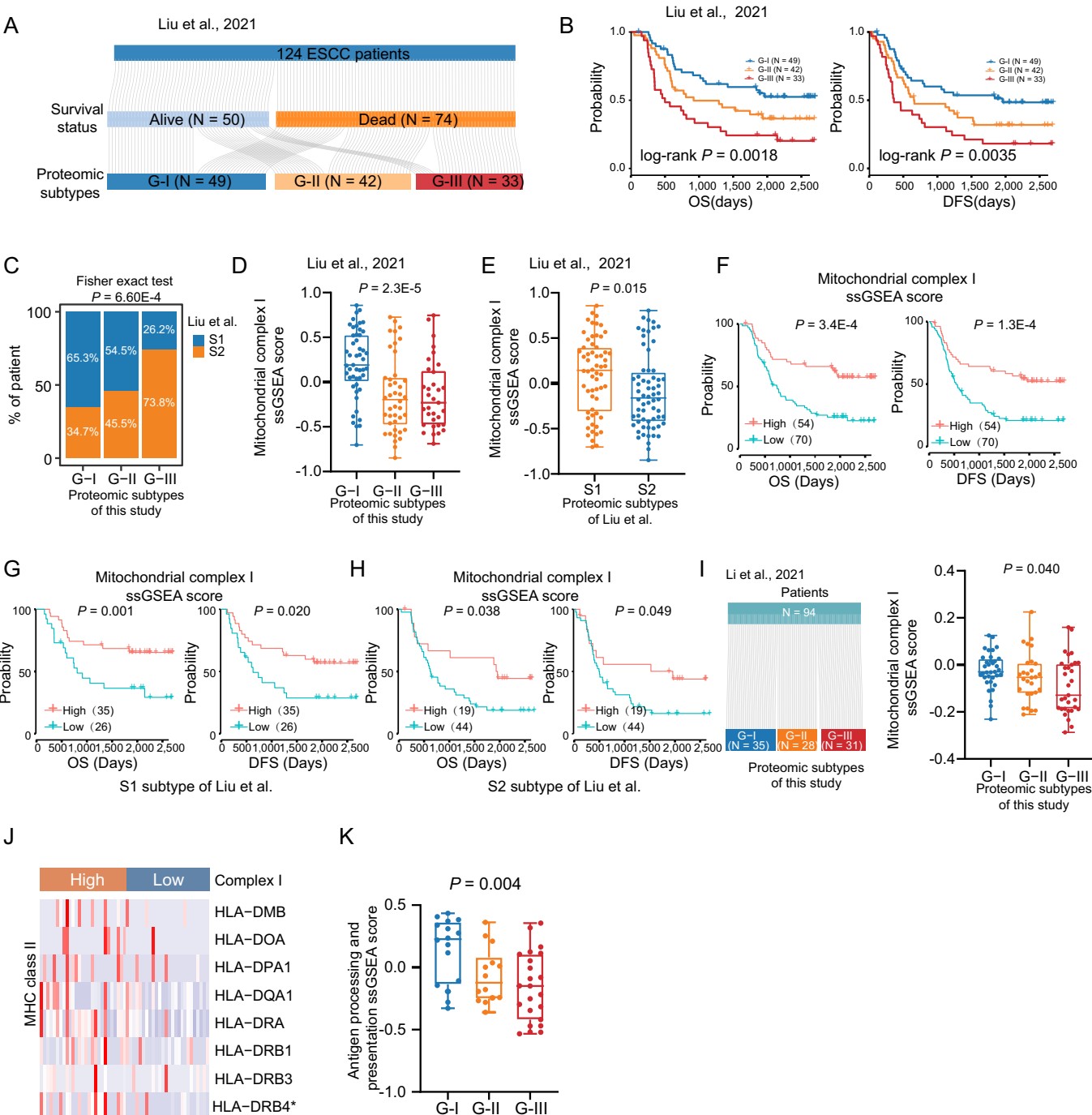

◀ **Figure EV4. The validation of proteomics subtypes in other ESCC cohorts.**

(A) Clustering of 124 ESCC patients using the proteomic data from Liu et al by the proteomic subtyping algorithm of this study. (B) Survival analysis (OS and DFS) of ESCC patients from Liu et al classified by the proteomic subtyping algorithm of this study (*P* value from two-sided log-rank test). (C) Barplot showing the S1 and S2 subtypes of Liu et al distribution among the three proteomic subtypes of this study (two-sided Fisher's exact test). (D) Boxplots showing mitochondrial complex I ssGSEA score across the three proteomic subtypes in the study of Liu et al (ANOVA test). Boxplots represent the interquartile range (IQR), with the box spanning the 25th to 75th percentiles and the median indicated by a horizontal line. Whiskers mark minimum or maximum values. $n = 49$ (G-I), 42 (G-II), and 33 (G-III). (E) Boxplots showing mitochondrial complex I ssGSEA score between S1 ($n = 61$) and S2 ($n = 63$) subtypes from Liu et al (two-sided Student's *t* test). Boxplots are defined as in (D). (F) Survival analysis (OS and DFS) of mitochondrial complex I ssGSEA score in the study of Liu et al (two-sided log-rank test). (G, H) Survival analysis (OS and DFS) of mitochondrial complex I ssGSEA score in the S1 and S2 subtypes identified by Liu et al, respectively (two-sided log-rank test). (I) The application of our proteomic subtypes in the study of Li et al. Boxplots displaying mitochondrial complex I ssGSEA score across the three proteomic subtypes in the study of Li et al (ANOVA test). Boxplots are defined as in (D). $n = 35$ (G-I), 28 (G-II), and 31 (G-III). (J) Differential expression of MHC class II proteins between mitochondrial complex I high and low groups (two-sided Wilcoxon rank-sum test). *$P < 0.05$. (K) Boxplot shows the ssGSEA score of the antigen processing and presentation pathway among three proteomic subtypes (ANOVA test). Boxplots are defined as in (D). $n = 16$ (G-I), 14 (G-II), and 23 (G-III).

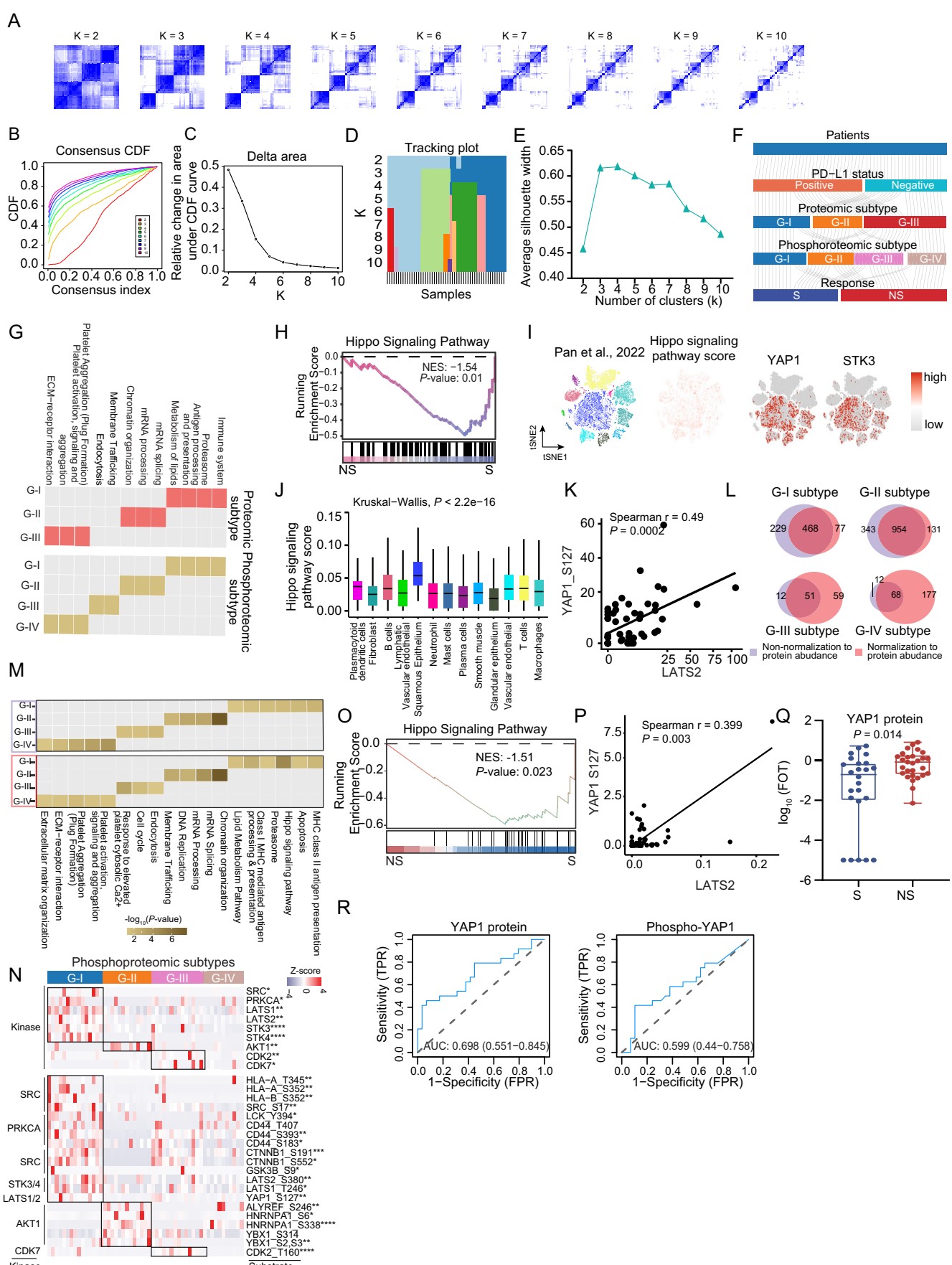

◀ **Figure EV5. eusoft-did-translate-comment-enPhosphoproteomic subtyping of ESCC immunotherapy cohort and correlation with immunotherapy response.**

(A–D) The consensus clustering analysis of 53 ESCC samples and four subtypes was generated. $K$ was tested from 2 to 10 (A). Consensus matrices, as well as the consensus cumulative distribution function (CDF) plot (B), delta area (change in CDF area) plot (C), and tracking plot (D) are shown. (E) Average silhouette-width of identified subtypes ($K = 2$–10). The average silhouette width takes the maximum value at the 4 subtypes ($K = 4$). (F) Sankey plot showing the flow of the ESCC patients featured with different PD-L1 expression status to proteomic subtypes, phosphoproteomic subtypes, and immunotherapy response groups. (G) The comparison of pathway enrichment between proteomic subtypes and phosphoproteomic subtypes. (H) GSEA enrichment plot of the hippo signaling pathway in S and NS groups at the phosphoproteome level. $P$ value from phenotype-based permutation test. (I) The t-SNE maps of individual cell AUC score overlay for hippo signaling pathway score using the scRNA-seq data from Pan et al, and the protein involved in the hippo signaling pathway. (J) The boxplots show the AUC score of the hippo signaling pathway in each cell cluster (two-sided Kruskal–Wallis test). Boxplots represent the interquartile range (IQR), with the box spanning the 25th to 75th percentiles and the median indicated by a horizontal line. Whiskers extend to the most extreme data points within 1.5×IQR. $n = 62$ (plasmacytoid dendritic cells), 3101 (fibroblast), 2901 (B cells), 337 (lymphatic vascular endothelial), 11,658 (squamous epithelium), 360 (neutrophil), 452 (mast cells), 844 (plasma cells), 1140 (smooth muscle), 1564 (glandular epithelium), 1963 (vascular endothelial), 7184 (T cells), and 5671 (macrophages). (K) Spearman correlation of LATS2 and its phosphorylated substrate YAP1 S127. $P$ value was from a two-sided Spearman's correlation test. (L) The overlap of significant expression phosphoproteins with or without normalization to the corresponding protein abundance in each phosphoproteomic subtype. (M) Comparison of pathway enrichment with or without normalization to the corresponding protein abundance. (N) Heatmap showing the kinases and their substrates in each phosphoproteomic subtype after normalization (two-sided Wilcoxon rank-sum test). *$P < 0.05$, **$P < 0.01$, ***$P < 0.001$, ****$P < 0.0001$. (O) GSEA enrichment plot of the hippo signaling pathway in S and NS groups at the phosphoproteome level after normalization. $P$ value from phenotype-based permutation test. (P) Spearman correlation of LATS2 and its phosphorylated substrate YAP1 S127 after normalization to its protein abundance. $P$ value was from two-sided Spearman's correlation test. (Q) Boxplot showing the YAP1 protein expression in S ($n = 24$) and NS ($n = 29$) patients (two-sided Wilcoxon rank-sum test). Boxplots represent the interquartile range (IQR), with the box spanning the 25th to 75th percentiles and the median indicated by a horizontal line. Whiskers mark minimum or maximum values. (R) Area under receiver operating characteristic (ROC) curves of YAP1 protein and phosphorylation in the prediction of ESCC immunotherapy response.

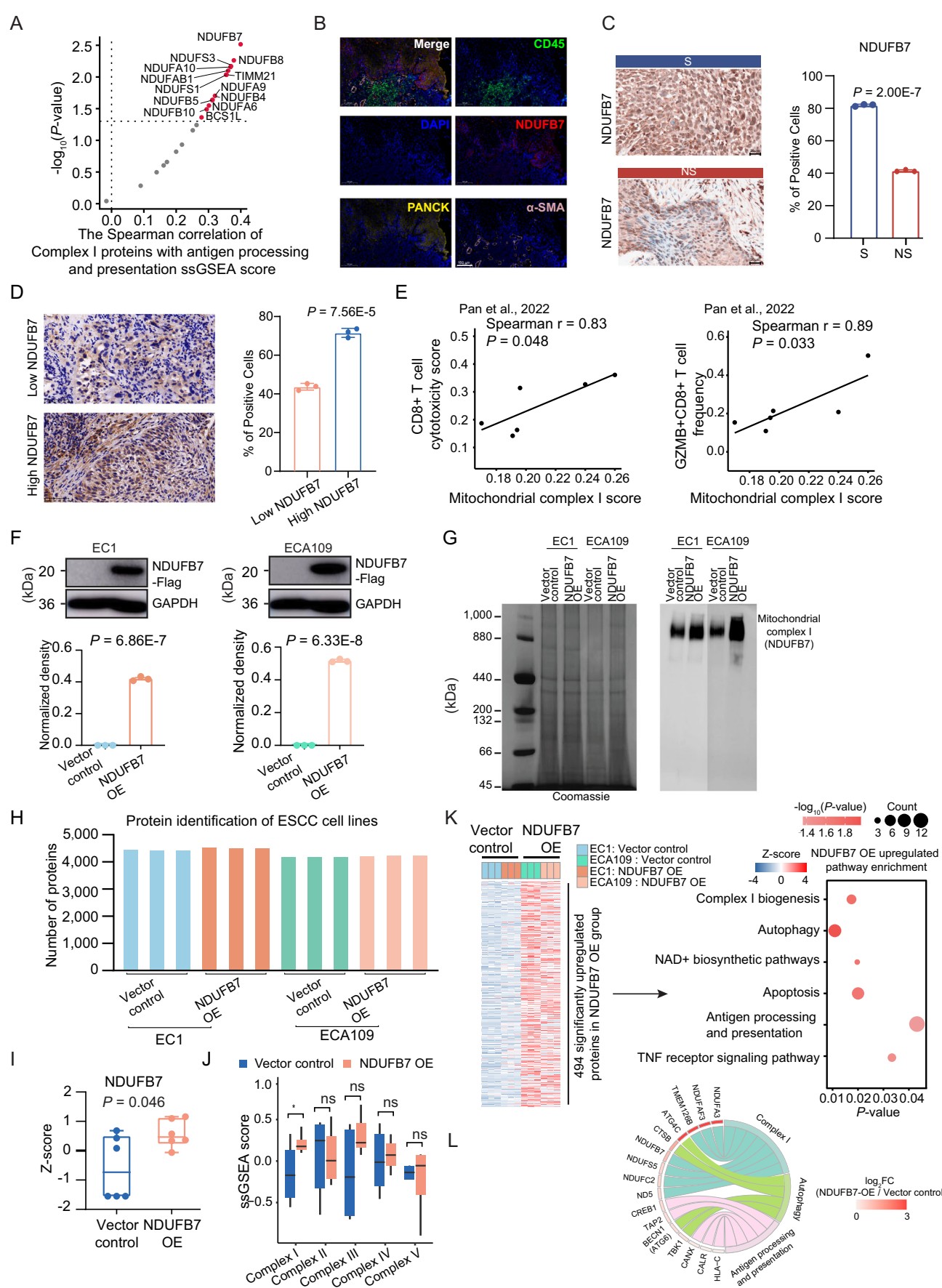

◀ **Figure EV6. A possible mechanism by which mitochondrial complex I modulates immunotherapy sensitivity via CD8 + T cell-mediated killing in vitro.**

(A) Volcano showing the Spearman correlation of mitochondrial complex I proteins with antigen processing and presentation ssGSEA score (two-sided Spearman's correlation test). (B) Representative images of multiplex immunofluorescence staining for NDUFB7, PANCK, CD45, and α-SMA in ESCC biopsy tissues. (C) The representative images of immunohistochemistry (IHC) staining of NDUFB7 expression in sensitive (S) and non-sensitive (NS) groups. Boxplot showing the qualification of NDUFB7 stained by immunohistochemistry (IHC) in the representative samples in the S and NS groups ($n = 3$ independent experiments, two-sided Student's $t$ test, mean with standard deviation). Scale bar: 20 μm. (D) ESCC tumor with different expression of NDUFB7 determined by IHC ($n = 3$ independent experiments, two-sided Student's $t$ test, mean with standard deviation). (E) Correlation between mitochondrial complex I score and CD8 + T cells cytotoxicity score or GZMB + CD8 + T cell frequency in ESCC patients in scRNA-seq data of Pan et al (two-sided Spearman correlation test). (F) Immunoblot of NDUFB7 (tagged with a Flag) and GAPDH in EC1 and ECA109 cells, and the normalization of qualified western blots ($n = 3$ independent experiments, two-sided Student's $t$ test, mean with standard deviation). (G) Analysis of mitochondrial complex I by BN-PAGE and immunoblotting in NDUFB7 OE ESCC cells. Coomassie staining was used as a loading control. (H) Barplots showing the numbers of identified proteins in EC1 and ECA109 cells with overexpression of NDUFB7 or vector control. (I) Boxplots showing the differential expression of NDUFB7 between NDUFB7 OE ($n = 6$) and vector control ($n = 6$) groups (two-sided Student's $t$ test). Boxplots represent the interquartile range (IQR), with the box spanning the 25th to 75th percentiles and the median indicated by a horizontal line. Whiskers mark minimum or maximum values. (J) Boxplot of the multiple mitochondrial complexes ssGSEA score between NDUFB7 OE ($n = 6$) and vector control ($n = 6$) groups (two-sided Wilcoxon rank-sum test). Boxplots are defined as in (I). $P = 0.047$ (Complex I), 0.0944 (Complex II), 0.098 (Complex III), 0.333 (Complex IV), and 0.918 (Complex V). *$P < 0.05$, ns not significant. (K) Heatmap showing the differential expression of the overrepresented proteins in the NDUFB7 OE ESCC cells (left). The bubble plot shows the significantly enriched pathways in NDUFB7 OE ESCC cells (right) (hypergeometric test). (L) Circular plot showing differentially expressed proteins involved in mitochondrial complex I, autophagy, and antigen processing and presentation pathways between NDUFB7 OE and vector control groups.

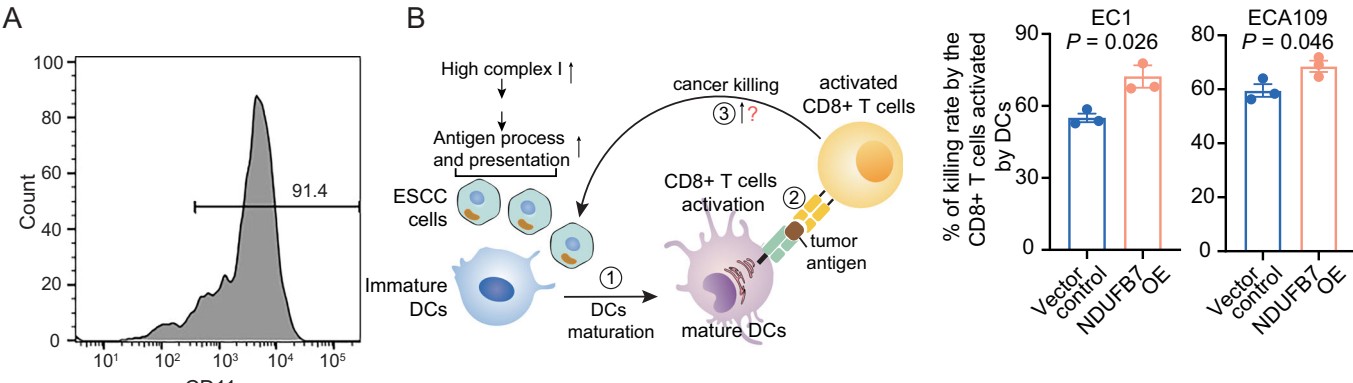

**Figure EV7. Mitochondrial complex I increases the anti-tumor ability of CD8 + T cells activated by dendritic cells.**

(A) Flow cytometry histogram representing the percentage of DCs isolated from PBMC measured by their surface markers CD11c. (B) Diagram showing the co-cultured system composed of ESCC cells, DCs, and activated CD8 + T cells. Boxplots showing the killing effect of CD8 + T cells activated by DCs co-cultured with NDUFB7 OE ESCC cells and vector controls ($n = 3$ independent experiments, two-sided Student's $t$ test, mean with standard deviation).

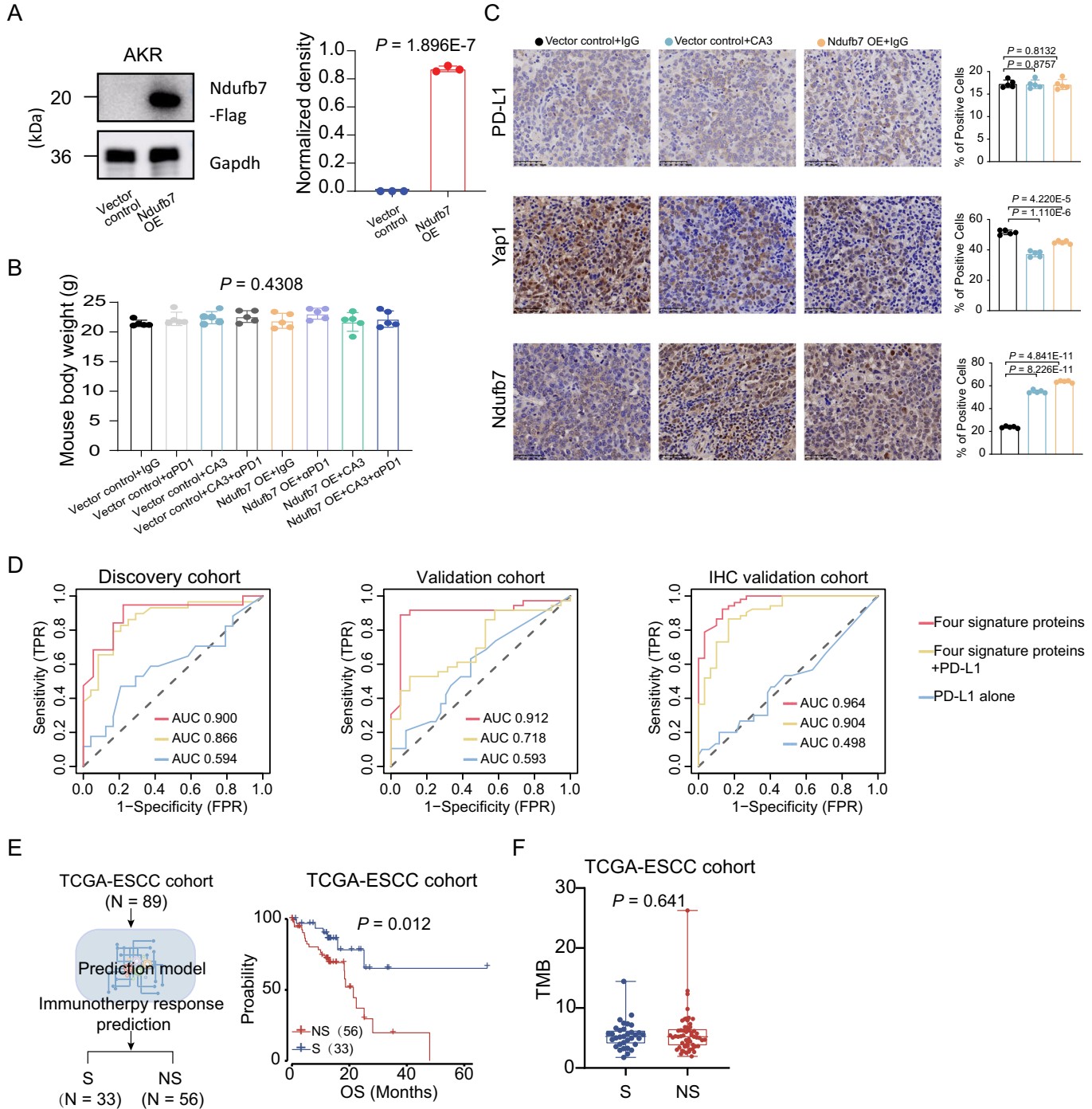

**Figure EV8.   The mitochondrial complex I of ESCC increases the therapeutic efficacy of anti-PD1 treatment in vivo.**

(**A**) Immunoblot analysis of Ndufb7 (tagged with a Flag) and Gapdh in mouse ESCC cell AKR, and the normalization of qualified western blots ($n = 3$ independent experiments, two-sided Student's $t$ test, mean with standard deviation). (**B**) The mouse body weights in the AKR allografts model ($n = 5$/group, ANOVA test, mean with standard deviation). (**C**) Representative IHC staining images and the quantification of the positive cells of PD-L1, Ndufb7, and Yap1 in the vector control tumor, vector control tumor with CA3 treatment, and Ndufb7 OE tumor ($n = 5$/group, ANOVA test, mean with standard deviation). (**D**) The ROC curves of the four signature proteins with or without the combination of PD-L1 in the discovery cohort, validation cohort, and IHC validation cohort, respectively. (**E**) The predicted result of immunotherapy response for ESCC patients in the TCGA-ESCC cohort using our predictive model. (**F**) Boxplots showing the differences of TMB level between S ($n = 33$) and NS ($n = 56$) patients in the TCGA-ESCC cohort (two-sided Student's $t$ test). Boxplots represent the interquartile range (IQR), with the box spanning the 25th to 75th percentiles and the median indicated by a horizontal line. Whiskers mark minimum or maximum values.

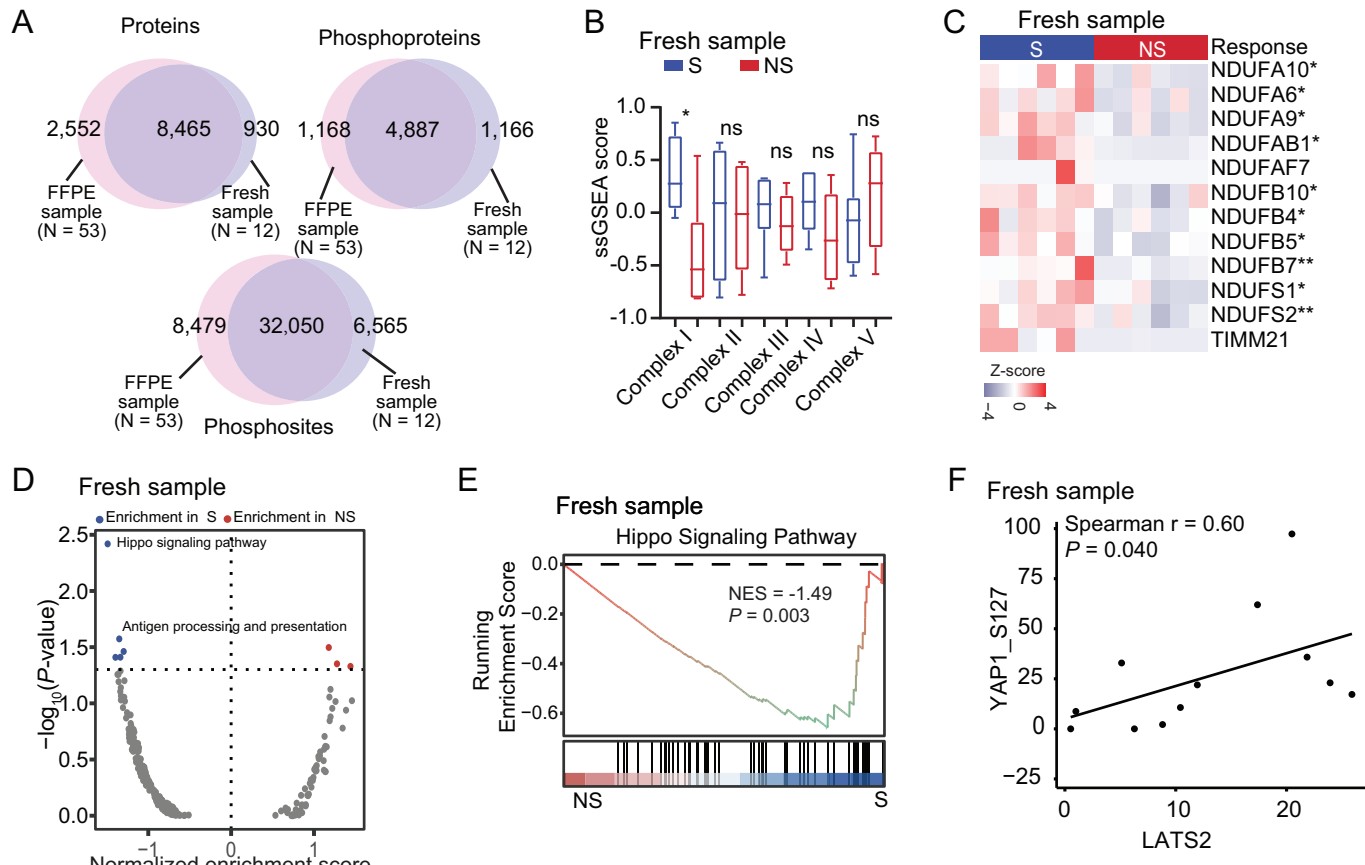

**Figure EV9.  The comparative results of proteomic and phosphoproteomic analysis from fresh tissues.**

(A) The comparison of protein, phosphoprotein, and phosphosite identification between FFPE and fresh tissues. (B) The ssGSEA score of mitochondrial complexes I–V between S ($n = 6$) and NS ($n = 6$) patients with fresh tissues. $P$ value from two-sided Wilcoxon rank-sum test. Boxplots represent the interquartile range (IQR), with the box spanning the 25th to 75th percentiles and the median indicated by a horizontal line. Whiskers mark minimum or maximum values. $P = 0.026$ (Complex I), 0.937 (Complex II), 0.485 (Complex III), 0.240 (Complex IV), and 0.481 (Complex V). *$P < 0.05$, ns not significant. (C) Heatmap showing mitochondrial complex I proteins in S and NS patients with fresh tissues (two-sided Wilcoxon rank-sum test). *$P < 0.05$, **$P < 0.01$, ***$P < 0.001$. (D) Volcano plot showing the pathway enrichment between S and NS groups based on the phosphoproteome by GSEA analysis in fresh tissues. $P$ value from Phenotype-based permutation test. (E) GSEA enrichment plot of the hippo signaling pathway in S patients in fresh tissues. $P$ value from Phenotype-based permutation test. (F) Spearman correlation of LATS2 and its phosphorylated substrate YAP1 S127 in fresh tissues. $P$ value was from two-sided Spearman's correlation test.

