## [Peer Review File · EMBO Molecular Medicine]

Proteome profiles of esophageal squamous cell carcinoma tie mitochondrial complex I to immunotherapy

Fahan Ma, Yan Li, Chan Xiang, Bing Wang, Jie Lv, Zhanxian Shang, Weiguang Zhang, Zhaoyu Qin, Yan Pu, Kai Li, Jinzhi Wei, Su-Bei Tan, Jinwen Feng, Haohua Teng, Peipei Zhang, Jiaying Deng, Yunzhi Wang, Chao Zhang, Sha Tian, Guichao Li, Mingqiang Kang, Changsheng Du, Yuchen Han and Chen Ding

Corresponding authors: Chen Ding (chend@fudan.edu.cn) , Yuchen Han (ychan@cmu.edu.cn), Changsheng Du (duchangsheng@tongji.edu.cn), Mingqiang Kang (9199115045@fjmu.edu.cn)

Review Timeline:

Submission Date:	9th Jan 26
Editorial Decision:	12th Feb 26
Revision Received:	16th Mar 26
Accepted:	18th Mar 26

Editor: Jingyi Hou

Transaction Report:

Please note that the manuscript was previously reviewed at another journal and the reports were taken into account in the decision making process at EMBO Molecular Medicine. Since the original reviews are not subject to EMBO Press' transparent review process policy, the reports and author response cannot be published.

Note: With the exception of the correction of typographical or spelling errors that could be a source of ambiguity, letters and reports are not edited. Depending on transfer agreements, referee reports obtained elsewhere may or may not be included in this compilation. Referee reports are anonymous unless the Referee chooses to sign their reports.

12th Feb 2026

Dear Chen,

Thank you for submitting your revised manuscript. I apologize for the delay in providing the formal decision, as the formatting check has taken longer than anticipated.

We have received feedback from a mitochondrial biology expert who was specifically asked to evaluate the Complex I and mitochondrial-related aspects of the study. The reviewer provided a very thorough assessment and raised several concerns that largely overlap with those previously noted by reviewers at the other journal.

In particular, the reviewer recommends substantial textual revisions (in their report and comments to the editor), including:

1. Complete removal of the Complex I activity analyses, along with the associated conclusions; and
2. Complete removal of conclusions linking Complex I to autophagy and YAP/TAZ signaling, as these interpretations are not sufficiently supported by the current data.

More broadly, the reviewer considers it appropriate and well supported to carefully correlate Complex I expression levels (protein or RNA) with cell, tumor, or patient subgroups, but advises against extrapolating these findings to underlying mitochondrial mechanisms. We would therefore ask you to revise the manuscript in line with these recommendations. Please submit the revised version with track changes, as we need to review all the modifications.

On a more editorial level, please do the following:

1. Remove "Authors' contribution" section from the manuscript file.
2. Figures:
 - Main figures and supplementary figures should be uploaded as separate, high-resolution files.
 - Supplementary figures should be renamed as Figure EV1, Figure EV2, etc.
 - Please note that Figures 4, 5, 6, and 7 contain blots and microscopy images that appear highly pixelated. Kindly provide higher-resolution versions, if possible.
3. Please provide Table 1 in an editable format.
4. "Supplementary tables 1 - 8" should be renamed to Dataset EV1 - EV8. Please include the legend within the Dataset with a separate sheet labelled 'Legend'. The callouts need to be updated accordingly.
5. Please resolve the name discrepancy between Subei Tan (manuscript) and Su-Bei Tan (submission system). Additionally, provide the institutional email addresses for the co-corresponding authors, Prof. Kang and Prof. Du.
6. "Declarations of interest" should be renamed to "Disclosure and Competing Interests Statement"
7. Please download and fill our Reagents and Tools Table template (.docx), which you can find in our author guidelines: <https://link.springer.com/journal/44320/submission-guidelines#structuredmethods>. When submitting your revised manuscript, please DO NOT include the Reagents and Tools Table in the Methods section of the manuscript but upload it as a separate file choosing the file type "Reagent Table".
8. The funding information appears incomplete in the submission system. Please ensure that the funding details entered online are consistent with those listed in the manuscript.
9. At EMBO Press we ask authors to provide source data for the main manuscript figures. You will receive a separate email with instructions for providing source data with your revised manuscript, including how to upload and organize the files.
10. Figure callouts: Please ensure that all figure panels are called out sequentially. Currently, Suppl. Fig. S4A-E is called out before Suppl. Fig. S3, and Suppl. Fig. S8A is called out before Suppl. Fig. S7F-N. This ordering needs to be corrected.
11. The references need to be formatted according to the EMBO Molecular Medicine reference style. Citations should be listed in alphabetical order. Please list up to 10 co-authors of a paper before adding et al. in the reference list. Remove DOIs for all published articles.
12. "Data and materials availability" should be renamed to "Data availability". Please provide accession IDs for the iprox datasets. Please remove the reviewer access codes and make sure the datasets will be made publicly available upon the

acceptance of the manuscript.

13. Please provide a 'Synopsis' to further enhance discoverability. Synopses are displayed on the journal webpage and are freely accessible to all readers. They include a short stand first (maximum of 300 characters, including space) as well as 2-5 one-sentences bullet points that summarizes the paper. Please write the bullet points to summarize the key NEW findings. They should be designed to be complementary to the abstract - i.e. not repeat the same text. We encourage inclusion of key acronyms and quantitative information (maximum of 30 words / bullet point). Please use the passive voice. Please attach these in a separate file or send them by email, we will incorporate them accordingly.

Please provide visual abstract to illustrate your article as a PNG file 550 px wide x 300-600 px high.

14. The paper explained: EMBO Molecular Medicine articles are accompanied by a summary of the articles to emphasize the major findings in the paper and their medical implications for the non-specialist reader. Please provide a draft summary of your article highlighting

15. Please enter the names of all co-correspondence authors into the Author checklist.

16. Correct the order and headings of the manuscript sections to: Abstract / The Paper Explained / Introduction / Results / Discussion / Methods / Data Availability / Acknowledgements / Disclosure and Competing Interests Statement / References / Figure Legends / Tables / Expanded View Figure Legends.

17. Please address the following issues related to figure legends:

- Please define the annotated p values ****/**/*/* as well as provide the exact p-values for the same in the legend of figure 2H as appropriate.
- Please note that the exact p values are not provided in the legends of figures 1I, 2I, 3D, L; 4H, 5E, H, J, K; 6D
- Please note that the box plots need to be defined in terms of minima, maxima, centre, bounds of box and whiskers, and percentile in the legends of figures 1F, I; 2H, I; 3D, J, L; 4F, 5E, 6G,7E
- Please note that information related to n is missing in the legends of figures 1F, I; 2I, 3D, J, L; 4F, J, N, O; 5B, C, D, E, F, J-M; 6G
- Please note that the error bars are not defined in the legends of figures 4J, N, O; 5B, C, D, F, J-M

Please feel free to let me know if you have any questions.

I look forward to reading a new revised version of your manuscript as soon as possible.

Sincerely,
Jingyi

Jingyi Hou
Senior Editor
EMBO Molecular Medicine

*** Instructions to submit your revised manuscript ***

In the event of acceptance, this file will be published in conjunction with your paper and will include the anonymous referee reports, your point-by-point response and all pertinent correspondence relating to the manuscript. If you do NOT want this file to

be published, please inform the editorial office at contact@embomolmed.org.

**** Reviewer's comments ****

Referee #1 (Comments on Novelty/Model System for Author):

I was asked to specifically evaluate the Complex I/mitochondrial biology part. It should be noted that the activity assays to measure Complex I are not reliable, which is a point previously raised by Reviewer 3 and 4.

Referee #1 (Remarks for Author):

I was asked for my input regarding the conclusions related to mitochondrial biology, mainly the section "High mitochondrial complex I protein expression of ESCC cells and tumor organoids enhances tumor killing by activated CD8⁺ T cells:

I have read the revised manuscript and the point-by-point response. Overall, I share the same concerns raised by reviewer 3 and 4, which have not been fully resolved and therefore cast doubt as to the ability of the authors to lay claim to the role of Complex I beyond it being a faithful biomarker for the subclassification of patient tumor types. Indeed, the colorimetric assay used to measure complex I in the main and supplemental figures is not a standard in the field even though it has been sold and cited as such, and this assay is subject to all the caveats acutely raised by the previous reviewers. The seahorse assay they propose to corroborate these results simply examine the impact on oxygen consumption rates between basal respiration and OCR following the inhibition of complex I with rotenone. In my opinion, it does not directly address the activity of complex I directly as it is not known the OXPHOS system contributes to OCR at baseline. There are several pathways that could positively and negatively contribute to said OCR. Rotenone inhibition is routinely used in conjunction with succinate to favor Complex II-mediated respiration since the oxaloacetate (produced by the TCA and therefore indirectly Complex I) inhibits its activity. Hence, the data provided by the authors support the proteomic changes in Complex I steady state levels but not the activity measurements.

Of note, the BN-PAGE analyses of over-expressing NDUF7 are rather convincing, as they show an increase in steady-state Complex I levels as well as the corresponding subunits (by proteomics) in ECA109 NDUF7-oe cells (despite unequal loading in Figure S8A) and to a lesser extent in EC1 cells. In both cell lines, Complex I activity, measured by the aforementioned possibly unreliable assay, showed a corresponding increase, which would be entirely consistent with increased enzymatic activity and their model, provided that Complex I activity was limited by assembly in the absence of over-expression.

More broadly, I believe the authors are on solid ground to correlate very carefully the Complex I expression (protein or RNA) and cell/tumor/patient subgroups, without extrapolating on the underlying mitochondrial biology, as this would require that the assumptions regarding the rate limiting steps of Complex I activity be regulated at the transcriptional (scRNAseq, ssGEA) and post-translational (proteomics) levels, which is unknown and undemonstrated. Moreover, it would assume that the increased activity of Complex I they purport is in fact truly reliable. Beyond this assumption, how modulation of Complex I is related to the cell and tumor biology effects they report in terms of molecular mechanism and cause/consequence cannot be determined with the existing data provided in the manuscript. The link between Complex I protein levels and autophagy or YAP/TAZ is also unclear, although it should be noted the hypotheses they put forward can neither proven nor disproven at this time. However, in the manuscript, they cite Reference 64 to support the link between Complex I and YAP1. Said paper does not support their research. In Ref 64, the paper correlates altered metabolism (and overall OXPHOS) and ROS in response to YAP/TAZ - seahorse studies show a general alteration of bioenergetics and not simply Complex I. In the current manuscript, the authors endeavor to associate a specific Complex I phenotype with YAP1 co-regulation, which is not the same.

Editor's comments

In particular, the reviewer recommends substantial textual revisions (in their report and comments to the editor), including:

1. Complete removal of the Complex I activity analyses, along with the associated conclusions; and
2. Complete removal of conclusions linking Complex I to autophagy and YAP/TAZ signaling, as these interpretations are not sufficiently supported by the current data.

More broadly, the reviewer considers it appropriate and well supported to carefully correlate Complex I expression levels (protein or RNA) with cell, tumor, or patient subgroups, but advises against extrapolating these findings to underlying mitochondrial mechanisms. We would therefore ask you to revise the manuscript in line with these recommendations. Please submit the revised version with track changes, as we need to review all the modifications.

Response:**(1) As for the removal of Complex I activity analyses.**

We sincerely appreciate your dedicated support and guidance in refining our manuscript. According to your comments, in the revision, we have removed the description of mitochondrial complex I activity analyses in the “**Phosphoproteomic subtyping of ESCC immunotherapy cohort reveals the association of hippo pathway with ESCC immunotherapy**” and “**High mitochondrial complex I protein expression of ESCC cells and tumor organoids enhances tumor killing by activated CD8+ T cells**” of the Results section. The corresponding methodological descriptions and associated figures have also been comprehensively removed in the revised version.

(2) As for the removal of conclusions linking Complex I to autophagy and YAP/TAZ signaling.

We sincerely appreciate your constructive comments, which have been instrumental in enhancing the clarity and scientific rigor of our work. According to your comments, in the revision, we have made comprehensive revisions throughout the manuscript. Specifically, all

statements and interpretations suggesting a regulatory role of mitochondrial complex I in autophagy have been removed from the Results section (“**The mitochondrial complex I enhances ESCC immunotherapy sensitivity via upregulating antigen processing and presentation**”). Similarly, any conclusions implying a link between YAP1 and mitochondrial complex I have been fully removed from the “**Phosphoproteomic subtyping of ESCC immunotherapy cohort reveals the association of hippo pathway with ESCC immunotherapy**” section. These changes have been consistently applied across the manuscript, including in the Results, Discussion, and Figure Legends, to ensure the narrative is now exclusively supported by the data. All figures associated with these results have also been removed.

Reviewer's comments

Referee #1 (Comments on Novelty/Model System for Author):

I was asked to specifically evaluate the Complex I/mitochondrial biology part. It should be noted that the activity assays to measure Complex I are not reliable, which is a point previously raised by Reviewer 3 and 4.

Referee #1 (Remarks for Author):

I was asked for my input regarding the conclusions related to mitochondrial biology, mainly the section "High mitochondrial complex I protein expression of ESCC cells and tumor organoids enhances tumor killing by activated CD8⁺ T cells:

I have read the revised manuscript and the point-by-point response. Overall, I share the same concerns raised by reviewer 3 and 4, which have not been fully resolved and therefore cast doubt as to the ability of the authors to lay claim to the role of Complex I beyond it being a faithful biomarker for the subclassification of patient tumor types. Indeed, the colorimetric assay used to measure complex I in the main and supplemental figures is not a standard in the field even though it has been sold and cited as such, and this assay is subject to all the caveats acutely raised by the previous reviewers. The

seahorse assay they propose to corroborate these results simply examine the impact on oxygen consumption rates between basal respiration and OCR following the inhibition of complex I with rotenone. In my opinion, it does not directly address the activity of complex I directly as it is not known the OXPHOS system contributes to OCR at baseline. There are several pathways that could positively and negatively contribute to said OCR. Rotenone inhibition is routinely used in conjunction with succinate to favor Complex II-mediated respiration since the oxaloacetate (produced by the TCA and therefore indirectly Complex I) inhibits its activity. Hence, the data provided by the authors support the proteomic changes in Complex I steady state levels but not the activity measurements.

Of note, the BN-PAGE analyses of over-expressing NDUF7 are rather convincing, as they show an increase in steady-state Complex I levels as well as the corresponding subunits (by proteomics) in ECA109 NDUF7-oe cells (despite unequal loading in Figure S8A) and to a lesser extent in EC1 cells. In both cell lines, Complex I activity, measured by the aforementioned possibly unreliable assay, showed a corresponding increase, which would be entirely consistent with increased enzymatic activity and their model, provided that Complex I activity was limited by assembly in the absence of over-expression.

More broadly, I believe the authors are on solid ground to correlate very carefully the Complex I expression (protein or RNA) and cell/tumor/patient subgroups, without extrapolating on the underlying mitochondrial biology, as this would require that the assumptions regarding the rate limiting steps of Complex I activity be regulated at the transcriptional (scRNAseq, ssGEA) and post-translational (proteomics) levels, which is unknown and undemonstrated. Moreover, it would assume that the increased activity of Complex I they purport is in fact truly reliable. Beyond this assumption, how modulation of Complex I is related to the cell and tumor biology effects they report in terms of molecular mechanism and cause/consequence cannot be determined with the existing data provided in the manuscript. The link between Complex I protein levels and

autophagy or YAP/TAZ is also unclear, although it should be noted the hypotheses they put forward can neither be proven nor disproven at this time. However, in the manuscript, they cite Reference 64 to support the link between Complex I and YAP1. Said paper does not support their research. In Ref 64, the paper correlates altered metabolism (and overall OXPHOS) and ROS in response to YAP/TAZ - Seahorse studies show a general alteration of bioenergetics and not simply Complex I. In the current manuscript, the authors endeavor to associate a specific Complex I phenotype with YAP1 co-regulation, which is not the same.

Response:

We sincerely appreciate the reviewer's constructive, rigorous, and professional comments, which have provided valuable insights into the role of mitochondrial complex I and inspired us to focus and make an appropriate claim for our results. According to the reviewer's comments, we have divided the response into two parts: (1) As for the removal of mitochondrial complex I activity analyses; (2) As for the removal of the link between mitochondrial complex I and autophagy, as well as YAP1.

(1) As for the removal of mitochondrial complex I activity analyses.

We are grateful for the reviewer's thorough and constructive comments, which have been invaluable in refining our manuscript. In our work, aiming to identify the biomarkers for immunotherapy response guidance and explore the potential proteomic features associated with immunotherapy, we collected treatment-naïve biopsies from ESCC patients and conducted proteomic profiling. Through proteomic subtyping, we identified three proteomic subtypes associated with ESCC immunotherapy response. In the immunotherapy-sensitive subtype, we found high expression of mitochondrial complex I protein, especially NDUF7. NDUF7 is a subunit for mitochondrial complex I, and its expression was higher in sensitive patients than in non-sensitive patients. *In vitro* assays utilizing ESCC cell lines or patient-derived organoids revealed that elevated NDUF7 expression was associated with increased tumor cell susceptibility to CD8+ T cell-mediated cytotoxicity and improved response to anti-PD1 therapy. Moreover, the enhanced anti-tumor effects associated with NDUF7 expression were consistently confirmed in allogeneic tumor transplantation

experiments. Therefore, we proposed that the expression of mitochondrial complex I in ESCC cells may affect the immunotherapy efficacy.

We sincerely apologize for the statement in the previous version, where we equated the expression levels of mitochondrial complex I with its activity. We agree with the reviewer that the proteomic data primarily reflect the abundance of mitochondrial complex I proteins, rather than its activity. The commercially available colorimetric assay we initially employed, while providing an accessible readout, is not the gold-standard enzymatic assay for mitochondrial complex I and carries well-documented caveats regarding specificity and interpretation. Its results should not be equated with definitive functional enzymatic activity. The Seahorse metabolic analysis (OCR changes upon Rotenone inhibition) offers an important cellular phenotypic readout of electron transport chain perturbation, but as the reviewer astutely notes, it is an indirect and systemic measurement. The mitochondrial complex I function is very complicated and integrates contributions from various pathways (e.g., Complex II-mediated respiration favored under specific substrate conditions, as mentioned by the reviewer) and does not isolate or directly quantify the activity of mitochondrial complex I itself. In addition, we speculated that NDUFB7 overexpression might influence the assembly of mitochondrial complex I, which was confirmed by the BN-PAGE analyses. These results might reflect an increase in steady-state mitochondrial complex I levels as well as the corresponding subunits. Overall, in the revision, according to the reviewer's comments, we have completely removed mitochondrial complex I activity analyses in the revised manuscript.

(2) As for the removal of the link between mitochondrial complex I and autophagy, as well as YAP1.

To further investigate how mitochondrial complex I may influence immunotherapy response, we explored its potential relationship with autophagy based on correlation analyses of proteomic data. We agree with the reviewer that the current results lack direct experimental evidence to establish a causal link between mitochondrial complex I and autophagy in the context of immunotherapy response. The correlation observed in our proteomic data, while suggestive, does not constitute mechanistic proof. In the revision, the link between

mitochondrial complex I and autophagy has been completely removed.

Additionally, through phosphoproteomic analysis, we further identified that alterations in the phosphorylation cascade of core kinases within the hippo pathway and the phosphorylation status of its effector protein YAP1 represent key signaling events associated with immunotherapy response. Based on the consistency between phosphoproteomic subtypes and proteomic subtypes, as well as the correlation analyses, we speculated the regulation of YAP1 on mitochondrial complex I. However, the regulation of YAP1 on mitochondrial complex I is unclear. Further *in vitro* experiments in ESCC cells demonstrated that the interaction between YAP1 and TEAD1 upregulates the transcriptional expression of DNAJC15, which in turn exerts an inhibitory effect on mitochondrial complex I. We agree with the reviewer that the existing data in the manuscript is not sufficient to determine the molecular mechanisms and cause. The expression of YAP1 might generate general alterations of bioenergetics and not simply mitochondrial complex I. Therefore, in the revision, according to the reviewer comments, we have completely removed the statement of the link between YAP1 and mitochondrial complex I. We are grateful for the reviewer's insight, which has helped to significantly improve the focus and accuracy of our work and provide a clear roadmap for future investigation.

18th Mar 2026

Dear Chen,

We are pleased to inform you that your manuscript is accepted for publication and is now being sent to our publisher to be included in the next available issue of EMBO Molecular Medicine.

You may qualify for financial assistance for your publication charges - either via a Springer Nature fully open access agreement or an EMBO initiative. Check your eligibility: <https://link.springer.com/journal/44321/how-to-publish-with-us>

Kind regards,
Jingyi

Jingyi Hou
Senior Editor
EMBO Molecular Medicine

>>> Please note that it is EMBO Molecular Medicine policy for the transcript of the editorial process (containing referee reports and your response letter) to be published as an online supplement to each paper. If you do NOT want this, you will need to inform the Editorial Office via email immediately. More information is available here: <https://link.springer.com/partners/embo-press/editorial-policies#Peer%20review>